# A Dynamic Multiscale Anti-Aliasing Network for Time Series Forecasting

## Abstract

Real-world time series inherently exhibit complex temporal patterns. Within chaotic systems, significant mixing and entanglement occur between different time-varying modes. Given that time series exhibit distinctly different patterns at various sampling scales, downsampling to extract multiscale features is a common approach. However, conventional downsampling causes high-frequency components in the original signal, those exceeding the new Nyquist frequency, to undergo spectral folding. This erroneously introduces spurious low-frequency patterns, perceived as low-frequency noise, thereby leading to the ***aliasing problem***. To address this problem, we propose a Decomposition-Prevention-Fusion architecture framework called **DMANet**, which introduces the Dynamic Multiscale Anti-Aliasing Network. Specifically, DMANet comprises two key components: Multiscale Convolutional Downsampling, designed to capture temporal dependencies and inter-channel interactions, and an Anti-Aliasing Operation, which includes Pre-Sampling Anti-Aliasing Filtering and Post-Sampling Interpolation. These designs guarantee the fidelity of multiscale features before and after downsampling. We show that by mitigating the risk of aliasing, our proposed simple convolutional downsampling architecture achieves performance competitive with common baselines and larger Transformer-based models prevalent in existing studies across multiple benchmark datasets. Our codes are available at `https://anonymous.4open.science/r/DMANet-ED7A`.

## 1 Introduction

Time series analysis is widely applied in various fields such as health Morid et al. (2023), economics Sezer et al. (2020), transportation Shu et al. (2021), and weather Volkovs et al. (2024). With the widespread adoption of physical and virtual sensors, vast amounts of time series data are continuously generated, offering unprecedented opportunities for in-depth analysis and modeling. In contrast to image, video, and text data, which often possess defined syntax or intuitive patterns, time series data consist of scalar values continuously recorded at each time point. Semantic information in time series data is mainly derived from temporal changes Wu et al. (2023).

The complexity and non-stationarity inherent in real-world systems mean that observed time series often exhibit intricate temporal patterns (e.g., ascents, descents, fluctuations, sudden drifts). These patterns interact, and such interactions become particularly pronounced in chaotic systems, where significant overlap and aliasing can occur Wu et al. (2024). This challenge intensifies when distinct temporal patterns emerge at multiple scales, resulting in the entanglement of various temporal variations Shang et al. (2024) Kou et al. (2025). Therefore, time series analysis must carefully consider the intricate interactions and dynamic relationships among temporal patterns.

To address the complex time-varying entanglement in time series, an increasing number of studies focus on leveraging prior knowledge to decompose time series into more interpretable and simpler components that provide a basis for forecasting. For example, models such as Autoformer Wu et al. (2021) and Dlinear Zeng et al. (2023) decompose series into seasonal and trend components. TimesNet Wu et al. (2023) and Peri-midformer Wu et al. (2024) leverage the periodicity of time series by dividing long sequences into shorter segments based on period length, enabling separate modeling of inter- and intra-period dependencies. Beyond time-domain decomposition, frequency-domain analysis offers a valuable complementary perspective to understand temporal entanglement. Techniques

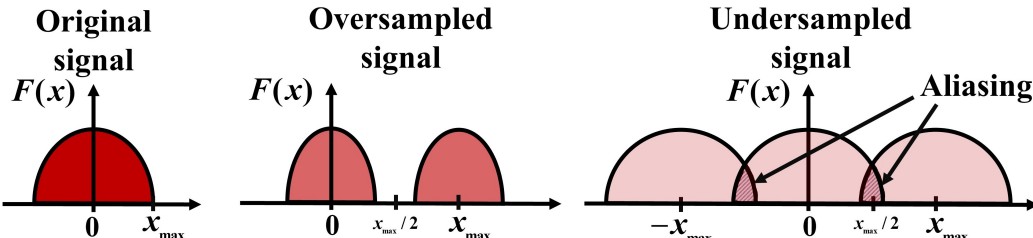

Figure 1: The illustration demonstrates the occurrence of aliasing when sampling a time series signal from the perspective of its frequency spectrum. **Left:**The frequency spectrum of a signal with maximal frequency $x_{max}$. **Center:** After sampling at a sufficiently high rate, replicated spectra do not overlap means that no aliasing occurs. **Right:** After undersampling, spectral replicas overlap, causing aliasing due to mixed frequency components. More details in Appendix.A.3

like the Fourier transform allow signals to be decomposed into orthogonal frequency components, where low frequencies might represent long-term periodic variations and high frequencies capture abrupt events, revealing intrinsic patterns often obscured in the time domain.

However, as time series exhibit distinct temporal patterns at varying sampling scales Wang et al. (2024a), future variations are jointly determined by the interplay of multiple scales Hu et al. (2025) Liu et al. (2025). Despite the effectiveness of the aforementioned methods in decomposing specific aspects, modeling complex time-varying entanglement remains a critical challenge. Increasingly, multiscale decomposition approaches, exemplified by TimeMixer Wang et al. (2024a), aim to model multiscale variations by decomposing them into different temporal granularities. These methods often select downsampling operations, progressively reducing temporal resolution using techniques such as strided convolutions or pooling layers to expand the models' receptive field and capture dependencies across different scales.

However, existing downsampling processes are susceptible to critical ***aliasing risks*** as shown in Figure.1 (see Appendix.A for a detailed explanation). When downsampling operators such as strided convolutions or pooling are used, high-frequency components of the original signal that exceed the new Nyquist frequency undergo spectral folding Shannon (1949); Nyquist (1928). If undersampled, these folded components are incorrectly represented as spurious low-frequency patterns, compromising the precision and reliability of the extracted multiscale features Chen et al. (2024a). In high-sensitivity domains such as industrial fault diagnosis Ahmed et al. (2022), such distortions and the introduction of incorrect frequency can hinder diagnostic capabilities for domain experts.

Motivated by these observations, we posit that directly addressing the aliasing problem inherent in downsampling processes, particularly within convolutional architectures, represents a key breakthrough for constructing reliable multiscale time series models capable of effectively modeling time-varying entanglement. Technically, we introduce a novel multiscale convolutional downsampling framework centered around a Decomposition-Prevention-Fusion architecture, designed to mitigate aliasing during the downsampling process. Our contributions can be summarized as follows:

- We reexamine the multiscale downsampling framework for time series from a synergistic time-frequency perspective, proposing a Decomposition-Prevention-Fusion architecture that effectively disentangles time-series features to address the challenges posed by complex time-varying entanglement.
- We introduce novel mechanisms for pre-emptive prevention and post-hoc suppression of aliasing explicitly within the multiscale decomposition process, thereby further leveraging the potential of convolutional downsampling for time-series analysis.
- Through extensive experiments, we demonstrate that our proposed method achieves state-of-the-art performance with a parameter-efficient design across multiple benchmarks.

## 2  RELATED WORK

**Frequency-aware Models.**   In time series analysis, the frequency domain can effectively capture periodic information that is difficult to represent in the time domain, thus becoming an important

complement to time domain modeling. Some methods aim to enhance time domain operations by incorporating frequency domain features as auxiliary information. For example, FEDformer performs attention weight aggregation in the frequency domain Zhou et al. (2022b). Film separates the signal from the noise in historical information through Fourier filtering Zhou et al. (2022a). Meanwhile, approaches are proposed which replace time domain input with frequency domain representations directly. FITS Xu et al. (2024) and FreTS Yi et al. (2024b) use frequency-domain MLPs for prediction, significantly reducing computational complexity. FreDF Wang et al. (2025) introduces an additional loss function in the frequency domain to supervise the alignment of the model's spectrum with the real values. However, the effectiveness of frequency domain methods is constrained by the spectrum utilization bottleneck. FilterNet Yi et al. (2024a) through simple filters demonstrates that traditional feature selection strategies in the frequency domain, such as top-$K$ or random-$K$, may lead to the loss of key frequency band information. Although Fredformer Piao et al. (2024b) and proposes a frequency band equal learning mechanism and CFPT Kou et al. (2025) introduced a dual-branch architecture featuring a cross-frequency interaction module, it still does not address the issue of modeling dynamic interactions between frequency bands.

**Decomposition-based Models.**   Real-world time series are often composed of various underlying patterns. To take advantage of the features of different patterns, recent methods tend to decompose the sequence into multiple subcomponents, including trend-seasonal decomposition, multiperiod decomposition, and multiscale decomposition Huang et al. (2025). Methods such as Autoformer Wu et al. (2021) and DLinear Zeng et al. (2023) use moving averages to decouple seasonal and trend components, followed by modeling with attention mechanisms or MLP layers. TimesNet Wu et al. (2023) and PDF Dai et al. (2024) utilize Fourier analysis to decouple the sequence into multiple subperiodic sequences based on computational periods. FRENet Zhang et al. (2024) introduces a frequency-based rotation network that can capture the features of dynamically complex periods. Furthermore, TimeMixer Wang et al. (2024a) uses past decomposable mixes for multiscale representation learning and future multi-prediction mixes to enhance forecasting with complementary skills. TimeStacker Liu et al. (2025) progressively stacks features from patches of varying sizes and employs a frequency-based self-attention mechanism. However, the information fidelity of multiscale decomposition is facing challenges. Downsampling operations may lead to the loss of fine-grained features due to spectral aliasing. To address the limitations, this paper proposes a multiscale decomposition framework based on frequency domain adaptive filtering, which automatically suppresses aliasing noise through frequency band masking, ensuring the integrity of multiscale feature transfer.

## 3 MODEL FRAMEWORK

### 3.1 OVERALL ARCHITECTURE

In this section, we explain the workflow of DMANet based on a single sample for clarity. The overall architecture adopts a Decomposition-Prevention-Fusion paradigm shown in Figure.2. The input sequence is initially normalized and projected into the latent space. Then, a hierarchical extractor progressively decomposes the sequence into multiscale representations through stacked depth-wise and point-wise convolutions, with gradual downsampling of temporal resolution to capture both intra-variable and inter-variable interactions, respectively. Adhering to the prevention design principle, anti-aliasing filters are performed before each downsampling step. Next, in the upsampling phase, learnable spectral filters are adopted to expand the channel, while zero-padding expands the temporal length, effectively suppressing aliasing distortions. In the fusion stage, multiscale features are integrated using Softmax, while residual connections are made between stacked encoder blocks. Finally, the hierarchical representations are decoded. This architecture fully leverages the potential of convolutional downsampling through joint time-frequency operations.

### 3.2 NORMALIZATION AND EMBEDDING

First, we apply RevIN to the input data $X \in \mathbb{R}^{C \times L}$ to reduce the discrepancy between the training and testing data distributions Kim et al. (2022). Following this, an initial linear embedding layer re-encodes the normalized series into a latent space that is more suitable for pattern extraction and anti-aliasing. During this encoding process, we add a learnable positional encoding to preserve crucial temporal context by providing an absolute positional reference. The resulting embedded

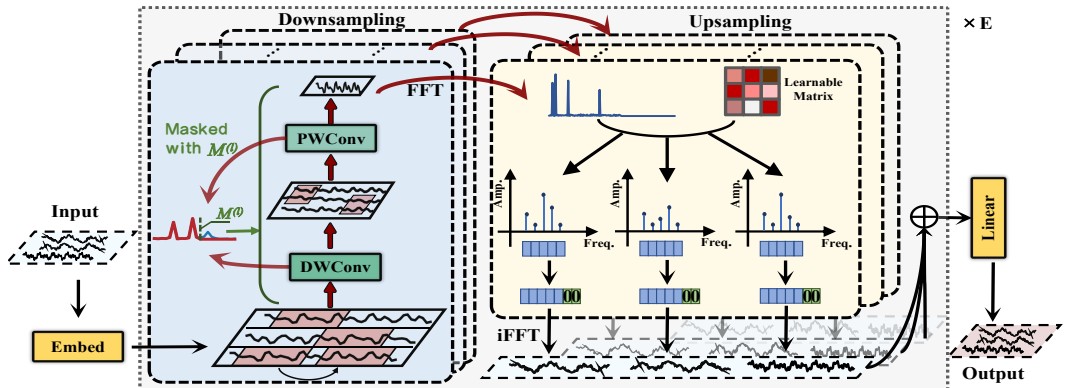

Figure 2: The overall architecture of DMANet. The input time series is first projected into a latent space via an Embedding Layer. The framework then employs a Downsampling Block to extract multiscale features using depth-wise and point-wise convolutions. Crucially, before each downsampling step, a **Pre-Sampling Anti-Aliasing operation** is performed: features are transformed to the frequency domain using FFT, filtered with a dynamic low-pass mask, and transformed back via iFFT to mitigate aliasing. Subsequently, in the **Post-Sampling Interpolation** phase, multiscale features are restored to the original resolution and fused in the frequency domain using a learnable matrix and zero-padding. Finally, a linear layer decodes the features to produce the forecast.

representation is denoted as $X' = \text{Linear}(\text{RevIN}(X)) + W$, $X' \in \mathbb{R}^{C \times T}$, where $T$ represents the dimension of the embedded representation and $W$ represents the positional encoding. This $X'$ serves as the input to the subsequent multiscale extracting layers. The detailed rationale for this embedding-first approach is provided in the Appendix.G.1.

## 3.3 MULTISCALE CONVOLUTIONAL DOWNSAMPLING

To capture features at varying temporal resolutions, we process the embedded $X'$ through a hierarchy of $H$ downsampling layers. Unlike methods relying solely on pooling Wang et al. (2024a), we employ convolutions for efficient multiscale feature extraction. This process generates a set of downsampled feature maps $X_{\text{down}} = \{x^{(0)}, x^{(1)}, x^{(2)}, \ldots, x^{(H)}\}$, where $x^{(0)}$ is the initial input $X'$ and $x^{(l)} \in \mathbb{R}^{C_l \times T_l}$. The temporal dimension decreases at each layer: $T_0 = T$ and $T_l = \lfloor T_{l-1}/s \rfloor$ for $l \geq 1$, with $s$ being the fixed downsampling stride. The number of channels $C_l$ can also vary across layers.

**Depth-wise Convolution.** In the $l$-th layer, the input $x^{(l-1)} \in \mathbb{R}^{C_{l-1} \times T_{l-1}}$ first undergoes a depth-wise convolution (DWConv). This operation applies distinct filters to each input channel, focusing on modeling temporal dependencies within channels without cross-channel interference:

$$f^{(l)} = \text{DWConv}(x^{(l-1)}; \text{ stride} = s, \text{ groups} = C_{l-1}) \in \mathbb{R}^{C_{l-1} \times T_l}. \tag{1}$$

**Point-wise Convolution.** Following the depth-wise convolution, a point-wise convolution (PWConv, that is, a $1 \times 1$ convolution) performs a linear transformation across channels. This enhances inter-channel communication and maps the feature from $C_{l-1}$ channels to $C_l$ channels:

$$x^{(l)} = \text{PWConv}(f^{(l)}) \in \mathbb{R}^{C_l \times T_l}. \tag{2}$$

When iterating this process up to the $H$-th layer, we can obtain the produced multiscale feature set $X_{\text{down}} = \{x^{(0)}, x^{(1)}, x^{(2)}, \ldots, x^{(H)}\}$. This design efficiently separates the learning of temporal patterns and channel interactions while significantly reducing parameters and computation, providing rich hierarchical information for subsequent interpolation and fusion.

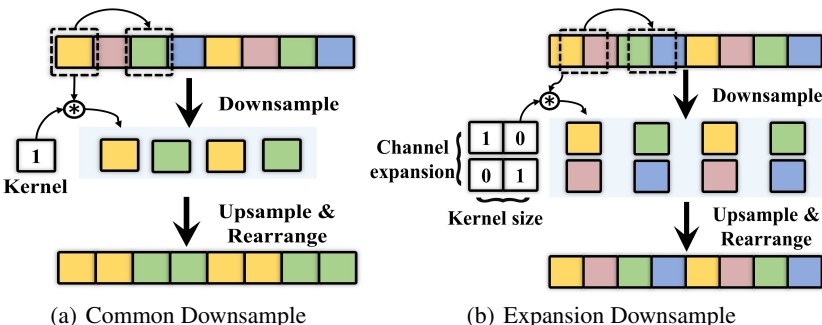

(a) Common Downsample          (b) Expansion Downsample

Figure 3: Illustration of how a larger kernel and channel expansion preserve sampling information during downsampling. **Left:** Pointwise downsampling with a stride of 2 and no channel expansion results in a sampling rate of 1/2, discarding half the input information. This leads to aliasing, where high-frequency content is misrepresented as low frequencies. **Right:** Downsampling with 2×1 identity kernels and 2× channel expansion ensures that every input element is sampled and preserved in separate channels. This approach maintains the effective sampling rate at 1,as all pixels are sampled.

### 3.4 ANTI-ALIASING OPERATION

In Section.3.3, the depth-wise convolution and point-wise convolution are proposed. To reduce the negative effect of aliasing, a Pre-Sampling Filtering and Post-Sampling Interpolation should be performed before feeding $x^{(l-1)}$ and after acquiring $x^{(l)}$, respectively.

**Pre-Sampling Filtering.** Downsampling inevitably introduces the risk of aliasing, where high-frequency components fold into lower frequency bands, potentially corrupting the signal or losing critical information, especially with larger strides $s$. Inspired by the work in computer visionGrabinski et al. (2022a)Chen et al. (2024a), we introduce the concept of Equivalent Sampling Rate (ESR) to dynamically compute the appropriate Nyquist frequency for anti-aliasing filtering during downsampling. We provide a detailed proof for ESR in Appendix.B. As shown in Figure.3, the size of the convolutional kernel and the transformation of channels play a role in determining the sampling ability. Concretely, the ESR at the $l$-th layer can be calculated with the following strategy:

$$\text{ESR}^{(l)} = \frac{\min\left(K, C_l/C_{l-1}\right)}{s}, \tag{3}$$

where $K$ is the kernel size of the depth-wise convolution, $C_{l-1}$ and $C_l$ are the input and output channels for the layer's point-wise convolution, and $s$ is the stride. Before applying the down-sampling convolution, we use FFT ($\mathcal{F}$) to transform the input $x^{(l-1)}$ into the frequency domain: $\mathcal{X}^{(l-1)} = \mathcal{F}(x^{(l-1)})$. The layer-specific Nyquist frequency is $f_{\text{Nyquist}}^{(l)} = \text{ESR}^{(l)}/2$. Based on $f_{\text{Nyquist}}^{(l)}$, we construct a low-pass frequency mask $\mathbf{M}^{(l)}$:

$$\mathbf{M}^{(l)}[i] = \begin{cases} 1, & f_i \leq f_{\text{Nyquist}}^{(l)} \\ 0, & f_i > f_{\text{Nyquist}}^{(l)} \end{cases}, \tag{4}$$

where $f_i$ is the $i$-th frequency compotent in $\mathcal{X}^{(l-1)}$. Then, we apply the mask in an element-wise strategy ($\odot$), and transform back using IFFT ($\mathcal{F}^{-1}$):

$$\tilde{\mathcal{X}}^{(l-1)} = \mathcal{X}^{(l-1)} \odot \mathbf{M}^{(l)}, \ x_{\text{filtered}}^{(l-1)} = \mathcal{F}^{-1}(\tilde{\mathcal{X}}^{(l-1)}). \tag{5}$$

This filtered signal $x_{\text{filtered}}^{(l-1)}$, which serves as the real $x^{(l-1)}$, is then fed into the depth-wise convolution, effectively suppressing high frequencies prone to aliasing during downsampling.

**Post-Sampling Interpolation.** After obtaining the downsampled feature $x^{(l)} \in \mathbb{R}^{C_l \times T_l}$, we propose an anti-aliasing interpolation method and frequency domain channel expansion to restore the temporal resolution to a target length $T$ and expand channels to the model dimension $C$.

First, the downsampled feature is transformed to the frequency domain using $\mathcal{F}$. Inspired by frequency domain filtering strategies Yi et al. (2024a), we introduce our designed learnable complex-valued filters $\mathcal{H}_\phi^{(l)} \in \mathbb{C}^{C \times C_l \times F_l}$, which are also transformed to the frequency domain. We then perform channel expansion through a weighted sum in the frequency domain, effectively implementing channel mixing. This operation computes each output channel $c$ as a learned combination of all $C_l$ input channels at each frequency $f$, fusing cross-channel information while preserving the spectral structure:

$$\mathcal{X}^{(l)} = \mathcal{F}(x^{(l)}) \in \mathbb{C}^{C_l \times F_l}, \ F_l = \lfloor T_l/2 \rfloor + 1, \tag{6}$$

$$\mathcal{S}^{(l)}[c, f] = \sum_{k=1}^{C_l} \mathcal{X}^{(l)}[k, f] \odot \mathcal{H}_\phi^{(l)}[c, k, f], \ \text{for } c \in [1, C] \text{ and } f \in [1, F_l], \tag{7}$$

where $\mathcal{S}^{(l)} \in \mathbb{C}^{C \times F_l}$. To restore the sequence length to $T$, we calculate the corresponding target frequency count $F = \lfloor T/2 \rfloor + 1$. We apply zero-padding (Pad) in the frequency domain to extend the spectrum $\mathcal{S}^{(l)}$ from $F_l$ components to $F$ components. This zero-padding primarily serves to interpolate the signal in the time domain while implicitly acting as a low-pass filter, further mitigating potential aliasing introduced during the process. Finally, $\mathcal{F}^{-1}$ transforms the padded spectrum back to the time domain, producing the interpolated feature map for level $l$:

$$\tilde{\mathcal{S}}^{(l)} = \text{Pad}^{(l)}(\mathcal{S}^{(l)}) \in \mathbb{C}^{C \times F}, \ Y^{(l)} = \mathcal{F}^{-1}(\tilde{\mathcal{S}}^{(l)}, n = T) \in \mathbb{R}^{C \times T}. \tag{8}$$

Executing channel expansion before zero-padding ensures that the spectral information for each expanded channel is complete before interpolation, avoiding potential spectral leakage or distortion and contributing to high-fidelity reconstruction in multi-channel time series tasks.

## 3.5 MULTISCALE FEATURE FUSION AND DECODING OUTPUT

**Feature Fusion.** The above operations are processed in a single encoder block, and the interpolation step generates a set of feature maps $\{Y^{(1)}, Y^{(2)}, \ldots, Y^{(H)}\}$, each residing at the target resolution $T$, but derived from different temporal scales. To integrate these multiscale information, we employ adaptive weighting. A learnable weight vector $w \in \mathbb{R}^H$ is introduced, and its Softmax normalization yields attention scores $\alpha_p$ for each scale. The final output $\hat{Y}^e$ of the $e$-th encoder block is the weighted sum of these multiscale features:

$$\alpha_p = \frac{\exp(w_p)}{\sum_{k=1}^H \exp(w_k)}, \hat{Y}^e = \sum_{p=1}^H \alpha_p \cdot Y^{(p)} \in \mathbb{R}^{C \times T}. \tag{9}$$

Our model stacks $E$ such multiscale encoder blocks. To facilitate the training of this deep architecture and preserve information flow, we incorporate residual connections around each encoder block, where $\hat{Y}^{(0)} = X'$ means the initial embedded representation, and $\hat{Y}^e$ is the output of the $e$-th encoder layer:

$$\hat{Y}^e = \text{MultiScaleEncoder}(\hat{Y}^{e-1}) + \hat{Y}^{e-1}, \ \text{for } e = 1, 2, 3, \ldots, E. \tag{10}$$

**Decoding Output.** The output $\hat{Y}^E$ from the final encoder block, representing rich multiscale features, first passes through a Layer Normalization step, $Y^* = \text{LayerNorm}(\hat{Y}^E)$. Then a simple Feed-Forward Network (FFN) decoder projects these features into the future prediction horizon $L_{\text{next}}$. Then we apply the inverse RevIN transformation (iRevIN) to obtain the final forecast $\hat{X}_o$:

$$\hat{X}_o = \text{iRevIN}(\text{FFN}(Y^*)) \in \mathbb{R}^{C \times L_{\text{next}}}. \tag{11}$$

## 4 EXPERIMENTS

### 4.1 EXPERIMENTAL SETTINGS

**Datasets.** We conduct experiments on many real-world public datasets for long-term forecasting: ETT (four subsets), Weather, ECL, Solar-Energy, PEMS (four subsets). For short-term forecasting, we adopt the ILI, COVID-19, NASDAQ, Wiki, SP500, DowJones, CarSales, Power, Website, Unemp. These datasets are standard benchmarks Wang et al. (2024a) Liu et al. (2024) Yue et al. (2025). Details and statistics of these multivariate time series datasets are summarized in Appendix.C.

Table 1: Long-term forecasting results ($L = 96$). All results are averaged across four forecasting horizon: $T \in \{96, 192, 336, 720\}$. The best and second-best results are highlighted in **bold** and underlined, respectively. See Appendix.D (Table.11 and Table.12) for full results.

| Models | DMANet Ours | | iTransformer 2024 | | TimeMixer 2024a | | FilterNet 2024a | | Fredformer 2024a | | FITS 2024 | | FreTS 2024b | | TimePro 2025 | | FreDF 2025 | | SOFTS 2024 | | TimeXer 2024b | |
|---|---|---|---|---|---|---|---|---|---|---|---|---|---|---|---|---|---|---|---|---|---|---|
| Metric | MSE | MAE | MSE | MAE | MSE | MAE | MSE | MAE | MSE | MAE | MSE | MAE | MSE | MAE | MSE | MAE | MSE | MAE | MSE | MAE | MSE | MAE |
| ETTh1 | **0.428** | **0.429** | 0.454 | 0.447 | 0.447 | 0.440 | 0.440 | 0.432 | 0.432 | 0.447 | 0.448 | 0.488 | 0.474 | 0.438 | 0.438 | 0.437 | 0.435 | 0.449 | 0.442 | 0.437 | 0.437 |
| ETTh2 | **0.361** | **0.388** | 0.383 | 0.407 | 0.364 | 0.395 | 0.378 | 0.397 | 0.367 | 0.396 | 0.383 | 0.408 | 0.550 | 0.515 | 0.377 | 0.403 | 0.371 | 0.396 | 0.373 | 0.400 | 0.368 | 0.396 |
| ETTm1 | **0.373** | **0.385** | 0.407 | 0.410 | 0.381 | 0.395 | 0.384 | 0.398 | 0.393 | 0.403 | 0.387 | 0.408 | 0.407 | 0.415 | 0.391 | 0.400 | 0.392 | 0.399 | 0.393 | 0.403 | 0.382 | 0.397 |
| ETTm2 | **0.268** | **0.310** | 0.288 | 0.332 | 0.275 | 0.323 | 0.276 | 0.322 | 0.279 | 0.324 | 0.286 | 0.328 | 0.335 | 0.379 | 0.281 | 0.326 | 0.278 | 0.319 | 0.287 | 0.330 | 0.274 | 0.322 |
| Weather | **0.236** | **0.262** | 0.258 | 0.279 | 0.240 | 0.271 | 0.248 | 0.278 | 0.246 | 0.272 | 0.249 | 0.276 | 0.255 | 0.363 | 0.251 | 0.276 | 0.254 | 0.274 | 0.255 | 0.278 | 0.241 | 0.271 |
| Electricity | 0.170 | 0.264 | 0.178 | 0.270 | 0.182 | 0.272 | 0.201 | 0.285 | 0.175 | 0.269 | 0.217 | 0.295 | 0.202 | 0.290 | **0.169** | 0.262 | 0.170 | **0.259** | 0.174 | 0.264 | 0.171 | 0.270 |
| Solar-Energy | 0.227 | **0.249** | 0.233 | 0.262 | **0.216** | 0.280 | 0.263 | 0.286 | 0.232 | 0.274 | 0.397 | 0.398 | 0.283 | 0.338 | 0.232 | 0.266 | 0.279 | 0.292 | 0.229 | 0.256 | 0.237 | 0.302 |

**Baseline.** Our primary analysis focuses on long-term forecasting with a 96-step lookback window (Table 11, Table 12 and Table 15). In addition to this main task, we also conducted evaluations on univariate long-term forecasting (Table 16), short-term forecasting (Table 17), and long-term forecasting with an extended 720-step lookback window (Tables 13 and 14). Across these diverse settings, we chose a comprehensive set of recent state-of-the-art models to serve as baselines. This includes MLP-based models (SOFTS Han et al. (2024), TimeMixer Wang et al. (2024a), DLinear Zeng et al. (2023)), CNN-based models (TVNet Li et al. (2025), ModernTCN Donghao & Xue (2024), PDF Dai et al. (2024)), frequency-based models (FreDF Wang et al. (2025), FilterNet Yi et al. (2024a) etc.), Transformer-based models (TimeXer Wang et al. (2024b), iTransformer Liu et al. (2024), etc.), and recent architectures based on Mamba (TimePro Ma et al. (2025)) and KAN (TimeKAN Huang et al. (2025)). To ensure a clear and focused presentation in the main text, our primary results tables (Table.1, Table.2 and Table.3) feature a curated selection of the most competitive and representative SOTA models. A complete list of all evaluated baselines is described in Appendix.C, with their comprehensive results available in Appendix.D for a thorough comparison.

**Implementation Details.** The experiments in this paper were conducted using an NVIDIA GeForce RTX 3090 24GB GPU. Inspired by FreDF Wang et al. (2025), we uses the Mean Absolute Error (MAE) in the frequency domain. For details on the hyperparameter settings of the models presented in Appendix.C.

## 4.2 MAIN RESULTS

**Long-term Forecasting.** The long-term forecasting results, reported in Table.1 (more results in Appendix.D), demonstrate that DMANet consistently achieves optimal or near-optimal performance across all datasets. Its performance is comparable to TimeMixer, highlighting the general effectiveness of time-series decomposition architectures. However, a key distinction lies in their downsampling mechanisms: while TimeMixer's reliance on average pooling is susceptible to information loss, DMANet's spectral preservation mechanism effectively suppresses aliasing artifacts, enabling a more faithful layer-wise learning of multi-granularity representations. Conversely, when compared to models with channel-wise self-attention like iTransformer, DMANet's reliance on simpler convolutional operations for dependency modeling suggests a potential area for future optimization, particularly on high-dimensional datasets. Furthermore, guided by the principles of scaling laws in Time Series Forecasting (TSF), we extended the lookback window $L$ to 720 in Table.2 (full results can be found in Table.13 and Table.14). In this long-context setting, DMANet exhibits robust noise resilience, maintaining state-of-the-art performance and surpassing other convolutional counterparts like ModernTCN and TVNet. This result further validates DMANet's superior adaptability in capturing multi-scale temporal dependencies, even with extended input lengths.

Table 2: Long-term forecasting results ($L = 720$). For baseline, the input length $L$ is searched from $\{192, 336, 512, 720\}$, while DMANet is fixed 720. The best and second-best results are highlighted in **bold** and underlined, respectively. See Appendix.D (Table.13 and Table.14) for full results.

| Models | DMANet | | iTransformer | | TimeMixer | | ModernTCN | | TVNet | | TSLANet | | PDF | | PatchTST | | FITS | | TimesNet | | DLinear | |
|---|---|---|---|---|---|---|---|---|---|---|---|---|---|---|---|---|---|---|---|---|---|---|
| Metric | MSE | MAE | MSE | MAE | MSE | MAE | MSE | MAE | MSE | MAE | MSE | MAE | MSE | MAE | MSE | MAE | MSE | MAE | MSE | MAE | MSE | MAE |
| ETTm1 | **0.338** | **0.369** | 0.361 | 0.390 | 0.356 | 0.380 | 0.351 | 0.381 | 0.348 | 0.379 | 0.348 | 0.383 | 0.342 | 0.376 | 0.349 | 0.381 | 0.357 | 0.377 | 0.408 | 0.415 | 0.356 | 0.378 |
| ETTm2 | **0.248** | **0.307** | 0.269 | 0.327 | 0.257 | 0.318 | 0.253 | 0.314 | 0.256 | 0.316 | 0.256 | 0.316 | 0.250 | 0.313 | 0.256 | 0.314 | 0.254 | 0.313 | 0.292 | 0.331 | 0.259 | 0.324 |
| Weather | **0.218** | **0.252** | 0.232 | 0.270 | 0.226 | 0.264 | 0.224 | 0.264 | 0.221 | 0.261 | 0.325 | 0.337 | 0.227 | 0.263 | 0.224 | 0.261 | 0.244 | 0.280 | 0.255 | 0.282 | 0.242 | 0.293 |
| Electricity | **0.154** | **0.252** | 0.163 | 0.258 | 0.169 | 0.265 | 0.156 | 0.253 | 0.165 | 0.254 | 0.165 | 0.257 | 0.160 | 0.253 | 0.171 | 0.270 | 0.169 | 0.265 | 0.190 | 0.290 | 0.167 | 0.264 |

**Short-term Forecasting.** The short-term forecasting results, presented in Table.3, validate the superiority of DMANet in handling non-stationary time series. Across a diverse set of challenging datasets such as ILI, COVID-19, and DowJones, DMANet consistently achieves the best performance. It significantly outperforms other methods, including strong frequency-domain baselines like Fredformer and FilterNet. These results underscore DMANet's exceptional capability in short-term and non-stationary forecasting, attributable to its synergistic design: the convolutional architecture excels at preserving local features, while the anti-aliasing structure effectively mitigates disruptive high-frequency noise.

Table 3: Short-term forecasting results. The best and second-best results are highlighted in **bold** and underlined, respectively. See Appendix.D (Table.17) for full results and setting details.

| Models | DMANet | | TimeMixer | | FilterNet | | FITS | | DLinear | | Fredformer | | PatchTST | |
|---|---|---|---|---|---|---|---|---|---|---|---|---|---|---|
| Metric | MSE | MAE | MSE | MAE | MSE | MAE | MSE | MAE | MSE | MAE | MSE | MAE | MSE | MAE |
| ILI | **1.763** | **0.824** | 2.020 | 0.878 | 2.073 | 0.885 | 4.130 | 1.465 | 3.083 | 1.217 | 1.947 | 0.899 | 2.128 | 0.885 |
| COVID-19 | 1.910 | **0.670** | 2.234 | 0.782 | 2.088 | 0.780 | 2.875 | 0.979 | 3.483 | 1.102 | **1.902** | 0.765 | 2.221 | 0.820 |
| NASDAQ | **0.177** | **0.273** | 0.186 | 0.281 | 0.197 | 0.289 | 0.210 | 0.302 | 0.228 | 0.331 | 0.194 | 0.285 | 0.198 | 0.286 |
| Wiki | 6.506 | **0.393** | 6.572 | 0.409 | 6.572 | 0.411 | 8.515 | 0.553 | 6.634 | 0.481 | 6.705 | 0.406 | 6.523 | 0.404 |
| SP500 | **0.225** | **0.329** | 0.241 | 0.353 | 0.254 | 0.365 | 0.291 | 0.412 | 0.277 | 0.391 | 0.261 | 0.378 | 0.246 | 0.361 |
| DowJones | **11.957** | **0.850** | 13.948 | 0.877 | 13.439 | 0.873 | 13.755 | 0.893 | 12.688 | 0.857 | 12.992 | 0.858 | 12.916 | 0.862 |
| CarSales | 0.338 | **0.333** | 0.338 | 0.336 | **0.336** | 0.335 | 0.379 | 0.365 | 0.373 | 0.368 | 0.340 | 0.338 | 0.338 | 0.335 |
| Power | **1.373** | **0.899** | 1.484 | 0.937 | 1.614 | 0.986 | 1.711 | 1.028 | 1.549 | 0.972 | 1.588 | 0.981 | 1.650 | 0.998 |
| Website | 0.137 | **0.252** | 0.143 | 0.261 | 0.136 | 0.255 | 0.278 | 0.383 | 0.204 | 0.319 | **0.135** | 0.254 | 0.141 | 0.259 |
| Unemp | **0.064** | **0.146** | 0.094 | 0.183 | 0.079 | 0.166 | 0.308 | 0.394 | 0.154 | 0.292 | 0.075 | 0.163 | 0.078 | 0.160 |

## 4.3 ABLATION STUDY

In this section, we investigate key components of DMANet, including our novel Anti-aliasing Filter, the Convolutional Downsampling and Frequency Upsampling Mechanisms, and the Basic Settings.

Table 4: Ablation study of DMANet. All results are averaged across four different forecasting horizon. The best and second-best results are highlighted in **bold** and underlined, respectively.

| Catagories | | Downsampling Replace | | | | | | Upsampling Replace | | | | | | Basic Settings | | | |
|---|---|---|---|---|---|---|---|---|---|---|---|---|---|---|---|---|---|---|
| Cases | DMANet | | Linear Down | | Self-Attention | | Standard Conv | | Linear Up | | Interpolate | | Trans Conv | | w/o ReVIN | | MSE Loss | |
| | MSE | MAE | MSE | MAE | MSE | MAE | MSE | MAE | MSE | MAE | MSE | MAE | MSE | MAE | MSE | MAE | MSE | MAE |
| Electricity | **0.172** | **0.265** | 0.176 | 0.271 | 0.180 | 0.276 | 0.179 | 0.274 | 0.174 | 0.268 | 0.184 | 0.274 | 0.179 | 0.271 | 0.215 | 0.316 | 0.173 | 0.268 |
| ETTm1 | **0.373** | **0.385** | 0.379 | 0.389 | 0.379 | 0.389 | 0.377 | 0.388 | 0.377 | 0.388 | 0.377 | 0.388 | 0.378 | 0.388 | 0.423 | 0.445 | 0.381 | 0.393 |
| Unemp | **0.064** | **0.146** | 0.081 | 0.171 | 0.076 | 0.161 | 0.075 | 0.163 | 0.077 | 0.164 | 0.073 | 0.161 | 0.068 | 0.155 | 0.759 | 0.414 | 0.076 | 0.166 |
| NASDAQ | **0.177** | **0.273** | 0.190 | 0.283 | 0.184 | 0.279 | 0.185 | 0.281 | 0.183 | 0.279 | 0.182 | 0.277 | 0.184 | 0.278 | 2.337 | 1.132 | 0.195 | 0.288 |

**Basic Settings.** Ablation analysis showed that removing DMANet's ReVIN significantly hurts performance by failing to mitigate distribution shift. The frequency-domain MAE loss is also preferred over MSE for anti-aliasing due to enabling direct frequency adjustment.

**Convolution Downsampling.** To evaluate the effectiveness of convolutional downsampling, we experimented with the following alternative strategies: **(1) LinearDown:** two separate linear for downsampling; **(2) Standard Conv:** standard convolution with stride; **(3) Self-attention:** employing self-attention to capture temporal dependencies, combined with average pooling along the temporal and convolutional downsampling along the channel. The results are summarized in Table.4.

Overall, the combination of depth-wise convolution and point-wise convolution demonstrates the best performance. Notably, replacing convolution with linear or with self-attention followed by average pooling results in a performance drop, which highlights the capability of depthwise convolution in learning temporal dependencies. In addition, using standard convolution alone leads to a substantial increase in parameter count and a worsening of most metrics, suggesting that focusing solely on depthwise convolution to extract temporal dependencies is a more reasonable design.

**Frequency Upsampling.** To validate the unique advantages of frequency-domain upsampling, we conducted experiments comparing our approach with three alternative upsampling methods that do not explicitly target frequency information: **(1) Linear Up:** two separate linear for upsampling;

Table 5: Results on generic baseline. More details are in Appendix.C.6

| Models | DMANet | | Base | | w/o-Pre | | w/o-Post | |
|---|---|---|---|---|---|---|---|---|
| Metric | MSE | MAE | MSE | MAE | MSE | MAE | MSE | MAE |
| Unemp | **0.064** | **0.146** | 0.082 | 0.173 | 0.075 | 0.161 | 0.069 | 0.155 |
| DowJones | **11.957** | **0.850** | 13.294 | 0.871 | 12.379 | 0.853 | 12.423 | 0.855 |
| ETTm1 | **0.373** | **0.385** | 0.384 | 0.393 | 0.378 | 0.387 | 0.377 | 0.388 |
| Weather | **0.236** | **0.262** | 0.239 | 0.265 | 0.239 | 0.264 | 0.238 | 0.264 |

Table 6: Results on plug-in design. [†] denotes integration with ESR filter. ↓ indicates reduction vs. vanilla models.

| Models | TimeMixer[†] | | MICN[†] | | SCINet[†] | |
|---|---|---|---|---|---|---|
| Metric | MSE | MAE | MSE | MAE | MSE | MAE |
| Power | 1.444 (↓ 0.004) | 0.922 (↓ 0.015) | 1.976 (↓ 0.005) | 0.973 (↓ 0.013) | 0.182 (↓ 0.023) | 0.278 (↓ 0.003) |
| NASDAQ | 0.182 (↓ 0.004) | 0.278 (↓ 0.003) | 0.198 (↓ 0.005) | 0.295 (↓ 0.002) | 0.234 (↓ 0.005) | 0.335 (↓ 0.004) |
| ETTh1 | 0.448 (↓ 0.010) | 0.439 (↓ 0.005) | 0.511 (↓ 0.069) | 0.503 (↓ 0.031) | 0.506 (↓ 0.027) | 0.478 (↓ 0.019) |
| Electrity | 0.181 (↓ 0.003) | 0.271 (↓ 0.002) | 0.193 (↓ 0.003) | 0.304 (↓ 0.005) | 0.217 (↓ 0.002) | 0.319 (↓ 0.002) |

Table 7: Ablation study on different filter designs. Performance is compared against our DMANet (utilizing the ESR filter), heuristic filters (Max, Random), and classical filters (Ideal, Chebyshev, Gaussian, Butterworth). All results are averaged over four horizons. Details are in Appendix.C.5.

| Models | DMANet | | Max | | Random | | Ideal | | Chebyshev | | Gaussian | | Butterworth | |
|---|---|---|---|---|---|---|---|---|---|---|---|---|---|---|
| Metric | MSE | MAE | MSE | MAE | MSE | MAE | MSE | MAE | MSE | MAE | MSE | MAE | MSE | MAE |
| ILI | **1.763** | **0.824** | 1.957 | 0.855 | 2.043 | 0.872 | 1.994 | 0.849 | 1.990 | 0.855 | 1.974 | 0.862 | 1.940 | 0.849 |
| Unemp | **0.064** | **0.146** | 0.073 | 0.157 | 0.074 | 0.159 | 0.073 | 0.159 | 0.075 | 0.164 | 0.071 | 0.154 | 0.072 | 0.158 |
| DowJones | **11.957** | **0.850** | 12.300 | 0.852 | 12.261 | 0.851 | 12.397 | 0.855 | 12.382 | 0.855 | 12.351 | 0.854 | 12.402 | 0.855 |
| ETTm1 | **0.373** | **0.385** | 0.376 | 0.387 | 0.376 | 0.387 | 0.375 | 0.387 | 0.375 | 0.387 | 0.374 | **0.385** | 0.375 | 0.387 |
| PEMS08 | **0.090** | **0.198** | 0.117 | 0.218 | 0.113 | 0.210 | 0.109 | 0.206 | 0.110 | 0.207 | 0.108 | 0.205 | 0.109 | 0.206 |

**(2) Interpolate:** simply interpolation along the temporal and channel; **(3) Trans Conv:** utilizes transposed convolution mirroring the structure of the downsampling counterpart. As shown clearly in Table.4, replacing our frequency-domain upsampling with any of these alternatives resulted in a significant performance degradation. This indicates that these methods fail to effectively preserve or reconstruct the crucial frequency components of time series during the upsampling process.

In contrast, our strategy first performs expansion in the channel dimension, followed by high-frequency truncation in the frequency domain. This carefully designed approach ensures the structural integrity and independence of each channel in the frequency domain and completely avoids the spectral leakage and the aliasing problem inherent to interpolation-based methods. As a result,

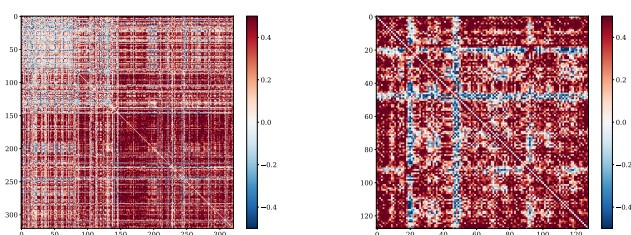

Figure 4: Visualization of dependency differences, comparing feature representations with and without the anti-aliasing filter. Red indicates an increase, while blue indicates a decrease. **(a) Left:** Channel-wise dependency. **(b) Right:** Temporal dependency differences.

our method demonstrates outstanding performance in the high-fidelity reconstruction of time series.

**Anti-aliasing Design Analysis.** We conducted a multi-dimensional analysis to validate the effectiveness and universality of our anti-aliasing mechanisms. First, regarding generalizability, we integrated our ESR-based filter as a plug-in module into other downsampling-based models (e.g., TimeMixer, MICN) in Table.6. The consistent performance gains suggest the potential of our anti-aliasing approach to serve as a generic enhancement for existing methods. Second, regarding component contribution in Table.5, we constructed a generic baseline to rigorously disentangle the effects of Pre-Sampling Filtering and Post-Sampling Interpolation. The results demonstrate that both components provide distinct and synergistic benefits: the pre-sampling filter effectively prevents high-frequency corruption, while the frequency-domain interpolation ensures high-fidelity reconstruction (see Appendix.H for detailed analysis). Finally, regarding filter superiority in Table.7, we benchmarked DMANet against heuristic (Max, Random) and classical filters (Ideal, Chebyshev, etc.). DMANet outperforms all competing filters, validating that our dynamic, architecture-aware cutoff strategy offers superior adaptability compared to static or manual designs.

### 4.4 MODEL ANALYSIS

**Efficiency and Robustness.** We provide comprehensive results on real-world datasets (Appendix.E, Table.23) , including the Params and MACs. DMANet demonstrates a balance between

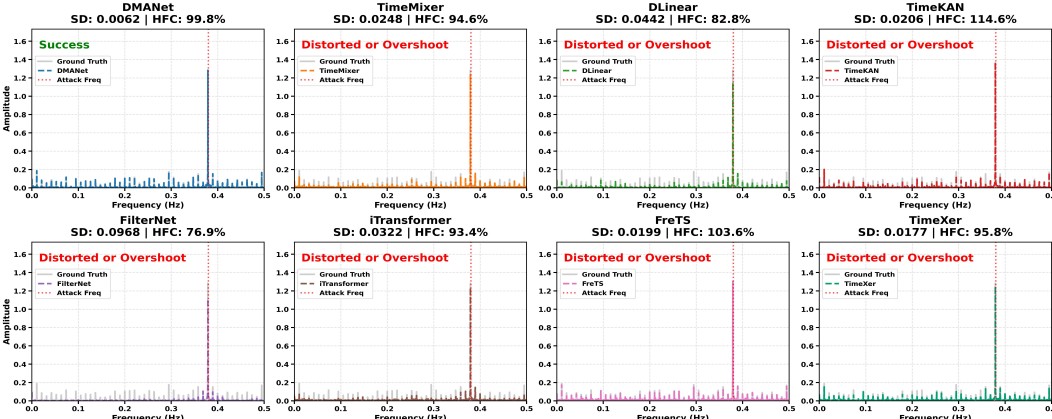

Figure 5: Analysis of Aliasing Risks via Spectral Injection Attack. **Spectral Distortion (SD)** measures the Euclidean distance between the predicted and actual spectral distributions, while **High-Frequency Capture (HFC)** quantifies the model's ability to preserve the injected signal energy.

performance and efficiency. Compared to others, it requires fewer MACs and less params while achieving better accuracy. Meanwhile, on synthetic data (Appendix.E, Table.22), we performed a fine-grained analysis of computational costs. An ablation study revealed that our anti-aliasing filter is not a performance bottleneck, introducing negligible overhead (a worst-case latency increase of only 2.4%). The robustness of our model was validated through a series of noise injection experiments, detailed in Appendix.F. We introduced five types of synthetic noise (e.g., high-frequency) at various intensities $\epsilon$. Our anti-aliasing architecture demonstrated great resilience, as its performance degraded with increasing noise, thereby validating its ability to mitigate signal disturbances.

**Dependency Modeling.** Figure.4 presents feature dependency heatmaps from the ECL dataset, which reveal the effect of anti-aliasing filter. The filter smooths fine-grained dependencies that are susceptible to aliasing during downsampling. By suppressing these potentially noisy or misleading correlations, the filtering process accentuates the underlying structural patterns in both the temporal and channel. It allows the subsequent layers to more easily extract stable and meaningful features from a cleaner, more coherent representation. Detailed dependency passing analysis in Appendix.H.

**Analysis of Aliasing Risks.** To verify the aliasing risks, we conducted a Spectral Injection Attack experiment by injecting a high-frequency signal into the ETTh1 dataset. This frequency exceeds the Nyquist limit of standard downsampling, theoretically inducing aliasing. Figure 5 illustrates the spectral reconstruction results: DMANet achieves superior fidelity with the lowest Spectral Distortion (0.0062) and HFC rate of 99.8%, accurately reconstructing the signal without artifacts. In stark contrast, FilterNet (76.9% HFC) and DLinear (82.8% HFC) suffer from the signal attenuation, acting as uncontrolled low-pass filters. Meanwhile, FreTS and TimeXer, despite capturing the target frequency ($> 95\%$ HFC), exhibit high spectral distortion (SD $> 0.017$), indicating that they fail to disentangle the signal from aliasing noise. Similarly, TimeKAN exhibits spectral instability and overshoot (114.6% HFC). This experiment demonstrates the limitations of existing architectures in handling Nyquist sampling, and highlights DMANet's unique capability to maintain spectral fidelity.More details can be found in Appendix.I.

## 5 CONCLUSION

This paper presents DMANet, a novel architecture that tackles the critical aliasing problem in multiscale time series forecasting through a Decomposition-Prevention-Fusion framework, employing pre-sampling anti-aliasing based on Equivalent Sampling Rate and post-sampling interpolation for high-fidelity features. Extensive experiments on diverse benchmarks demonstrate DMANet's state-of-the-art performance and robustness, validating the significance of the anti-aliasing design. DMANet offers a promising direction by explicitly integrating signal processing principles to enhance time series analysis robustness.

# 6 ETHICS STATEMENT

Our research is primarily foundational, focusing on a technical challenge within time series analysis, i.e., the problem of aliasing in multiscale deep learning models. We have considered the ethical implications of our work and believe that it follows the scientific standards.

## 6.1 SOCIETAL IMPACT

The primary goal of DMANet is to improve the fidelity and reliability of time series forecasting models by mitigating the spectral distortion caused by aliasing. This has a positive social impact by enhancing the trustworthiness of predictive systems in high-sensitivity domains. For example, in industrial fault diagnosis, preventing the introduction of spurious frequency patterns can improve the accuracy of diagnostic tools and support expert decision-making. Similarly, more reliable models are beneficial in fields such as economics, transportation planning, and weather forecasting. We acknowledge that, like any advanced forecasting technology, our methods could potentially be misused. However, our work is a general-purpose technical improvement, not an application-specific tool, and we advocate for its responsible use in future research and applications.

## 6.2 DATA USAGE

All experiments were carried out on publicly available and well-established benchmark datasets, e.g., ETT, Weather, ECL, PEMS, ILI, COVID-19. A complete list and description of these datasets are provided in the Appendix.C. In this study, no sensitive or private user data was used, thus avoiding concerns related to privacy and data protection.

## 6.3 BIAS AND FAIRNESS

Although our work does not directly address dataset bias, it contributes to model fairness by tackling a source of technical error. By avoiding the aliasing problem, our model is less likely to learn from misleading artifacts in the data. This enhances the model's robustness and ensures its predictions are based on a more faithful representation of the underlying signal, which is a prerequisite for fair and reliable decision-making.

# 7 REPRODUCIBILITY STATEMENT

## 7.1 CODE

The complete source codes for DMANet, including model implementations and scripts to reproduce experimental results, are available in our anonymous repository at `https://anonymous.4open.science/r/DMANet-ED7A`. The repository includes instructions for setting up the environment, preparing the data, and running the training and evaluation scripts. Upon acceptance, the repository will be made public and accessible.

## 7.2 DATASETS

Our study utilizes multiple publicly available real-world datasets for long-term and short-term forecasting, including ETT, Weather, ECL, Solar-Energy, PEMS, ILI, COVID-19, and others. Detailed descriptions, statistics, sources, and data-splitting protocols (Train/Validation/Test ratios) for each dataset are provided in Appendix.C. The data processing follows the established protocols of previous benchmark studies to ensure fair comparison.

## 7.3 EXPERIMENTAL SETUP

The Section.4.1 outlines the overall setup, while Appendix.C provides more implementation details, including the hyperparameter search spaces for all tasks, the specific configurations for each dataset, detailed descriptions of the baseline models, and the fair comparison settings. All experiments were conducted on a single NVIDIA GeForce RTX 3090 24GB GPU using the PyTorch framework

with the same version. In addition, the full results shown in Appendix.D complement the summary tables in the main text. Furthermore, we rigorously validate our results through statistical tests on experiments conducted with five random seeds, confirming that DMANet's superior performance is statistically significant with 99% confidence.

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

## A   PRELIMINARIES

### A.1   PROBLEM STATEMENT

**Time Series.**   Time series $\mathbf{X} \in \mathbb{R}^{C \times N}$ refers to a sequence of data points ordered by time, where $N$ denotes the total number of timestamps and $C$ represents the number of channels at each times-tamp. Time series forecasting involves predicting future data points based on historical time series observations. The historical observations can be represented as $X = [\mathbf{x}_1, \mathbf{x}_2, \ldots, \mathbf{x}_L] \in \mathbb{R}^{C \times L}$, and $L$ is the length of the historical look-back window. The future data for the next $L_{\text{next}}$ time steps, denoted as $\hat{X}_o = [\mathbf{x}_{L+1}, \mathbf{x}_{L+2}, \ldots, \mathbf{x}_{L+L_{\text{next}}}] \in \mathbb{R}^{C \times L_{\text{next}}}$, correspond to the fore-cast horizon. Given these, time series forecasting models are required to learn mapping functions $\mathbf{F} : X \in \mathbb{R}^{C \times L} \to \hat{X}_o \in \mathbb{R}^{C \times L_{\text{next}}}$.

**Aliasing.**   This issue arises when different high-frequency components in a continuous signal are indistinguishably mapped to the same low-frequency components after sampling or improper downsampling. Formally, let the sampling interval be $\Delta t$, with the Nyquist frequency defined as $f_{\text{Nyquist}} = \frac{1}{2\Delta t}$. Any frequency component $f > f_{\text{Nyquist}}$ in the signal will alias to a spurious fre-quency $\tilde{f} = |f - k \cdot f_s|$ in the sampled sequence $X$, where $f_s = \frac{1}{\Delta t}$ is the sampling rate, and $k \in \mathbb{Z}^+$ ensures $\tilde{f} \le f_{\text{Nyquist}}$. This may occur when the sampling rate or downsampling operations fail to meet the Nyquist criterion, that is, the sampling frequency must be at least twice the highest frequency in the original signal. If not resolved, high-frequency components would fold back into lower frequencies during downsampling, creating spurious artifacts.

### A.2   PROBLEM DESCRIPTION: ALIASING IN MULTI-SCALE TIME SERIES DOWNSAMPLING

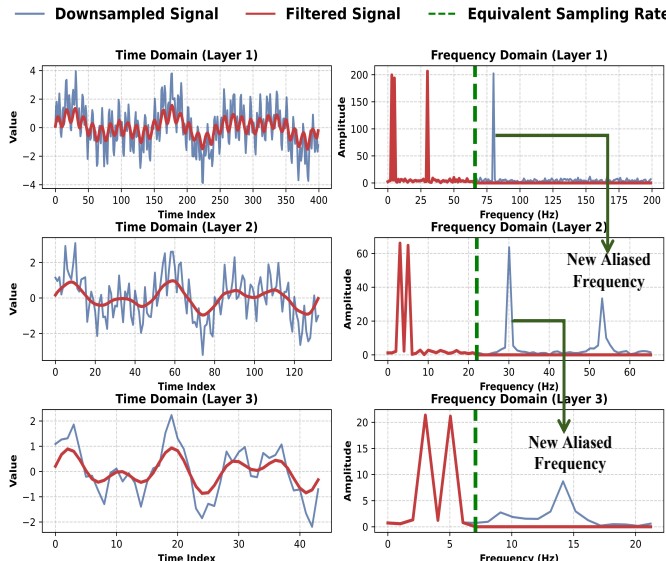

Figure 6: Left: filtering effect of the anti-aliasing filters; Right: emergence of new aliased frequen-cies.

In Figure.6, we present a case study exploring the critical role of anti-aliasing filters in signal preser-vation during multi-scale downsampling. By downsampling a synthetic signal containing both high-frequency and low-frequency components, we demonstrate the occurrence of aliasing during the reduction of the sampling rate.

The synthetic time series signal used in the study consists of several frequency components: low-frequency components (3 Hz and 5 Hz), high-frequency components (30 Hz and 80 Hz), and Gaus-

sian noise. The signal is initially sampled at a rate of 400 Hz. Subsequently, we perform multi-scale downsampling at different levels (each with a window size of 3), resulting in sampling rates of 400 Hz, 133 Hz, and 44 Hz.

According to the Nyquist sampling theorem, the Nyquist frequency is half of the sampling rate. Therefore, a 400 Hz sampling rate is sufficient to accurately sample the frequency components of the original signal. The calculation of aliasing frequencies is derived from the spectral periodicity characteristics of the Nyquist sampling theorem Nyquist (1928), based on the formula:

$$f_{\text{alias}} = |f_o - k \cdot f_s|,  \tag{12}$$

where $f_{\text{alias}}$ is the aliased frequency, $f_o$ is the original high-frequency component, $k$ is an integer representing the multiple mapping to the sampling frequency, and $f_s$ is the sampling rate.

In the first layer, the Nyquist frequency is 200 Hz, corresponding to a sampling rate of 400 Hz. Given that the highest frequency component of the signal is 80 Hz, which is well below the Nyquist frequency, no aliasing occurs; all frequency components can be accurately sampled and reconstructed. In practical applications, an anti-aliasing filter limits frequency components above 66 Hz, thereby preventing aliasing and removing high-frequency noise.

In the second layer, the Nyquist frequency is reduced to 66.67 Hz (corresponding to a sampling rate of approximately 133.33 Hz), which results in aliasing of the original 80 Hz high-frequency component. According to the aliasing formula (12), for $f_o = 80$ Hz and with $k = 1$:

$$f_{\text{alias}} = |80 - 1 \times 133.33| \approx 53.33 \, \text{Hz}.  \tag{13}$$

This calculation indicates that the 80 Hz component folds into the lower frequency region, specifically within the 50–60 Hz range, thereby introducing non-original frequency components and causing spectral distortion. The anti-aliasing filter in this layer effectively removes frequencies above 22 Hz to mitigate this issue.

In the third layer, the Nyquist frequency further decreases to 22.22 Hz (with a corresponding sampling rate of approximately 44.44 Hz), leading to the aliasing of the original 30 Hz component. Using the aliasing formula with $k = 1$:

$$f_{\text{alias}} = |30 - 1 \times 44.44| \approx 14.44 \, \text{Hz},  \tag{14}$$

indicating that the 30 Hz component folds around 14 Hz. Additionally, due to the interaction between the sampling rate and the sampling process, frequency components in the 10–15 Hz range cannot be accurately represented, even though they lie below the Nyquist frequency. This aliasing phenomenon becomes particularly significant as the frequencies approach the Nyquist limit. Nevertheless, the anti-aliasing filter is still able to extract the true frequency information with reasonable accuracy, thereby alleviating the impact of aliasing.

### A.3   An intuitive explanation of aliasing-related concepts

This appendix provides a detailed explanation of the core signal processing concepts illustrated in Figure.7, which motivate the design of DMANet. We structure this explanation to clarify the relationship between sampling, spectral overlap, aliasing, and our proposed solution.

### A.3.1   The Core Problem: Spectral Overlap and Aliasing

The central challenge DMANet addresses is aliasing, a form of signal distortion that occurs during downsampling. To fully understand this phenomenon, it is crucial to distinguish between its **physical cause spectral overlap** and its **perceptual consequence aliasing**. These two terms describe different links in a cause-and-effect chain, where aliasing is the direct result of spectral overlap due to improper sampling Zhou et al. (2025).

- **Sampling and Spectral Replicas**: When a signal is sampled, its original spectrum is periodically replicated along the frequency axis, creating what are known as spectral replicas. Grabinski et al. (2022b) The act of sampling also limits the frequency range we can observe without distortion to a new, narrower baseband (from 0 Hz to the new Nyquist frequency).

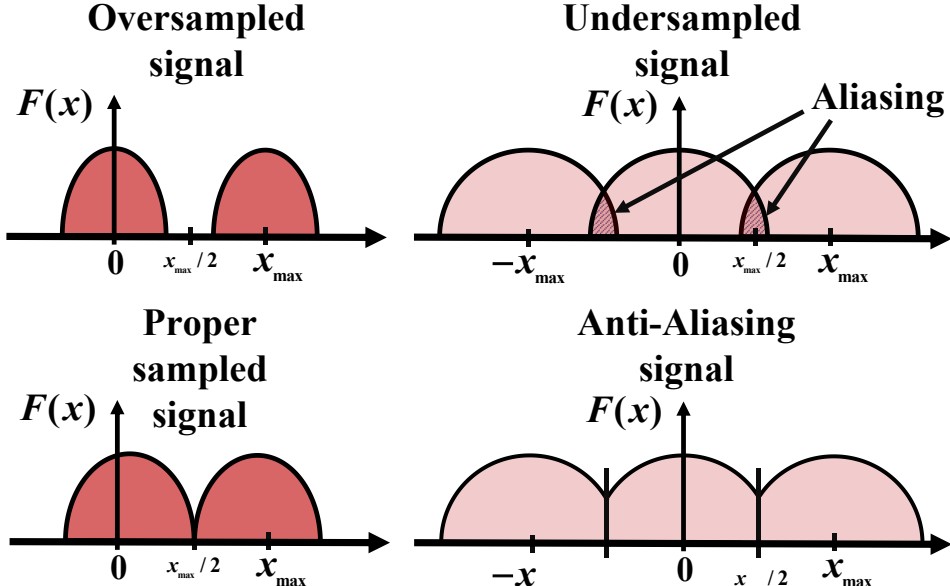

Figure 7: A conceptual illustration of the sampling process in the frequency domain. **Top-left:** Oversampling provides a wide guard band, preventing aliasing. **Top-right:** Undersampling causes spectral replicas to overlap, leading to aliasing where high frequencies (hatched areas) are misrepresented as low frequencies. **Bottom-left:** Proper (or critical) sampling meets the Nyquist criterion exactly, with replicas touching but not overlapping. **Bottom-right:** An anti-aliasing filter removes high-frequency content before sampling, ensuring that even with a lower sampling rate, no overlap occurs.

- **The Cause: Spectral Overlap.** If the sampling rate is too low to satisfy the Nyquist criterion, the separation between these spectral replicas becomes insufficient. This prevents the formation of a safety margin (Guard Band), causing them to physically overlap. This physical overlap, illustrated in the undersampled signal panel of Figure 7, is the root cause of the problem.

- **The Consequence: Aliasing.** This overlap is the direct cause of aliasing. Any high-frequency component from the original signal that exceeds the new Nyquist frequency is folded back into the new, observable low-frequency baseband. This process causes the original high-frequency information to appear as a spurious low frequency. This spurious frequency then mixes with the true low-frequency components within the baseband, becoming indistinguishable from them. Ultimately, this misrepresentation of high-frequency information as low-frequency information corrupts the signal's fidelity and is the core problem we address in our paper.

### A.3.2 A DEEPER DIVE INTO THE SCENARIOS OF FIGURE.7

Figure.7 visualizes four key scenarios:

- **The Undesirable Case (Top-Right):** The undersampled scenario is precisely the adverse outcome our work aims to prevent. The resulting aliasing (hatched areas) erroneously introduces spurious low-frequency patterns, a distortion that can severely hinder analysis and forecasting.

- **The Ideal and Safe Case (Top-Left):** The oversampled scenario is ideal because it successfully avoids aliasing. The empty space between the spectral replicas is a Guard Band, which does not imply information loss but rather a safe margin ensuring that the original spectrum can be unambiguously recovered.

- **The Theoretical Boundary (Bottom-Left):** Proper sampled (or Critical Sampling) occurs when the sampling rate is exactly twice the signal's maximum frequency ($f_s = 2 \cdot f_{max}$).

The spectral replicas touch edge-to-edge without overlapping. While theoretically sound, operating at this exact boundary is risky in practice.

- **The DMANet Approach (Bottom-Right):** The Anti-Aliasing signal panel illustrates the core principle of our work. Before an operation that would otherwise cause undersampling, an **anti-aliasing filter** is applied. This low-pass filter removes the high-frequency components (the part of the spectrum above $x_{max}/2$) that would cause overlap. After this pre-filtering, even a lower sampling rate can be safely applied without generating aliasing artifacts.

# B PROOFS OF THE EQUIVALENT SAMPLING RATE (ESR)

## B.1 NOTATIONS AND SIGNAL MODELING

### B.1.1 ORIGINAL DISCRETE-TIME SIGNAL

First, we define $X[n] \in \mathbb{R}^{C_i}$ as a discrete-time signal with $C_i$ channels. The $i$-th channel of $X[n]$, that is, $X_i[n]$, is obtained by sampling a continuous-time signal $x_i(t)$:

$$X_i[n] = x_i(nT_s), \; T_s = \frac{1}{f_s}, \; i = 1, \ldots, C_i, \tag{15}$$

where $f_s$ is the original sampling rate and $T_s$ is the sampling period. It is assumed that each continuous-time signal $x_i(t)$ is band-limited to $|\omega| \leq \omega_B$.

### B.1.2 MODULE STRUCTURE

The input signal $X[n]$ passes sequentially through a depth-wise convolution and a point-wise convolution defined in the architecture of DMANet:

$$X[n] \xrightarrow{\text{DepthwiseConv1d}(K,S)} U[m] \xrightarrow{\text{PointwiseConv1d}} V[m]. \tag{16}$$

First, the DepthwiseConv1d operation, characterized by a kernel $h_i[k]$ of length $K$ for each $i$-th input channel, where $1 \leq k \leq K$, and a stride $S$, transforms $X[n]$ into an intermediate signal $U[m] \in \mathbb{R}^{C_i}$. Subsequently, this intermediate signal $U[m]$ is processed by a PointwiseConv1d operation. This second stage uses a convolution matrix $W \in \mathbb{R}^{C_o \times C_i}$, with elements $w_{j,i}$, to map the $C_i$ channels of $U[m]$ to the $C_o$ output channels, producing the final output signal $V[m] \in \mathbb{R}^{C_o}$. Consequently, the output sampling rate of the entire module is $f_s' = f_s/S$.

## B.2 DOWNSAMPLING AND ALIASING CONDITIONS

As a baseline reference, consider directly downsampling the original signal $X[n]$ by a factor $S$ without any convolution filtering to obtain the signal $Y[m]$:

$$Y[m] = X[Sm]. \tag{17}$$

The new sampling rate is $f_s' = f_s/S$. To avoid aliasing caused directly by downsampling, the bandwidth $B = \omega_B/(2\pi)$ of the original continuous-time signal must satisfy the Nyquist-Shannon sampling theorem requirement, which is defined as:

$$B \leq \frac{f_s'}{2} = \frac{f_s}{2S}. \tag{18}$$

Expressed in terms of normalized angular frequency $\Omega_B = \omega_B T_s = 2\pi B T_s$, we solve this equation and represent the condition to avoid aliasing as:

$$\Omega_B \leq \frac{\pi}{S}. \tag{19}$$

This condition applies to ideal direct downsampling, assuming that perfect anti-aliasing filtering has been performed before downsampling to remove frequency components above $\pi/S$, corresponding to $\frac{f_s}{2S}$. Our proposed DMANet contains depth-wise and point-wise modules which perform filtering in its workflow, and its behaviors are more complex.

## B.3 LINEAR MAPPING WITH DEPTH-WISE CONVOLUTION AND POINT-WISE CONVOLUTION

The operations of these two modules can be expressed as a series of linear mappings.

### B.3.1 DEPTH-WISE CONVOLUTION

The output of the depth-wise convolution for the $i$-th channel, $U_i[m]$, is computed as follows:

$$U_i[m] = \sum_{k=0}^{K-1} h_i[k] X_i[mS + k]. \tag{20}$$

Here, $m$ is the index for the output sequence. Due to the stride $S$, the output $U_i[m]$ depends on the input $X_i[n]$ in the range from $n = mS$ to $n = mS + K - 1$.

### B.3.2 POINT-WISE CONVOLUTION

The $j$-th output channel $V_j[m]$ is obtained by a linear combination of $U_i[m]$:

$$V_j[m] = \sum_{i=1}^{C_i} w_{j,i} U_i[m]. \tag{21}$$

Then, we substitute equation 20 into the above equation to get the expression of $V_j[m]$:

$$V_j[m] = \sum_{i=1}^{C_i} w_{j,i} \left( \sum_{k=1}^{K} h_i[k] X_i[mS + k] \right) = \sum_{i=1}^{C_i} \sum_{k=1}^{K} w_{j,i} h_i[k] X_i[mS + k]. \tag{22}$$

### B.3.3 UNIFIED LINEAR MAPPING

To form a unified linear transformation at each output time $m$, we construct an input vector $\mathbf{x}_m$ that contains all original input samples involved in computing $V[m]$:

$$\mathbf{x}_m = \begin{bmatrix} X_1[mS] \\ \vdots \\ X_1[mS + K - 1] \\ \vdots \\ X_{C_i}[mS] \\ \vdots \\ X_{C_i}[mS + K - 1] \end{bmatrix} \in \mathbb{R}^{C_i K}.$$

This is a column vector formed by stacking $K$ consecutive samples, starting from $mS$, from each of the $C_i$ channels. Concurrently, a weight matrix $G \in \mathbb{R}^{C_o \times (C_i K)}$ is constructed. For the $j$-th row of $G$, its elements correspond to $w_{j,i} h_i[k]$ and are arranged according to the order of the elements in $\mathbf{x}_m$. Specifically, if the $p$-th element of $\mathbf{x}_m$ is $X_i[mS + k]$, then the weight in $G$ corresponds to the output $V_j[m]$ and this input element is $G_{j,p} = w_{j,i} h_i[k]$. Thus, the output $V[m]$ can be defined as:

$$V[m] = G\mathbf{x}_m, \quad V[m] \in \mathbb{R}^{C_o}. \tag{23}$$

This equation shows that at each output time $m$, the output vector $V[m]$ is a linear mapping of a local window $\mathbf{x}_m$ of the input signal.

### B.4 RANK CONSTRAINT AND DEGREES OF FREEDOM COUNTING

A fundamental property of linear algebra states that the rank of the matrix $G$, denoted as $\mathrm{rank}(G)$, is limited by its dimensions:

$$\mathrm{rank}(G) \leq \min\{\text{number of rows, number of columns}\} = \min\{C_o, C_i K\}. \tag{24}$$

We assume that the values of the weights $h_i[k]$ and $w_{j,i}$ are generic. They can be learned and are not overly sparse or linearly dependent, so that matrix $G$ can achieve its theoretically maximum possible rank. Then, the dimension of independent information, also named the degrees of freedom, that the module extracts from the $C_i K$-dimensional input window $\mathbf{x}_m$ and transmits to the $C_o$-dimensional output $V[m]$ at each output time $m$ is:

$$D = \mathrm{rank}(G) = \min\{C_i K, C_o\}. \tag{25}$$

This $D$ represents the maximum dimension of linearly independent information from the input segment $\mathbf{x}_m$ that the system can distinguish or represent, without considering noise or specific signal statistics.

To relate this total degree of freedom $D$ to each channel of the input signal, we can average it over the $C_i$ input channels. Thus, the equivalent temporal degrees of freedom $\alpha$ contributed by each input channel to produce one output sample $V[m]$ is:

$$\alpha = \frac{D}{C_i} = \frac{\min\{C_i K, C_o\}}{C_i} = \min\left\{ \frac{C_i K}{C_i}, \frac{C_o}{C_i} \right\} = \min\left\{ K, \frac{C_o}{C_i} \right\}. \tag{26}$$

Here, $\alpha$ can be understood as: for each input channel, its information, under the combined effect of temporal processing through kernel length $K$ and inter-channel mapping via $C_o/C_i$, is refined or compressed to be equivalent to $\alpha$ independent information units. These units contribute to the final output sample $V[m]$. The bottleneck here is determined by the smaller of $K$ (temporal context length per channel) and $C_o/C_i$ (channel transformation ratio).

## B.5 DEFINITION OF EQUIVALENT SAMPLING RATE

The actual output sampling rate of the module for each output channel is $f_s' = f_s/S$. At each output sampling instant, we have determined that each input channel contributes $\alpha = \min\{K, C_o/C_i\}$ equivalent temporal degrees of freedom.

The Equivalent Sampling Rate $f_{\text{ESR}}$ is defined as a rate such that if each of the original $C_i$ input channels were sampled at $f_{\text{ESR}}$, and each sample carried one independent degree of freedom, then its total degrees of freedom throughput would match that of the current depth-wise and point-wise modules.

The total rate of generating degrees of freedom is: $D = \min\{C_iK, C_o\} \times \frac{f_s}{S}$. If $C_i$ channels each operate at an equivalent sampling rate of $f_{\text{ESR}}$, their total degrees of freedom rate is $C_i \times f_{\text{ESR}}$:

$$C_i \times f_{\text{ESR}} = \min\{C_iK, C_o\} \times \frac{f_s}{S}. \tag{27}$$

Then, we can get the solve for $f_{\text{ESR}}$:

$$f_{\text{ESR}} = \frac{\min\{C_iK, C_o\}}{C_i} \times \frac{f_s}{S} = \min\left\{K, \frac{C_o}{C_i}\right\} \times \frac{f_s}{S}. \tag{28}$$

If we normalize the original sampling rate $f_s$ to 1, we obtain the normalized ESR:

$$\text{ESR}_{\text{norm}} = \frac{1}{S}\min\left\{K, \frac{C_o}{C_i}\right\}. \tag{29}$$

Based on this equivalent sampling rate $f_{\text{ESR}}$, we can define an equivalent Nyquist frequency $f_{\text{Nyq\_ESR}}$. This frequency represents the maximum bandwidth that the input signal can accommodate without information loss due to module structural limitations:

$$f_{\text{Nyq\_ESR}} = \frac{f_{\text{ESR}}}{2} = \frac{f_s}{2S}\min\left\{K, \frac{C_o}{C_i}\right\}. \tag{30}$$

We can use $f_{\text{ESR}}$ to quantify the information processing capability or information retention degree of the downsampling module consisting of depth-wise convolution and point-wise convolution relative to each input channel. It provides a useful metric to compare the effective information throughput of modules with different parameter configurations with $K$, $S$, $C_i$, and $C_o$. It is important to note that the anti-aliasing significance of $f_{\text{Nyq\_ESR}}$ also depends on whether the depth-wise convolution kernel $h_i[k]$ can effectively act as a low-pass filter to attenuate frequency components above $f_{\text{Nyq\_ESR}}$. If $h_i[k]$ is not an ideal low-pass filter, the frequency components of the original signal above $f_{\text{Nyq\_ESR}}$, even if not completely filtered out by $h_i[k]$, may not be accurately represented by the output $V[m]$ due to subsequent dimensionality reduction.

## B.6 THEORETICAL ANALYSIS: ORTHOGONALITY OF DMANET ARCHITECTURE AND FREQUENCY LOSS

To clarify that the effectiveness of DMANet stems from its architectural design rather than sole reliance on the loss function, we mathematically decompose the total forecasting error $\mathcal{E}_{\text{total}}$ into two orthogonal components: **Feature Representation Error ($\mathcal{E}_{\text{feat}}$)** and **Optimization Objective Bias ($\mathcal{E}_{\text{loss}}$)**. First, regarding $\mathcal{E}_{\text{feat}}$, theoretically, an ideal downsampling operation with stride $s$ truncates the spectrum using a low-pass filter (LPF), yielding an ideal feature $h_{\text{ideal}}$ such that $\mathcal{F}(h_{\text{ideal}})(\omega) = \mathcal{F}(X)(\omega) \cdot \mathbb{I}(|\omega| \leq \pi/s)$. However, standard strided convolutions lack this LPF,

causing high-frequency components to fold into the baseband and resulting in an aliased spectrum $\mathcal{F}(h_{\text{trad}})(\omega) = \sum_{k=-\infty}^{\infty} \mathcal{F}(X)(\omega - k\omega_s')$. Here, $k$ is the spectrum replication index and $\omega_s'$ equals to $2\pi/s$. Consequently, by Parseval's Theorem Blu & Unser (2001), the representation error is dominated by these aliasing terms ($k \neq 0$):

$$\mathcal{E}_{\text{feat}} = ||h_{\text{trad}} - h_{\text{ideal}}||^2 \approx \frac{1}{2\pi} \int_{|\omega| \leq \pi/s} \left| \sum_{k \neq 0} \mathcal{F}(X)(\omega - k\omega_s') \right|^2 d\omega, \tag{31}$$

DMANet directly minimizes this term by applying an ESR-based dynamic filter before downsampling to physically eliminate the aliasing sums ($\sum_{k \neq 0} \cdots \to 0$), ensuring the latent feature structurally approximates the ideal signal independent of the loss function. Conversely, regarding $\mathcal{E}_{\text{loss}}$, time-domain MSE minimization often suffers from gradient bias due to the strong autocorrelation ($\rho \neq 0$) in time series labels; FreDF minimizes this bias by exploiting the Theorem of Spectral Decorrelation, which effectively decouples error terms in an orthogonal basis ($\mathcal{E}_{\text{loss\_Freq}} \ll \mathcal{E}_{\text{loss\_MSE}}$) Wang et al. (2025). Thus, the total error is unified as:

$$\mathcal{E}_{\text{total}} \approx \underbrace{\mathcal{L}(p(h_{\text{ideal}} + \mathcal{E}_{\text{feat}}), Y)}_{\text{Feature Fidelity (Architecture-dependent)}} + \underbrace{\mathcal{E}_{\text{loss}}}_{\text{Gradient Quality (Loss-dependent)}}, \tag{32}$$

where $p(\cdot)$ denotes the predictor head mapping latent features to forecasts, and $\mathcal{L}$ is the loss function. This formulation demonstrates that while removing Frequency Loss degrades performance by increasing $\mathcal{E}_{\text{loss}}$, the DMANet architecture remains essential for minimizing $\mathcal{E}_{\text{feat}}$ (spectral aliasing), proving their relationship is **orthogonal and synergistic**.

## C    IMPLEMENTATION DETAILS

We summarized details of datasets, evaluation metrics, experiments in this section.

### C.1    DATASETS DETAILS

We evaluated the performance of different models on several well-established datasets for long-term forecasting, including Weather, Electricity, Solar-Energy, PeMS(PEMS03, PEMS04, PEMS07, PEMS08), and the ETT series (ETTh1, ETTh2, ETTm1, ETTm2). Furthermore, to demonstrate DMANet's capability in handling highly non-stationary data, we conducted an extensive series of supplementary experiments on short-term forecasting across datasets from various domains. These include Health & Medical (ILI, COVID-19), Web Events (Wiki, Website), Finance (NASDAQ, SP500, DowJones), Market (CarSales), Energy (Power), and Society (Unemp). We detail the descriptions of the dataset in Table.8.

### C.2    BASELINE DETAILS

Acknowledging that the performance of different methods varies across scenarios, we conducted a comprehensive comparison of various approaches under three distinct settings: long-term forecasting with a lookback window of 96, long-term forecasting with a lookback window of 720, and short-term forecasting. The evaluated methods are categorized as follows:

- **Frequency-domain methods:** TimeStacker Liu et al. (2025), FilterNet Yi et al. (2024a), FITS Xu et al. (2024), Fredformer Piao et al. (2024a), FEDformer Zhou et al. (2022b).

- **CNN-based methods:** ModernTCN Donghao & Xue (2024), TVNet Li et al. (2025), TSLANet Eldele et al. (2024), TimesNet Wu et al. (2023), PDF Dai et al. (2024), MICN Wang et al. (2023).

- **MLP-based methods:** SOFTS Han et al. (2024), TimeMixer Wang et al. (2024a), DLinear Zeng et al. (2023), TiDE Das et al. (2023), RLinear Li et al. (2023b), MTS-Mixer Li et al. (2023c)

- **Transformer-based methods:** TimeXer Wang et al. (2024b),iTransformer Liu et al. (2024), Crossformer Zhang & Yan (2022), Pathformer Chen et al. (2024b), Stationary Liu et al. (2022b), Pyraformer Liu et al. (2022a), Autoformer Wu et al. (2021)

- **LLM-based methods:** GPT4TS Zhou et al. (2023), Time-LLM Jin et al. (2024)

- **KAN-based methods:** TimeKAN Huang et al. (2025)

- **Mamba-based methods:** TimePro Ma et al. (2025)

- **Retrieval-Augmented methods:** RAFT Han et al. (2025)

### C.3    IMPLEMENTATION DETAILS.

Regarding evaluation metrics, we used mean square error (MSE) and mean absolute error (MAE) for both long-term and short-term forecasting. All experiments were conducted using PyTorch on a single NVIDIA GeForce RTX 3090 24GB GPU. We applied an early stopping strategy to all baselines when the validation loss did not decrease for three consecutive epochs. Notably, inspired by FreDF Wang et al. (2025), we argue that formulating the loss function in the frequency domain is advantageous for learning an anti-aliasing architecture. Consequently, we directly adopted the frequency-domain MAE as the loss function for both long-term and short-term forecasting. More detailed settings can be found in Appendix.C.5.

### C.4    FAIR COMPARISON SETTINGS.

To ensure a fair comparison and address challenges related to scaling laws, we maintained a consistent lookback window of 96 for all experiments in Table.11 and Table 12, and 720 for all experiments in Table.13 and Table.14. Our baseline comparisons mimic the experimental protocols established in TimesNet Wu et al. (2023), including same data processing and splitting procedures. For most

Table 8: Detailed dataset descriptions and statistics. **Dim** denotes the number of variates for each dataset. **Frequency** refers to the time interval between consecutive steps. **Split** indicates the data partitioning ratio (Train/Validation/Test). **Prediction len.** represents the prediction lengths. Our long-term forecasting employs a fixed input length of 96 or 720. For the majority of datasets, we evaluate across prediction horizons of 96, 192, 336, 720. A distinct setting is applied to the PeMS datasets, which are evaluated on shorter horizons of 12, 24, 48. For short-term forecasting, we adopt two settings: one with an input of 12 steps to predict 3, 6, 9, 12 steps, and another with an input of 36 steps to predict 24, 36, 48, 60 steps.

| Dataset | Dim | Frequency | Total len. | Split | Prediction len. | Information |
|---|---|---|---|---|---|---|
| ETTh1, ETTh2 | 7 | Hourly | 17420 | 6:2:2 | {96,192,336,720} | Electricity |
| ETTm1, ETTm2 | 7 | 15 mins | 69680 | 6:2:2 | {96,192,336,720} | Electricity |
| Weather | 21 | 10 mins | 52696 | 7:1:2 | {96,192,336,720} | Weather |
| ECL | 321 | Hourly | 26304 | 7:1:2 | {96,192,336,720} | Electricity |
| Solar-Energy | 137 | 10 mins | 52560 | 7:1:2 | {96,192,336,720} | Energy |
| PEMS03 | 358 | 5 mins | 26209 | 6:2:2 | {12,24,48} | Transportation |
| PEMS04 | 307 | 5 mins | 16992 | 6:2:2 | {12,24,48} | Transportation |
| PEMS07 | 883 | 5 mins | 28224 | 6:2:2 | {12,24,48} | Transportation |
| PEMS08 | 170 | 5 mins | 17856 | 6:2:2 | {12,24,48} | Transportation |
| ILI | 7 | Weekly | 966 | 7:1:2 | {24,36,48,60} | Health |
| COVID-19 | 55 | Daily | 335 | 7:1:2 | {3,6,9,12} | Health |
| NASDAQ | 12 | Daily | 3914 | 7:1:2 | {24,36,48,60} | Finance |
| SP500 | 5 | Daily | 8077 | 7:1:2 | {24,36,48,60} | Finance |
| DowJones | 27 | Daily | 6577 | 7:1:2 | {24,36,48,60} | Finance |
| CarSales | 10 | Daily | 6728 | 7:1:2 | {24,36,48,60} | Market |
| Power | 2 | Daily | 1186 | 7:1:2 | {24,36,48,60} | Energy |
| Website | 4 | Daily | 2167 | 7:1:2 | {3,6,9,12} | Web |
| Wiki | 99 | Daily | 730 | 7:1:2 | {3,6,9,12} | Web |
| Unemp | 53 | Monthly | 531 | 6:2:2 | {3,6,9,12} | Society |

methods, we adopted the results reported in their original papers. For some methods that did not report results on the Solar-Energy dataset, we reproduced their performance using their official code repositories. The results for FITS Xu et al. (2024) and FreTSWang et al. (2025) were replicated from the FilterNet report Yi et al. (2024a); for other methods, we used the long-term prediction results provided in the iTransformer repository Liu et al. (2024). These results are based on the experimental configurations provided in the original paper or official code for each model. We verified that all hyperparameters for these baselines were selected from their respective official repositories, ensuring consistency with our fair comparison setup, where the only variations were the input and output sequence lengths.

For the experiments with the lookback window extended to 720, we referred to established baseline results: results in Table.13 were replicated from DUET Qiu et al. (2024), the results for GPT4TS Zhou et al. (2023) and TimeLLM Jin et al. (2024) in Table.14 were replicated from TSLANet Eldele et al. (2024), and the remaining results in Table.14 were replicated from TVNet Li et al. (2025). For short-term forecasting, we followed the results from the FreEformer repository Yue et al. (2025).

### C.5 HYPERPARAMETER SETTINGS.

**Primary Long-term Forecasting Task**   For our model hyperparameter selection, in 96 lookback window long-term forecasting, we fixed $d_{\text{model}} = 512$, downsampling layer $l$ to 2, depth-wise convolution kernal size $K$ to 3, stride $s$ to 2, and set the proportion of channel changes $c$ to 0.5. And we only performed a limited search on the encoder layers $E$, learning rate $LR$, and batch size. Detailed configurations for each dataset can be found in Table.9.

**Other Long-term Forecasting Tasks**   For long-term forecasting with an extended 720 lookback window, as well as for the 96 lookback forecasting on PEMS datasets and 336 lookback univariate forecasting tasks, we implemented a more extensive hyperparameter search. This search was conducted for each forecast horizon within a given dataset to find the optimal configuration. The search space was defined as follows: $d_{model} \in \{256, 512\}$, Learning Rate $LR \in \{1 \times 10^{-3}, 2 \times 10^{-3}, 5 \times 10^{-3}, 1 \times 10^{-2}, 2 \times 10^{-2}\}$, Encoder Layers $E \in \{1, 2, 3\}$, Downsampling Layers $l \in \{2, 3, 4\}$, Batch Size $\in \{8, 16, 32, 64\}$. Other hyperparameters, such as the convolutional kernel size and stride, remained fixed across all experiments, consistent with the settings used in the primary 96 lookback forecasting task. In contrast to all baseline lookback windows searched from $\{192, 336, 512, 672, 720\}$ etc., We provide long-term forecasting for the fixed 720 lookback window.

**Short-term Forecasting**   We implemented a more extensive hyperparameter search like Other Long-term Forecasting Tasks. This search was conducted for each forecast horizon within a given dataset to find the optimal configuration. The search space was defined as follows: Downsampling Layers is fixed 2, $d_{model} \in \{256, 512\}$, Learning Rate $LR \in \{1 \times 10^{-3}, 2 \times 10^{-3}, 5 \times 10^{-3}, 1 \times 10^{-2}, 2 \times 10^{-2}\}$, Encoder Layers $E \in \{1, 2\}$, Batch Size $\in \{2, 4, 8, 16\}$. Other hyperparameters, such as the convolutional kernel size and stride, remained fixed across all experiments, consistent with the settings used in the primary 96-lookback forecasting task.

**Ablation Study on Pre-Sampling Filtering**   To validate our ESR-based filtering approach, we conducted an ablation study comparing it against alternatives that do not adhere to the Nyquist sampling theorem. Each experimental group differs from our full DMANet only in the cutoff frequency determination method within the Pre-Sampling Filtering module; all other structures and parameters remain identical. We categorize the compared methods into two groups: heuristic and classical filters.

HEURISTIC FILTERS   These methods serve as simple, non-theoretical baselines. They are designed to mimic intuitive or simplistic approaches to filtering that one might adopt without a rigorous signal processing foundation.

- **Max:** For each time series in the batch, this filter identifies the frequency bin with the maximum amplitude and sets the cutoff frequency to twice its index. All components below this dynamic cutoff are preserved, while those above are zeroed out.
- **Random:** This filter applies a stochastic mask to the frequency spectrum, where each frequency component is independently dropped with a probability of $p = 0.5$.

CLASSICAL FILTERS    These methods serve as benchmarks against well-established, theoretically-grounded filtering techniques. To ensure a fair comparison, a normalized cutoff frequency of 0.4 was used across all classic filter variants, preserving the lowest 80% of the frequency band.

- **Ideal:** A sharp cutoff filter where all frequency components above the cutoff frequency are set to zero.
- **Butterworth:** Known for its maximally flat passband, providing high-fidelity signal preservation. We used a 4th-order filter.
- **Gaussian:** A smooth filter often used to avoid ringing artifacts, with a sigma of 0.15.
- **Chebyshev (Type I):** Achieves a steeper rolloff than Butterworth at the cost of introducing ripples in the passband. We used a 4th-order filter with 0.5 dB of passband ripple.

Table 9: Experiment configuration of DMANet in 96 lookback window. All the experiments use the ADAM optimizer with the default hyperparameter configuration for $(\beta_1, \beta_2)$ as (0.9, 0.999).

| Dataset / Configurations | Model Hyper-parameter | | | Training Process | | | |
|---|---|---|---|---|---|---|---|
| | $E$ | $l$ | $d_{\text{model}}$ | LR* | Loss | Batch Size | Epochs |
| ETTh1 | 1 | 2 | 512 | $2 * 10^{-2}$ | MAE | 8 | 15 |
| ETTh2 | 1 | 2 | 512 | $1 * 10^{-2}$ | MAE | 8 | 15 |
| ETTm1 | 1 | 2 | 512 | $2 * 10^{-3}$ | MAE | 16 | 15 |
| ETTm2 | 2 | 2 | 512 | $5 * 10^{-3}$ | MAE | 32 | 15 |
| Weather | 1 | 2 | 512 | $5 * 10^{-3}$ | MAE | 16 | 15 |
| Electricity | 2 | 2 | 512 | $1 * 10^{-3}$ | MAE | 8 | 15 |
| Solar-Energy | 2 | 2 | 512 | $5 * 10^{-3}$ | MAE | 16 | 15 |

  ∗ LR means the initial learning rate.

## C.6    ABLATION STUDY ON COMPONENT CONTRIBUTIONS

To thoroughly investigate the individual and synergistic contributions of the Anti-Aliasing Down-sampling and Frequency-Domain Upsampling modules, we conducted a rigorous component disen-tanglement experiment. As presented in Table.5, we designed four distinct configurations to isolate the effect of each module.

- **Base:** Represents a standard convolutional architecture without our specific anti-aliasing designs. It employs standard strided convolution for downsampling and linear interpolation for upsampling.
- **w/o-Post:** Retains the proposed Pre-Sampling Anti-Aliasing filter but reverts the upsam-pling mechanism to standard linear interpolation. This setup isolates the net benefit of preventing aliasing during the feature extraction stage.
- **w/o-Pre:** Removes the Pre-Sampling filter (using standard strided convolution) but retains the proposed Post-Sampling Frequency Interpolation. This setup isolates the contribution of high-fidelity signal reconstruction in the frequency domain.

Here is the Analysis of Results. First, regarding the individual effectiveness of each component, comparing the variants against the Base model reveals that both modules independently contribute to significant performance gains. Specifically, the benefit of Anti-Aliasing Downsampling is evident as the w/o-Post variant consistently outperforms the Base model (e.g., on the Unemp dataset, MAE decreases from 0.173 to 0.155). This confirms that proactively filtering high-frequency noise before downsampling enables the encoder to learn cleaner, non-aliased latent representations, even when reconstructed with a suboptimal upsampler. Simultaneously, the w/o-Pre variant demonstrates the

benefit of Frequency Upsampling, showing clear improvement over the Base model (e.g., Unemp MAE reduces to $0.161$). This validates that our frequency-domain zero-padding strategy, which adheres to the Nyquist-Shannon sampling theorem, offers a more mathematically sound reconstruction basis than linear interpolation.

Furthermore, regarding the synergistic superiority, the full DMANet achieves the lowest error across all datasets (e.g., Unemp MAE drops to $0.146$), crucially surpassing both w/o-Pre and w/o-Post. This result highlights the architectural synergy: high-quality reconstruction is most effective only when the input features are initially free from aliasing artifacts. Conversely, clean downsampled features are best utilized when restored without the spectral distortion introduced by linear interpolation. Consequently, the combination of these two modules is not merely additive but essential for achieving state-of-the-art performance.

# D FULL RESULTS

## D.1 ERROR BARS

To evaluate the performance stability and robustness of DMANet, we conducted multiple independent runs with five different random seeds and compared its performance against the second-best model, TimeMixer. The results, averaged over four prediction horizons (96, 192, 336, and 720), are presented in Table 10. We report the mean and standard deviation of the MSE and MAE metricsacross the five experiments, as well as the confidence level of DMANet's superiority over TimeMixer. This performance improvement is statistically significant, with a 99% confidence level in all evaluated scenarios.

Table 10: Standard deviation and statistical tests for our DMANet method and second-best method (TimeMixer) on five datasets.

| Metric | MSE | | | MAE | | |
|---|---|---|---|---|---|---|
| Dataset | DMANet | TimeMixer | Confidence | DMANet | TimeMixer | Confidence |
| ETTm1 | **0.376±0.005** | 0.386±0.003 | 99% | **0.388±0.003** | 0.399±0.001 | 99% |
| ETTm2 | **0.269±0.007** | 0.278±0.001 | 99% | **0.311±0.005** | 0.325±0.001 | 99% |
| Weather | **0.238±0.005** | 0.245±0.001 | 99% | **0.263±0.005** | 0.276±0.001 | 99% |
| Electricity | **0.171±0.002** | 0.182±0.002 | 99% | **0.264±0.002** | 0.272±0.002 | 99% |
| Solar-Energy | **0.228±0.003** | 0.235±0.001 | 99% | **0.249±0.002** | 0.292±0.001 | 99% |

## D.2 LONG-TERM FORECASTING

Here, Table.11, Table.12, Table.13 and Table.14 present comprehensive evaluation results for long-term forecasting, including both configurations with fixed lookback windows $L = 96$ and extended window settings $L = 720$ designed to adhere to the scaling law inherent to TSF. In the $L = 96$ fixed-window experiments, **consistent hyperparameters** were maintained across all forecast horizons within each dataset. By contrast, the $L = 720$ experiments employed horizon-specific hyperparameter adjustments to enhance model adaptability while preserving scaling law compliance. Under both experimental paradigms, DMANet consistently demonstrates superior performance with statistically significant margins, thereby empirically validating its effectiveness and robustness. Notably, even when handling extended sequence lengths through augmented lookback windows, DMANet retains its inherent capability to adaptively model critical dependencies within extended temporal sequences.

The results for PESM dataset forecasting, presented in Table.15 for a lookback window of $L = 96$ and the forecasting horizon $T \in \{12, 24, 48\}$, demonstrate the exceptional capability of DMANet. Across all four PEMS datasets, DMANet consistently outperforms all baselines. This superiority is quantified by average reductions of 14.4% in MSE and 5.7% in MAE compared to a strong baseline, iTransformer. We attribute this robust performance to our convolutional architecture's inherent proficiency in preserving localized features and mitigating the interference of high-frequency noise, which are critical for high-dimensional short-term prediction.

## D.3 SHORT-TERM FORECASTING

The short-term forecasting results, presented in Table.17, validate the superiority of DMANet in handling highly non-stationary time series. Across a diverse set of challenging datasets including ILI (health), COVID-19 (pandemic), DowJones (finance), and Unemp (society), DMANet consistently achieves state-of-the-art performance, securing the top rank in 17 out of 20 metrics. It significantly outperforms other methods, including strong frequency-domain baselines like Fredformer and FilterNet. We attribute this exceptional capability in short-term and non-stationary forecasting to DMANet's synergistic design: its convolutional architecture excels at preserving local features, while the anti-aliasing structure effectively mitigates disruptive high-frequency noise. This robust performance on volatile, real-world data underscores the effectiveness of our approach in capturing the transient and complex patterns inherent to non-stationary signals.

## D.4 UNIVARIATE FORECASTING

Here we provide the univariate forecasting results on ETT datasets. There is a target feature oil temperature within those datasets, which is the univariate time series that we are trying to forecast. As shown in Table.16 , the anti-aliasing depth-wise convolution has better temporal modelling capabilities, allowing DMANet to achieve better performance than the state-of-the-art CNN-based ModerTCN in univariate forecasting tasks.

Table 11: Full results of long-term forecasting with a 96-step lookback window (Part I). The input sequence length $L$ is set to 96 for all baselines. All results are averaged across four different forecasting horizon: $T \in \{96, 192, 336, 720\}$. The best and second-best results are highlighted in **bold** and underlined, respectively. Among them, - means that the code has not yet been open sourced. We will put the summary table in the appendix of the next version.

| Models | | DMANet Ours | | TimeStacker 2025 | | TimeXer 2024b | | iTransformer 2024 | | TimeMixer 2024a | | FilterNet 2024a | | Fredformer 2024a | | FITS 2024 | | FreTS 2024b | |
|---|---|---|---|---|---|---|---|---|---|---|---|---|---|---|---|---|---|---|---|
| | Metric | MSE | MAE | MSE | MAE | MSE | MAE | MSE | MAE | MSE | MAE | MSE | MAE | MSE | MAE | MSE | MAE | MSE | MAE |
| ETTm1 | 96 | **0.308** | 0.343 | 0.311 | **0.337** | 0.318 | 0.356 | 0.334 | 0.368 | 0.320 | 0.357 | 0.318 | 0.358 | 0.326 | 0.361 | 0.355 | 0.375 | 0.339 | 0.374 |
| | 192 | **0.354** | 0.372 | 0.364 | **0.367** | 0.362 | 0.383 | 0.377 | 0.391 | 0.361 | 0.381 | 0.364 | 0.383 | 0.363 | 0.380 | 0.392 | 0.393 | 0.382 | 0.397 |
| | 336 | **0.384** | 0.394 | 0.389 | **0.391** | 0.395 | 0.407 | 0.426 | 0.420 | 0.390 | 0.404 | 0.396 | 0.406 | 0.395 | 0.403 | 0.424 | 0.414 | 0.421 | 0.426 |
| | 720 | **0.447** | 0.431 | 0.460 | **0.428** | 0.452 | 0.441 | 0.491 | 0.459 | 0.454 | 0.441 | 0.456 | 0.444 | 0.453 | 0.438 | 0.487 | 0.449 | 0.485 | 0.462 |
| | Avg. | **0.373** | 0.385 | 0.381 | **0.381** | 0.382 | 0.397 | 0.407 | 0.410 | 0.381 | 0.395 | 0.384 | 0.398 | 0.393 | 0.403 | 0.387 | 0.408 | 0.407 | 0.415 |
| ETTm2 | 96 | **0.165** | **0.244** | 0.171 | 0.250 | 0.171 | 0.256 | 0.180 | 0.264 | 0.175 | 0.258 | 0.174 | 0.257 | 0.177 | 0.259 | 0.183 | 0.266 | 0.190 | 0.282 |
| | 192 | **0.231** | **0.288** | 0.235 | 0.292 | 0.237 | 0.299 | 0.250 | 0.309 | 0.237 | 0.299 | 0.240 | 0.300 | 0.241 | 0.300 | 0.247 | 0.305 | 0.260 | 0.329 |
| | 336 | **0.289** | **0.325** | 0.293 | 0.329 | 0.296 | 0.338 | 0.311 | 0.348 | 0.298 | 0.340 | 0.297 | 0.339 | 0.302 | 0.340 | 0.307 | 0.342 | 0.373 | 0.405 |
| | 720 | **0.385** | **0.383** | 0.395 | 0.391 | 0.392 | 0.394 | 0.412 | 0.407 | 0.391 | 0.396 | 0.392 | 0.393 | 0.397 | 0.396 | 0.407 | 0.399 | 0.517 | 0.499 |
| | Avg. | **0.268** | **0.310** | 0.274 | 0.316 | 0.274 | 0.322 | 0.288 | 0.332 | 0.275 | 0.323 | 0.276 | 0.322 | 0.279 | 0.324 | 0.286 | 0.328 | 0.335 | 0.379 |
| ETTh1 | 96 | **0.370** | 0.391 | **0.379** | 0.385 | 0.382 | 0.403 | 0.386 | 0.405 | 0.375 | 0.400 | 0.375 | 0.394 | 0.376 | 0.394 | 0.386 | 0.396 | 0.399 | 0.412 |
| | 192 | **0.417** | 0.420 | 0.429 | **0.416** | 0.429 | 0.435 | 0.441 | 0.436 | 0.429 | 0.421 | 0.436 | 0.422 | 0.440 | 0.425 | 0.436 | 0.423 | 0.453 | 0.443 |
| | 336 | **0.457** | 0.440 | 0.459 | **0.436** | 0.468 | 0.448 | 0.487 | 0.458 | 0.484 | 0.458 | 0.476 | 0.443 | 0.472 | 0.440 | 0.478 | 0.444 | 0.503 | 0.475 |
| | 720 | 0.468 | 0.465 | **0.464** | **0.455** | 0.469 | 0.461 | 0.503 | 0.491 | 0.498 | 0.482 | 0.474 | 0.469 | 0.490 | 0.467 | 0.502 | 0.495 | 0.596 | 0.565 |
| | Avg. | **0.428** | 0.429 | 0.433 | **0.423** | 0.437 | 0.437 | 0.454 | 0.447 | 0.447 | 0.440 | 0.440 | 0.432 | 0.445 | 0.432 | 0.447 | 0.448 | 0.488 | 0.474 |
| ETTh2 | 96 | **0.280** | 0.329 | **0.280** | 0.327 | 0.286 | 0.338 | 0.297 | 0.349 | 0.289 | 0.341 | 0.292 | 0.343 | 0.292 | 0.343 | 0.295 | 0.350 | 0.350 | 0.403 |
| | 192 | **0.349** | **0.374** | 0.373 | 0.385 | 0.363 | 0.389 | 0.380 | 0.400 | 0.372 | 0.392 | 0.369 | 0.395 | 0.370 | 0.390 | 0.381 | 0.396 | 0.472 | 0.475 |
| | 336 | 0.393 | **0.410** | 0.407 | 0.416 | 0.414 | 0.423 | 0.428 | 0.432 | 0.386 | 0.414 | 0.420 | 0.432 | **0.385** | 0.413 | 0.426 | 0.438 | 0.564 | 0.528 |
| | 720 | 0.418 | 0.437 | 0.412 | **0.431** | **0.408** | 0.432 | 0.427 | 0.445 | 0.412 | 0.434 | 0.430 | 0.446 | 0.419 | 0.439 | 0.431 | 0.446 | 0.815 | 0.654 |
| | Avg. | **0.361** | **0.388** | 0.368 | 0.390 | 0.367 | 0.396 | 0.383 | 0.407 | 0.364 | 0.395 | 0.378 | 0.397 | 0.367 | 0.396 | 0.383 | 0.408 | 0.550 | 0.515 |
| Weather | 96 | **0.148** | **0.191** | 0.161 | 0.198 | 0.157 | 0.205 | 0.174 | 0.214 | 0.163 | 0.209 | 0.164 | 0.210 | 0.163 | 0.207 | 0.166 | 0.213 | 0.184 | 0.239 |
| | 192 | **0.199** | **0.238** | 0.207 | 0.241 | 0.204 | 0.247 | 0.221 | 0.254 | 0.208 | 0.250 | 0.214 | 0.252 | 0.211 | 0.251 | 0.213 | 0.254 | 0.223 | 0.275 |
| | 336 | 0.256 | 0.282 | 0.261 | **0.281** | 0.261 | 0.290 | 0.278 | 0.296 | **0.251** | 0.287 | 0.268 | 0.293 | 0.267 | 0.292 | 0.269 | 0.294 | 0.272 | 0.316 |
| | 720 | **0.339** | 0.336 | 0.343 | **0.334** | 0.340 | 0.341 | 0.358 | 0.349 | **0.339** | 0.341 | 0.344 | 0.342 | 0.343 | 0.341 | 0.346 | 0.343 | 0.340 | 0.363 |
| | Avg. | **0.236** | **0.262** | 0.243 | 0.264 | 0.241 | 0.271 | 0.258 | 0.279 | 0.240 | 0.271 | 0.248 | 0.278 | 0.246 | 0.272 | 0.249 | 0.276 | 0.255 | 0.363 |
| Electricity | 96 | **0.139** | **0.234** | 0.168 | 0.251 | 0.140 | 0.242 | 0.148 | 0.240 | 0.153 | 0.247 | 0.176 | 0.264 | 0.147 | 0.241 | 0.200 | 0.278 | 0.183 | 0.269 |
| | 192 | **0.157** | **0.250** | 0.176 | 0.262 | 0.157 | 0.256 | 0.162 | 0.253 | 0.166 | 0.256 | 0.185 | 0.270 | 0.165 | 0.258 | 0.200 | 0.280 | 0.187 | 0.276 |
| | 336 | **0.175** | **0.269** | 0.195 | 0.278 | 0.176 | 0.275 | 0.178 | 0.269 | 0.185 | 0.277 | 0.202 | 0.286 | 0.177 | 0.273 | 0.214 | 0.295 | 0.202 | 0.292 |
| | 720 | **0.210** | **0.301** | 0.235 | 0.310 | 0.211 | 0.306 | 0.225 | 0.317 | 0.225 | 0.310 | 0.242 | 0.319 | 0.213 | 0.304 | 0.255 | 0.327 | 0.237 | 0.325 |
| | Avg. | **0.170** | **0.264** | 0.194 | 0.275 | 0.171 | 0.270 | 0.178 | 0.270 | 0.182 | 0.272 | 0.201 | 0.285 | 0.175 | 0.269 | 0.217 | 0.295 | 0.202 | 0.290 |
| Solar-Energy | 96 | **0.184** | **0.217** | - | - | 0.215 | 0.295 | 0.203 | 0.237 | 0.189 | 0.241 | 0.224 | 0.264 | 0.200 | 0.275 | 0.328 | 0.396 | 0.252 | 0.319 |
| | 192 | **0.220** | **0.242** | - | - | 0.236 | 0.301 | 0.233 | 0.261 | 0.222 | 0.283 | 0.259 | 0.284 | 0.226 | 0.259 | 0.397 | 0.387 | 0.283 | 0.338 |
| | 336 | 0.247 | **0.266** | - | - | 0.252 | 0.307 | 0.248 | 0.273 | 0.231 | 0.292 | 0.284 | 0.298 | 0.254 | 0.277 | 0.433 | 0.410 | 0.299 | 0.344 |
| | 720 | 0.257 | **0.270** | - | - | 0.244 | 0.305 | 0.249 | 0.275 | **0.223** | 0.285 | 0.284 | 0.298 | 0.249 | 0.284 | 0.429 | 0.396 | 0.298 | 0.351 |
| | Avg. | 0.227 | **0.249** | - | - | 0.237 | 0.302 | 0.233 | 0.262 | **0.216** | 0.280 | 0.263 | 0.286 | 0.232 | 0.274 | 0.397 | 0.398 | 0.283 | 0.338 |

Table 12: Full results of long-term forecasting with a 96-step lookback window (Part II). The input sequence length $L$ is set to 96 for all baselines. All results are averaged across four different forecasting horizon: $T \in \{96, 192, 336, 720\}$. The best and second-best results are highlighted in **bold** and underlined, respectively.

| Models | DMANet Ours | | TimePro 2025 | | TimeKAN 2025 | | SOFTS 2024 | | FreDF 2025 | | PatchTST 2023 | | TimesNet 2023 | | DLinear 2023 | | MICN 2023 | |
|---|---|---|---|---|---|---|---|---|---|---|---|---|---|---|---|---|---|---|
| Metric | MSE | MAE | MSE | MAE | MSE | MAE | MSE | MAE | MSE | MAE | MSE | MAE | MSE | MAE | MSE | MAE | MSE | MAE |
| **ETTm1** 96 | **0.308** | **0.343** | 0.326 | 0.364 | 0.322 | 0.361 | 0.325 | 0.361 | 0.324 | 0.362 | 0.329 | 0.367 | 0.338 | 0.375 | 0.346 | 0.374 | 0.365 | 0.387 |
| 192 | **0.354** | **0.372** | 0.367 | 0.383 | 0.357 | 0.383 | 0.375 | 0.389 | 0.373 | 0.385 | 0.367 | 0.385 | 0.374 | 0.387 | 0.382 | 0.391 | 0.403 | 0.408 |
| 336 | **0.384** | **0.394** | 0.402 | 0.409 | **0.382** | 0.401 | 0.405 | 0.412 | 0.402 | 0.404 | 0.399 | 0.410 | 0.410 | 0.411 | 0.415 | 0.415 | 0.436 | 0.431 |
| 720 | 0.447 | **0.431** | 0.469 | 0.446 | **0.445** | 0.435 | 0.466 | 0.447 | 0.469 | 0.444 | 0.454 | 0.439 | 0.478 | 0.450 | 0.473 | 0.451 | 0.489 | 0.462 |
| Avg. | **0.373** | **0.385** | 0.391 | 0.400 | 0.376 | 0.395 | 0.393 | 0.403 | 0.392 | 0.399 | 0.415 | 0.400 | 0.400 | 0.406 | 0.404 | 0.408 | 0.423 | 0.422 |
| **ETTm2** 96 | **0.165** | **0.244** | 0.178 | 0.260 | 0.174 | 0.255 | 0.180 | 0.261 | 0.173 | 0.252 | 0.175 | 0.259 | 0.187 | 0.267 | 0.193 | 0.293 | 0.197 | 0.296 |
| 192 | **0.231** | **0.288** | 0.242 | 0.303 | 0.239 | 0.299 | 0.246 | 0.306 | 0.241 | 0.298 | 0.241 | 0.302 | 0.249 | 0.309 | 0.284 | 0.361 | 0.284 | 0.361 |
| 336 | **0.289** | **0.325** | 0.303 | 0.342 | 0.301 | 0.340 | 0.319 | 0.352 | 0.298 | 0.334 | 0.305 | 0.343 | 0.321 | 0.351 | 0.382 | 0.429 | 0.381 | 0.429 |
| 720 | **0.385** | **0.383** | 0.400 | 0.399 | 0.395 | 0.396 | 0.405 | 0.401 | 0.398 | 0.393 | 0.402 | 0.400 | 0.408 | 0.403 | 0.558 | 0.525 | 0.549 | 0.522 |
| Avg. | **0.268** | **0.310** | 0.281 | 0.326 | 0.277 | 0.322 | 0.287 | 0.330 | 0.278 | 0.319 | 0.281 | 0.326 | 0.291 | 0.333 | 0.354 | 0.402 | 0.305 | 0.349 |
| **ETTh1** 96 | 0.370 | **0.391** | 0.375 | 0.398 | **0.367** | 0.395 | 0.381 | 0.399 | 0.382 | 0.400 | 0.414 | 0.419 | 0.384 | 0.402 | 0.397 | 0.412 | 0.426 | 0.446 |
| 192 | 0.417 | **0.420** | 0.427 | 0.429 | **0.414** | **0.420** | 0.435 | 0.431 | 0.430 | 0.427 | 0.460 | 0.445 | 0.436 | 0.429 | 0.446 | 0.441 | 0.454 | 0.464 |
| 336 | 0.457 | 0.440 | 0.472 | 0.450 | **0.445** | **0.434** | 0.480 | 0.452 | 0.474 | 0.451 | 0.501 | 0.466 | 0.491 | 0.469 | 0.489 | 0.467 | 0.493 | 0.487 |
| 720 | 0.468 | 0.465 | 0.476 | 0.474 | **0.444** | **0.459** | 0.499 | 0.488 | 0.463 | 0.462 | 0.500 | 0.488 | 0.521 | 0.500 | 0.513 | 0.510 | 0.526 | 0.526 |
| Avg. | 0.428 | 0.429 | 0.438 | 0.438 | **0.417** | **0.427** | 0.449 | 0.442 | 0.437 | 0.435 | 0.469 | 0.454 | 0.458 | 0.450 | 0.461 | 0.457 | 0.475 | 0.480 |
| **ETTh2** 96 | **0.280** | **0.329** | 0.293 | 0.345 | 0.290 | 0.340 | 0.297 | 0.347 | 0.289 | 0.337 | 0.302 | 0.348 | 0.340 | 0.374 | 0.340 | 0.394 | 0.372 | 0.424 |
| 192 | **0.349** | **0.374** | 0.367 | 0.394 | 0.375 | 0.392 | 0.373 | 0.394 | 0.363 | 0.385 | 0.388 | 0.400 | 0.402 | 0.414 | 0.482 | 0.479 | 0.492 | 0.492 |
| 336 | **0.393** | **0.410** | 0.419 | 0.431 | 0.423 | 0.435 | 0.410 | 0.426 | 0.419 | 0.426 | 0.426 | 0.433 | 0.452 | 0.452 | 0.591 | 0.541 | 0.607 | 0.555 |
| 720 | 0.418 | 0.437 | 0.427 | 0.445 | 0.443 | 0.449 | **0.411** | **0.433** | 0.415 | 0.437 | 0.431 | 0.446 | 0.462 | 0.468 | 0.839 | 0.661 | 0.824 | 0.655 |
| Avg. | **0.361** | **0.388** | 0.377 | 0.403 | 0.383 | 0.404 | 0.373 | 0.400 | 0.371 | 0.396 | 0.387 | 0.407 | 0.414 | 0.427 | 0.563 | 0.519 | 0.574 | 0.531 |
| **Weather** 96 | **0.148** | **0.191** | 0.166 | 0.207 | 0.162 | 0.208 | 0.166 | 0.208 | 0.164 | 0.202 | 0.177 | 0.218 | 0.172 | 0.220 | 0.195 | 0.252 | 0.198 | 0.261 |
| 192 | **0.199** | **0.238** | 0.216 | 0.254 | 0.207 | 0.249 | 0.217 | 0.253 | 0.220 | 0.253 | 0.225 | 0.259 | 0.219 | 0.261 | 0.237 | 0.295 | 0.239 | 0.299 |
| 336 | **0.256** | **0.282** | 0.273 | 0.296 | 0.263 | 0.290 | 0.282 | 0.300 | 0.275 | 0.294 | 0.278 | 0.297 | 0.280 | 0.306 | 0.282 | 0.331 | 0.285 | 0.336 |
| 720 | 0.339 | **0.336** | 0.351 | 0.346 | **0.338** | 0.340 | 0.356 | 0.351 | 0.356 | 0.347 | 0.354 | 0.348 | 0.365 | 0.359 | 0.345 | 0.382 | 0.351 | 0.388 |
| Avg. | **0.236** | **0.262** | 0.251 | 0.276 | 0.242 | 0.272 | 0.255 | 0.278 | 0.254 | 0.274 | 0.259 | 0.281 | 0.259 | 0.287 | 0.265 | 0.315 | 0.268 | 0.321 |
| **Electricity** 96 | **0.139** | 0.234 | **0.139** | 0.234 | 0.174 | 0.266 | 0.143 | **0.233** | 0.144 | 0.233 | 0.195 | 0.285 | 0.168 | 0.272 | 0.210 | 0.302 | 0.180 | 0.293 |
| 192 | 0.157 | 0.250 | **0.156** | 0.249 | 0.182 | 0.273 | 0.158 | 0.248 | 0.159 | **0.247** | 0.199 | 0.289 | 0.184 | 0.289 | 0.210 | 0.305 | 0.189 | 0.302 |
| 336 | 0.175 | 0.269 | **0.172** | 0.267 | 0.197 | 0.286 | 0.178 | 0.269 | 0.172 | **0.263** | 0.215 | 0.305 | 0.198 | 0.300 | 0.223 | 0.319 | 0.198 | 0.312 |
| 720 | 0.210 | 0.301 | 0.209 | **0.299** | 0.236 | 0.320 | 0.218 | 0.305 | **0.204** | 0.294 | 0.256 | 0.337 | 0.220 | 0.320 | 0.258 | 0.350 | 0.217 | 0.330 |
| Avg. | 0.170 | 0.264 | **0.169** | 0.262 | 0.197 | 0.286 | 0.174 | 0.264 | 0.170 | **0.259** | 0.216 | 0.304 | 0.193 | 0.295 | 0.225 | 0.319 | 0.196 | 0.309 |
| **Solar-Energy** 96 | **0.184** | **0.217** | 0.196 | 0.237 | 0.254 | 0.318 | 0.200 | 0.230 | 0.232 | 0.256 | 0.234 | 0.286 | 0.250 | 0.292 | 0.290 | 0.378 | 0.257 | 0.325 |
| 192 | **0.220** | **0.242** | 0.231 | 0.263 | 0.285 | 0.326 | 0.229 | 0.253 | 0.276 | 0.288 | 0.267 | 0.310 | 0.296 | 0.318 | 0.320 | 0.398 | 0.278 | 0.354 |
| 336 | 0.247 | **0.266** | 0.250 | 0.281 | 0.315 | 0.338 | **0.243** | 0.269 | 0.301 | 0.306 | 0.290 | 0.315 | 0.319 | 0.330 | 0.353 | 0.415 | 0.298 | 0.375 |
| 720 | 0.257 | **0.270** | 0.253 | 0.285 | 0.313 | 0.340 | **0.245** | 0.272 | 0.308 | 0.316 | 0.289 | 0.317 | 0.338 | 0.337 | 0.357 | 0.413 | 0.299 | 0.379 |
| Avg. | **0.227** | **0.249** | 0.232 | 0.266 | 0.292 | 0.331 | 0.229 | 0.256 | 0.279 | 0.292 | 0.270 | 0.307 | 0.301 | 0.319 | 0.330 | 0.401 | 0.283 | 0.358 |

Table 13: Full results of long-term forecasting with a 720-step lookback window (Part I) The input length $L$ is fixed 720 for optimal horizon in the scaling law of TSF Shi et al. (2024). All results are averaged across four different forecasting horizon: $T \in \{96, 192, 336, 720\}$. The best and second-best results are highlighted in **bold** and underlined, respectively.

| Models | DMANet Ours | | PDF 2024 | | iTransformer 2024 | | Pathformer 2024b | | FITS 2024 | | TimeMixer 2024a | | PatchTST 2023 | | Crossformer 2022 | | TimesNet 2023 | | Dlinear 2023 | | Stationary 2022b | |
|---|---|---|---|---|---|---|---|---|---|---|---|---|---|---|---|---|---|---|---|---|---|---|
| Metric | MSE | MAE | MSE | MAE | MSE | MAE | MSE | MAE | MSE | MAE | MSE | MAE | MSE | MAE | MSE | MAE | MSE | MAE | MSE | MAE | MSE | MAE |
| ETTm1 96 | 0.287 | 0.340 | 0.286 | 0.340 | 0.300 | 0.353 | 0.290 | 0.335 | 0.303 | 0.345 | 0.293 | 0.345 | 0.289 | 0.343 | 0.314 | 0.367 | 0.340 | 0.378 | 0.300 | 0.345 | 0.415 | 0.410 |
| ETTm1 192 | 0.322 | 0.364 | 0.321 | 0.364 | 0.341 | 0.380 | 0.337 | 0.363 | 0.337 | 0.365 | 0.335 | 0.372 | 0.329 | 0.368 | 0.374 | 0.410 | 0.392 | 0.404 | 0.336 | 0.366 | 0.494 | 0.451 |
| ETTm1 336 | 0.352 | 0.381 | 0.354 | 0.383 | 0.374 | 0.396 | 0.374 | 0.384 | 0.368 | 0.384 | 0.368 | 0.386 | 0.362 | 0.390 | 0.413 | 0.432 | 0.423 | 0.426 | 0.367 | 0.386 | 0.577 | 0.490 |
| ETTm1 720 | 0.403 | 0.410 | 0.408 | 0.415 | 0.429 | 0.430 | 0.428 | 0.416 | 0.420 | 0.413 | 0.426 | 0.417 | 0.416 | 0.423 | 0.753 | 0.613 | 0.475 | 0.453 | 0.419 | 0.416 | 0.636 | 0.535 |
| ETTm1 Avg. | 0.341 | 0.374 | 0.342 | 0.376 | 0.361 | 0.390 | 0.357 | 0.375 | 0.357 | 0.377 | 0.356 | 0.380 | 0.349 | 0.381 | 0.464 | 0.456 | 0.408 | 0.415 | 0.356 | 0.378 | 0.531 | 0.472 |
| ETTm2 96 | 0.158 | 0.246 | 0.163 | 0.251 | 0.175 | 0.266 | 0.164 | 0.250 | 0.165 | 0.254 | 0.165 | 0.256 | 0.165 | 0.255 | 0.296 | 0.391 | 0.189 | 0.265 | 0.164 | 0.255 | 0.210 | 0.294 |
| ETTm2 192 | 0.214 | 0.287 | 0.219 | 0.290 | 0.242 | 0.312 | 0.219 | 0.288 | 0.219 | 0.291 | 0.225 | 0.298 | 0.221 | 0.293 | 0.369 | 0.416 | 0.254 | 0.310 | 0.224 | 0.304 | 0.338 | 0.373 |
| ETTm2 336 | 0.264 | 0.320 | 0.269 | 0.330 | 0.282 | 0.337 | 0.267 | 0.319 | 0.272 | 0.326 | 0.277 | 0.332 | 0.276 | 0.327 | 0.588 | 0.600 | 0.313 | 0.345 | 0.277 | 0.337 | 0.432 | 0.416 |
| ETTm2 720 | 0.345 | 0.373 | 0.349 | 0.382 | 0.375 | 0.394 | 0.361 | 0.377 | 0.359 | 0.381 | 0.360 | 0.387 | 0.362 | 0.381 | 0.750 | 0.612 | 0.413 | 0.402 | 0.371 | 0.401 | 0.554 | 0.476 |
| ETTm2 Avg. | 0.245 | 0.307 | 0.250 | 0.313 | 0.269 | 0.327 | 0.253 | 0.308 | 0.254 | 0.313 | 0.257 | 0.318 | 0.256 | 0.314 | 0.501 | 0.505 | 0.292 | 0.331 | 0.259 | 0.324 | 0.383 | 0.390 |
| Weather 96 | 0.141 | 0.188 | 0.147 | 0.196 | 0.157 | 0.207 | 0.148 | 0.195 | 0.172 | 0.225 | 0.147 | 0.198 | 0.149 | 0.196 | 0.143 | 0.210 | 0.168 | 0.214 | 0.170 | 0.230 | 0.188 | 0.242 |
| Weather 192 | 0.189 | 0.237 | 0.193 | 0.240 | 0.200 | 0.248 | 0.191 | 0.235 | 0.215 | 0.260 | 0.192 | 0.243 | 0.191 | 0.239 | 0.198 | 0.260 | 0.219 | 0.262 | 0.216 | 0.273 | 0.241 | 0.290 |
| Weather 336 | 0.239 | 0.275 | 0.245 | 0.280 | 0.252 | 0.287 | 0.243 | 0.274 | 0.261 | 0.295 | 0.247 | 0.284 | 0.242 | 0.279 | 0.258 | 0.314 | 0.278 | 0.302 | 0.258 | 0.307 | 0.341 | 0.341 |
| Weather 720 | 0.303 | 0.327 | 0.323 | 0.334 | 0.320 | 0.336 | 0.318 | 0.326 | 0.326 | 0.341 | 0.318 | 0.330 | 0.312 | 0.330 | 0.335 | 0.385 | 0.353 | 0.351 | 0.323 | 0.362 | 0.403 | 0.388 |
| Weather Avg. | 0.218 | 0.257 | 0.227 | 0.263 | 0.232 | 0.270 | 0.225 | 0.257 | 0.244 | 0.280 | 0.226 | 0.264 | 0.224 | 0.261 | 0.234 | 0.292 | 0.255 | 0.282 | 0.242 | 0.293 | 0.293 | 0.315 |
| Electricity 96 | 0.130 | 0.227 | 0.128 | 0.222 | 0.134 | 0.230 | 0.135 | 0.222 | 0.139 | 0.237 | 0.153 | 0.256 | 0.143 | 0.247 | 0.134 | 0.231 | 0.169 | 0.271 | 0.140 | 0.237 | 0.171 | 0.274 |
| Electricity 192 | 0.145 | 0.242 | 0.147 | 0.242 | 0.154 | 0.250 | 0.157 | 0.253 | 0.154 | 0.250 | 0.168 | 0.269 | 0.158 | 0.260 | 0.146 | 0.243 | 0.180 | 0.280 | 0.154 | 0.251 | 0.180 | 0.283 |
| Electricity 336 | 0.160 | 0.258 | 0.165 | 0.260 | 0.169 | 0.265 | 0.170 | 0.267 | 0.170 | 0.268 | 0.189 | 0.291 | 0.168 | 0.267 | 0.165 | 0.264 | 0.204 | 0.304 | 0.169 | 0.268 | 0.204 | 0.305 |
| Electricity 720 | 0.182 | 0.280 | 0.199 | 0.289 | 0.194 | 0.288 | 0.211 | 0.302 | 0.212 | 0.304 | 0.228 | 0.320 | 0.214 | 0.307 | 0.237 | 0.314 | 0.205 | 0.304 | 0.204 | 0.301 | 0.221 | 0.319 |
| Electricity Avg. | 0.154 | 0.252 | 0.160 | 0.253 | 0.163 | 0.258 | 0.168 | 0.261 | 0.169 | 0.265 | 0.184 | 0.284 | 0.171 | 0.270 | 0.171 | 0.263 | 0.190 | 0.290 | 0.167 | 0.264 | 0.194 | 0.295 |
| Solar 96 | 0.159 | 0.205 | 0.181 | 0.247 | 0.190 | 0.244 | 0.218 | 0.235 | 0.208 | 0.255 | 0.179 | 0.232 | 0.170 | 0.234 | 0.183 | 0.208 | 0.198 | 0.270 | 0.199 | 0.265 | 0.381 | 0.398 |
| Solar 192 | 0.183 | 0.230 | 0.200 | 0.259 | 0.193 | 0.257 | 0.196 | 0.220 | 0.229 | 0.267 | 0.201 | 0.259 | 0.204 | 0.302 | 0.208 | 0.226 | 0.206 | 0.276 | 0.220 | 0.282 | 0.395 | 0.386 |
| Solar 336 | 0.195 | 0.243 | 0.208 | 0.269 | 0.203 | 0.266 | 0.195 | 0.220 | 0.241 | 0.273 | 0.190 | 0.256 | 0.212 | 0.293 | 0.212 | 0.239 | 0.208 | 0.284 | 0.234 | 0.295 | 0.410 | 0.394 |
| Solar 720 | 0.193 | 0.244 | 0.212 | 0.275 | 0.223 | 0.281 | 0.208 | 0.237 | 0.248 | 0.277 | 0.203 | 0.261 | 0.215 | 0.307 | 0.215 | 0.256 | 0.232 | 0.294 | 0.243 | 0.301 | 0.377 | 0.376 |
| Solar Avg. | 0.183 | 0.231 | 0.200 | 0.263 | 0.202 | 0.262 | 0.204 | 0.228 | 0.232 | 0.268 | 0.193 | 0.252 | 0.200 | 0.284 | 0.205 | 0.232 | 0.211 | 0.281 | 0.224 | 0.286 | 0.391 | 0.389 |

Table 14: Full results of long-term forecasting with a 720-step lookback window (Part II).The input length $L$ is fixed 720 for optimal horizon in the scaling law of TSF Shi et al. (2024). All results are averaged across four different forecasting horizon: $T \in \{96, 192, 336, 720\}$. The best and second-best results are highlighted in **bold** and underlined, respectively.

| Models | DMANet (Ours) | | TVNet (2025) | | RLinear (2023a) | | MTS-Mixer (2023c) | | MICN (2023) | | ModernTCN (2024) | | FEDformer (2022b) | | RAFT (2025) | | TSLANet (2024) | | GPT4TS (2023) | | Time-LLM (2024) | |
|---|---|---|---|---|---|---|---|---|---|---|---|---|---|---|---|---|---|---|---|---|---|---|
| Metric | MSE | MAE | MSE | MAE | MSE | MAE | MSE | MAE | MSE | MAE | MSE | MAE | MSE | MAE | MSE | MAE | MSE | MAE | MSE | MAE | MSE | MAE |
| ETTm1 96 | 0.287 | 0.340 | 0.288 | 0.343 | 0.301 | 0.342 | 0.314 | 0.358 | 0.314 | 0.360 | 0.292 | 0.346 | 0.326 | 0.390 | 0.302 | 0.349 | 0.289 | 0.349 | 0.292 | 0.346 | 0.272 | 0.334 |
| ETTm1 192 | 0.322 | 0.364 | 0.326 | 0.367 | 0.355 | 0.363 | 0.354 | 0.386 | 0.359 | 0.387 | 0.332 | 0.368 | 0.365 | 0.415 | 0.329 | 0.367 | 0.328 | 0.370 | 0.332 | 0.372 | 0.310 | 0.358 |
| ETTm1 336 | 0.352 | 0.381 | 0.365 | 0.391 | 0.370 | 0.383 | 0.384 | 0.405 | 0.398 | 0.413 | 0.365 | 0.391 | 0.392 | 0.425 | 0.355 | 0.383 | 0.355 | 0.389 | 0.366 | 0.394 | 0.352 | 0.384 |
| ETTm1 720 | 0.403 | 0.410 | 0.412 | 0.413 | 0.425 | 0.414 | 0.427 | 0.432 | 0.459 | 0.464 | 0.416 | 0.417 | 0.446 | 0.458 | 0.406 | 0.413 | 0.421 | 0.425 | 0.417 | 0.421 | 0.383 | 0.411 |
| ETTm1 Avg. | 0.341 | 0.374 | 0.348 | 0.379 | 0.358 | 0.376 | 0.370 | 0.395 | 0.383 | 0.406 | 0.351 | 0.381 | 0.382 | 0.422 | 0.348 | 0.378 | 0.348 | 0.383 | 0.348 | 0.383 | 0.329 | 0.372 |
| ETTm2 96 | 0.158 | 0.246 | 0.161 | 0.254 | 0.164 | 0.253 | 0.177 | 0.259 | 0.167 | 0.260 | 0.166 | 0.256 | 0.180 | 0.271 | 0.164 | 0.256 | 0.169 | 0.259 | 0.173 | 0.262 | 0.161 | 0.253 |
| ETTm2 192 | 0.214 | 0.287 | 0.220 | 0.293 | 0.219 | 0.290 | 0.241 | 0.303 | 0.245 | 0.316 | 0.222 | 0.293 | 0.252 | 0.318 | 0.219 | 0.296 | 0.224 | 0.297 | 0.229 | 0.301 | 0.219 | 0.293 |
| ETTm2 336 | 0.264 | 0.320 | 0.272 | 0.316 | 0.273 | 0.326 | 0.297 | 0.338 | 0.295 | 0.324 | 0.272 | 0.324 | 0.324 | 0.364 | 0.275 | 0.336 | 0.275 | 0.329 | 0.286 | 0.341 | 0.271 | 0.329 |
| ETTm2 720 | 0.345 | 0.373 | 0.349 | 0.379 | 0.366 | 0.385 | 0.396 | 0.398 | 0.389 | 0.406 | 0.351 | 0.381 | 0.410 | 0.420 | 0.359 | 0.392 | 0.354 | 0.380 | 0.378 | 0.401 | 0.352 | 0.379 |
| ETTm2 Avg. | 0.245 | 0.307 | 0.251 | 0.311 | 0.256 | 0.314 | 0.277 | 0.325 | 0.277 | 0.336 | 0.253 | 0.314 | 0.292 | 0.343 | 0.254 | 0.320 | 0.256 | 0.316 | 0.226 | 0.326 | 0.251 | 0.313 |
| Weather 96 | 0.141 | 0.188 | 0.147 | 0.198 | 0.175 | 0.225 | 0.156 | 0.206 | 0.161 | 0.226 | 0.149 | 0.200 | 0.238 | 0.314 | 0.165 | 0.222 | 0.148 | 0.197 | 0.162 | 0.212 | 0.147 | 0.201 |
| Weather 192 | 0.189 | 0.237 | 0.194 | 0.238 | 0.218 | 0.260 | 0.199 | 0.248 | 0.220 | 0.283 | 0.196 | 0.245 | 0.275 | 0.329 | 0.211 | 0.264 | 0.193 | 0.241 | 0.204 | 0.248 | 0.189 | 0.235 |
| Weather 336 | 0.239 | 0.275 | 0.235 | 0.277 | 0.265 | 0.294 | 0.249 | 0.291 | 0.275 | 0.328 | 0.238 | 0.277 | 0.339 | 0.377 | 0.260 | 0.302 | 0.245 | 0.282 | 0.254 | 0.286 | 0.262 | 0.279 |
| Weather 720 | 0.303 | 0.327 | 0.308 | 0.331 | 0.329 | 0.339 | 0.336 | 0.343 | 0.311 | 0.356 | 0.314 | 0.334 | 0.389 | 0.409 | 0.327 | 0.355 | 0.325 | 0.337 | 0.326 | 0.337 | 0.304 | 0.316 |
| Weather Avg. | 0.218 | 0.257 | 0.221 | 0.261 | 0.247 | 0.279 | 0.235 | 0.272 | 0.242 | 0.298 | 0.224 | 0.264 | 0.310 | 0.357 | 0.241 | 0.286 | 0.325 | 0.337 | 0.237 | 0.270 | 0.225 | 0.257 |
| Electricity 96 | 0.130 | 0.227 | 0.142 | 0.223 | 0.140 | 0.235 | 0.141 | 0.243 | 0.159 | 0.267 | 0.129 | 0.226 | 0.186 | 0.302 | 0.133 | 0.232 | 0.136 | 0.229 | 0.139 | 0.238 | 0.131 | 0.224 |
| Electricity 192 | 0.145 | 0.242 | 0.165 | 0.241 | 0.154 | 0.248 | 0.163 | 0.261 | 0.168 | 0.279 | 0.143 | 0.239 | 0.197 | 0.311 | 0.149 | 0.247 | 0.152 | 0.244 | 0.153 | 0.251 | 0.160 | 0.248 |
| Electricity 336 | 0.160 | 0.258 | 0.164 | 0.269 | 0.171 | 0.264 | 0.176 | 0.277 | 0.196 | 0.308 | 0.161 | 0.259 | 0.213 | 0.328 | 0.161 | 0.259 | 0.168 | 0.262 | 0.169 | 0.266 | 0.160 | 0.248 |
| Electricity 720 | 0.182 | 0.280 | 0.190 | 0.284 | 0.209 | 0.297 | 0.212 | 0.308 | 0.203 | 0.312 | 0.191 | 0.286 | 0.233 | 0.344 | 0.197 | 0.297 | 0.205 | 0.293 | 0.206 | 0.297 | 0.192 | 0.298 |
| Electricity Avg. | 0.154 | 0.252 | 0.165 | 0.254 | 0.169 | 0.261 | 0.173 | 0.272 | 0.182 | 0.292 | 0.156 | 0.253 | 0.207 | 0.321 | 0.160 | 0.259 | 0.165 | 0.257 | 0.167 | 0.263 | 0.158 | 0.252 |

Table 15: Full results of long-term forecasting with a 96-step lookback window (Part III). The input sequence length $L$ is set to 96 for all baselines. All results are averaged across four different forecasting horizon: $T \in \{96, 192, 336, 720\}$. The best and second-best results are highlighted in **bold** and underlined, respectively.

| Models | | DMANet | | iTransformer | | Fredformer | | TimeMixer | | PatchTST | | Crossformer | | TimesNet | | TIDE | | DLinear | | FreTS | | FEDformer | |
|---|---|---|---|---|---|---|---|---|---|---|---|---|---|---|---|---|---|---|---|---|---|---|---|
| Metric | | MSE | MAE | MSE | MAE | MSE | MAE | MSE | MAE | MSE | MAE | MSE | MAE | MSE | MAE | MSE | MAE | MSE | MAE | MSE | MAE | MSE | MAE |
| PEMS03 | 12 | **0.064** | **0.167** | 0.071 | 0.174 | 0.068 | 0.174 | 0.076 | 0.188 | 0.099 | 0.216 | 0.090 | 0.203 | 0.085 | 0.192 | 0.178 | 0.305 | 0.122 | 0.243 | 0.083 | 0.194 | 0.126 | 0.251 |
| | 24 | **0.086** | **0.193** | 0.093 | 0.201 | 0.094 | 0.205 | 0.113 | 0.226 | 0.142 | 0.259 | 0.121 | 0.240 | 0.118 | 0.223 | 0.257 | 0.371 | 0.201 | 0.317 | 0.127 | 0.198 | 0.241 | 0.275 |
| | 48 | 0.132 | 0.239 | **0.125** | **0.236** | 0.152 | 0.262 | 0.191 | 0.292 | 0.211 | 0.319 | 0.202 | 0.317 | 0.155 | 0.260 | 0.379 | 0.463 | 0.333 | 0.425 | 0.202 | 0.310 | 0.227 | 0.348 |
| | *Avg.* | **0.094** | **0.200** | 0.096 | 0.204 | 0.105 | 0.214 | 0.127 | 0.235 | 0.151 | 0.265 | 0.138 | 0.253 | 0.119 | 0.271 | 0.271 | 0.380 | 0.219 | 0.295 | 0.137 | 0.234 | 0.167 | 0.291 |
| PEMS04 | 12 | **0.069** | **0.168** | 0.078 | 0.183 | 0.085 | 0.189 | 0.092 | 0.204 | 0.105 | 0.224 | 0.098 | 0.218 | 0.087 | 0.195 | 0.219 | 0.340 | 0.148 | 0.272 | 0.097 | 0.209 | 0.138 | 0.262 |
| | 24 | **0.082** | **0.185** | 0.095 | 0.205 | 0.117 | 0.224 | 0.128 | 0.243 | 0.153 | 0.257 | 0.131 | 0.256 | 0.103 | 0.215 | 0.292 | 0.398 | 0.224 | 0.340 | 0.144 | 0.258 | 0.177 | 0.293 |
| | 48 | **0.107** | **0.216** | 0.120 | 0.233 | 0.174 | 0.276 | 0.213 | 0.315 | 0.229 | 0.339 | 0.205 | 0.326 | 0.136 | 0.250 | 0.409 | 0.478 | 0.335 | 0.437 | 0.223 | 0.328 | 0.270 | 0.368 |
| | *Avg.* | **0.086** | **0.190** | 0.098 | 0.207 | 0.125 | 0.215 | 0.144 | 0.254 | 0.162 | 0.273 | 0.145 | 0.267 | 0.109 | 0.220 | 0.307 | 0.405 | 0.236 | 0.350 | 0.148 | 0.265 | 0.195 | 0.308 |
| PEMS07 | 12 | **0.057** | **0.152** | 0.067 | 0.165 | 0.063 | 0.158 | 0.073 | 0.184 | 0.095 | 0.207 | 0.094 | 0.200 | 0.082 | 0.181 | 0.173 | 0.304 | 0.115 | 0.242 | 0.078 | 0.185 | 0.109 | 0.225 |
| | 24 | **0.074** | **0.174** | 0.088 | 0.190 | 0.089 | 0.192 | 0.111 | 0.219 | 0.150 | 0.262 | 0.139 | 0.247 | 0.101 | 0.204 | 0.271 | 0.383 | 0.210 | 0.329 | 0.127 | 0.239 | 0.125 | 0.244 |
| | 48 | **0.109** | **0.211** | 0.110 | 0.215 | 0.136 | 0.241 | 0.237 | 0.328 | 0.253 | 0.340 | 0.311 | 0.369 | 0.134 | 0.238 | 0.446 | 0.495 | 0.398 | 0.458 | 0.220 | 0.317 | 0.165 | 0.288 |
| | *Avg.* | **0.080** | **0.179** | 0.088 | 0.190 | 0.096 | 0.197 | 0.140 | 0.244 | 0.166 | 0.270 | 0.181 | 0.272 | 0.106 | 0.208 | 0.297 | 0.394 | 0.241 | 0.343 | 0.142 | 0.247 | 0.133 | 0.282 |
| PEMS08 | 12 | **0.066** | **0.167** | 0.079 | 0.182 | 0.081 | 0.185 | 0.091 | 0.201 | 0.168 | 0.232 | 0.165 | 0.214 | 0.112 | 0.212 | 0.227 | 0.343 | 0.154 | 0.276 | 0.096 | 0.204 | 0.173 | 0.273 |
| | 24 | **0.085** | **0.192** | 0.115 | 0.219 | 0.112 | 0.214 | 0.137 | 0.246 | 0.224 | 0.281 | 0.215 | 0.260 | 0.141 | 0.238 | 0.318 | 0.409 | 0.248 | 0.353 | 0.152 | 0.256 | 0.210 | 0.310 |
| | 48 | **0.121** | **0.235** | 0.186 | 0.235 | 0.174 | 0.267 | 0.265 | 0.343 | 0.321 | 0.354 | 0.315 | 0.335 | 0.198 | 0.283 | 0.497 | 0.510 | 0.440 | 0.470 | 0.247 | 0.331 | 0.320 | 0.394 |
| | *Avg.* | **0.090** | **0.198** | 0.127 | 0.212 | 0.122 | 0.222 | 0.164 | 0.263 | 0.238 | 0.289 | 0.232 | 0.270 | 0.150 | 0.244 | 0.347 | 0.421 | 0.281 | 0.366 | 0.165 | 0.264 | 0.234 | 0.326 |

Table 16: Univariate long-term forecasting results on ETT datasets. Following PatchTST and ModernTCN, input length is fixed as 336 and prediction lengths are $T \in \{96, 192, 336, 720\}$. The best and second-best results are highlighted in **bold** and underlined, respectively.

| Models | | DMANet | | ModernTCN | | iTransformer | | TimeMixer | | PatchTST | | DLinear | | Pyraformer | | FEDformer | | Autoformer | |
|---|---|---|---|---|---|---|---|---|---|---|---|---|---|---|---|---|---|---|---|---|
| Metric | | MSE | MAE | MSE | MAE | MSE | MAE | MSE | MAE | MSE | MAE | MSE | MAE | MSE | MAE | MSE | MAE | MSE | MAE |
| ETTm1 | 96 | **0.026** | 0.122 | **0.026** | **0.121** | 0.029 | 0.127 | 0.029 | 0.128 | 0.029 | 0.126 | 0.028 | 0.123 | 0.127 | 0.281 | 0.033 | 0.140 | 0.056 | 0.183 |
| | 192 | **0.039** | **0.150** | 0.040 | 0.152 | 0.045 | 0.162 | 0.044 | 0.160 | 0.043 | 0.158 | 0.045 | 0.156 | 0.205 | 0.343 | 0.058 | 0.186 | 0.081 | 0.216 |
| | 336 | **0.052** | **0.172** | 0.053 | 0.173 | 0.059 | 0.189 | 0.058 | 0.185 | 0.056 | 0.183 | 0.061 | 0.182 | 0.302 | 0.457 | 0.084 | 0.231 | 0.076 | 0.218 |
| | 720 | **0.072** | **0.203** | 0.073 | 0.206 | 0.080 | 0.218 | 0.081 | 0.218 | 0.080 | 0.217 | 0.080 | 0.210 | 0.387 | 0.485 | 0.102 | 0.250 | 0.110 | 0.267 |
| | *Avg.* | **0.047** | **0.162** | 0.048 | 0.163 | 0.053 | 0.174 | 0.053 | 0.173 | 0.052 | 0.171 | 0.054 | 0.168 | 0.255 | 0.392 | 0.069 | 0.202 | 0.081 | 0.221 |
| ETTm2 | 96 | **0.063** | **0.182** | 0.065 | 0.183 | 0.071 | 0.193 | 0.068 | 0.187 | 0.071 | 0.192 | **0.063** | 0.183 | 0.074 | 0.208 | 0.067 | 0.198 | 0.065 | 0.189 |
| | 192 | 0.093 | 0.228 | 0.095 | 0.232 | 0.109 | 0.248 | 0.101 | 0.236 | 0.102 | 0.237 | **0.092** | **0.227** | 0.116 | 0.252 | 0.102 | 0.245 | 0.118 | 0.256 |
| | 336 | **0.117** | **0.260** | 0.119 | 0.261 | 0.141 | 0.289 | 0.133 | 0.278 | 0.130 | 0.274 | 0.119 | 0.261 | 0.143 | 0.295 | 0.130 | 0.279 | 0.154 | 0.305 |
| | 720 | **0.167** | **0.317** | 0.173 | 0.323 | 0.190 | 0.343 | 0.183 | 0.332 | 0.179 | 0.328 | 0.175 | 0.320 | 0.197 | 0.338 | 0.178 | 0.325 | 0.182 | 0.335 |
| | *Avg.* | **0.110** | **0.247** | 0.113 | 0.250 | 0.128 | 0.268 | 0.121 | 0.258 | 0.121 | 0.258 | 0.112 | 0.248 | 0.133 | 0.273 | 0.119 | 0.262 | 0.130 | 0.271 |
| ETTh1 | 96 | **0.054** | **0.176** | 0.055 | 0.179 | 0.059 | 0.185 | 0.057 | 0.181 | 0.056 | 0.181 | 0.056 | 0.180 | 0.099 | 0.277 | 0.079 | 0.215 | 0.071 | 0.206 |
| | 192 | **0.066** | **0.200** | 0.070 | 0.205 | 0.073 | 0.208 | 0.072 | 0.204 | 0.076 | 0.210 | 0.071 | 0.204 | 0.174 | 0.346 | 0.104 | 0.245 | 0.114 | 0.262 |
| | 336 | **0.073** | 0.215 | 0.074 | **0.214** | 0.084 | 0.223 | 0.085 | 0.227 | 0.094 | 0.242 | 0.098 | 0.244 | 0.198 | 0.370 | 0.119 | 0.270 | 0.107 | 0.258 |
| | 720 | **0.082** | **0.227** | 0.086 | 0.232 | 0.089 | 0.236 | 0.083 | 0.227 | 0.101 | 0.250 | 0.189 | 0.359 | 0.209 | 0.348 | 0.142 | 0.299 | 0.126 | 0.283 |
| | *Avg.* | **0.069** | **0.205** | 0.071 | 0.206 | 0.076 | 0.213 | 0.074 | 0.210 | 0.082 | 0.221 | 0.104 | 0.247 | 0.170 | 0.335 | 0.111 | 0.257 | 0.105 | 0.252 |
| ETTh2 | 96 | **0.121** | **0.269** | 0.124 | 0.274 | 0.136 | 0.287 | 0.133 | 0.283 | 0.130 | 0.276 | 0.131 | 0.279 | 0.152 | 0.303 | 0.128 | 0.271 | 0.153 | 0.306 |
| | 192 | **0.154** | **0.310** | 0.164 | 0.321 | 0.187 | 0.342 | 0.190 | 0.341 | 0.181 | 0.331 | 0.176 | 0.329 | 0.197 | 0.370 | 0.185 | 0.330 | 0.204 | 0.351 |
| | 336 | 0.174 | **0.336** | **0.171** | **0.336** | 0.219 | 0.374 | 0.226 | 0.379 | 0.226 | 0.379 | 0.209 | 0.367 | 0.238 | 0.385 | 0.231 | 0.378 | 0.246 | 0.389 |
| | 720 | **0.211** | **0.371** | 0.228 | 0.384 | 0.253 | 0.403 | 0.241 | 0.396 | 0.253 | 0.406 | 0.276 | 0.426 | 0.274 | 0.435 | 0.278 | 0.420 | 0.268 | 0.409 |
| | *Avg.* | **0.165** | **0.322** | 0.172 | 0.329 | 0.199 | 0.352 | 0.198 | 0.350 | 0.198 | 0.348 | 0.198 | 0.350 | 0.215 | 0.373 | 0.206 | 0.350 | 0.218 | 0.364 |

Table 17: Full results of short-term forecasting on supplementary datasets from domains including Health & Medical (ILI, COVID-19), Web Events (Wiki, Website), Finance (NASDAQ, SP500, DowJones), Market (CarSales), Energy (Power), and Society (Unemp). The best and second-best results are highlighted in **bold** and underlined, respectively.

| Model | DMANet | | TimeMixer | | FilterNet | | FITS | | DLinear | | Fredformer | | PatchTST | |
|---|---|---|---|---|---|---|---|---|---|---|---|---|---|---|
| Metric | MSE | MAE | MSE | MAE | MSE | MAE | MSE | MAE | MSE | MAE | MSE | MAE | MSE | MAE |
| ILI 24 | **1.746** | **0.813** | 2.110 | 0.879 | 2.190 | 0.870 | 4.265 | 1.523 | 3.158 | 1.243 | 2.098 | 0.894 | 2.046 | 0.849 |
| ILI 36 | 1.718 | 0.817 | 2.084 | 0.890 | 1.902 | 0.862 | 3.718 | 1.363 | 3.009 | 1.200 | **1.712** | 0.867 | 2.344 | 0.912 |
| ILI 48 | **1.744** | **0.826** | 1.961 | 0.866 | 2.051 | 0.882 | 3.994 | 1.422 | 2.994 | 1.194 | 2.054 | 0.922 | 2.123 | 0.883 |
| ILI 60 | **1.842** | **0.839** | 1.926 | 0.878 | 2.151 | 0.925 | 4.543 | 1.554 | 3.172 | 1.232 | 1.925 | 0.913 | 2.001 | 0.895 |
| ILI Avg | **1.763** | **0.824** | 2.020 | 0.878 | 2.073 | 0.885 | 4.130 | 1.465 | 3.083 | 1.217 | 1.947 | 0.899 | 2.128 | 0.885 |
| Covid19 3 | **1.098** | **0.489** | 1.237 | 0.547 | 1.195 | 0.555 | 2.039 | 0.790 | 2.386 | 0.909 | 1.165 | 0.548 | 1.220 | 0.573 |
| Covid19 6 | 1.735 | **0.625** | 2.003 | 0.739 | 1.839 | 0.711 | 2.683 | 0.919 | 3.220 | 1.053 | **1.465** | 0.685 | 1.982 | 0.762 |
| Covid19 9 | **2.167** | **0.722** | 2.594 | 0.860 | 2.537 | 0.897 | 3.147 | 1.050 | 3.803 | 1.160 | 2.145 | 0.845 | 2.633 | 0.916 |
| Covid19 12 | **2.640** | **0.843** | 3.103 | 0.981 | 2.782 | 0.956 | 3.630 | 1.156 | 4.524 | 1.288 | 2.833 | 0.984 | 3.050 | 1.030 |
| Covid19 Avg | 1.910 | **0.670** | 2.234 | 0.782 | 2.088 | 0.780 | 2.875 | 0.979 | 3.483 | 1.102 | **1.902** | 0.765 | 2.221 | 0.820 |
| NASDAQ 24 | **0.118** | **0.214** | 0.122 | 0.221 | 0.130 | 0.230 | 0.140 | 0.244 | 0.155 | 0.274 | 0.128 | 0.226 | 0.127 | 0.224 |
| NASDAQ 36 | **0.158** | **0.260** | 0.183 | 0.279 | 0.175 | 0.273 | 0.184 | 0.284 | 0.196 | 0.306 | 0.170 | 0.268 | 0.174 | 0.269 |
| NASDAQ 48 | **0.200** | **0.296** | 0.200 | 0.298 | 0.224 | 0.314 | 0.234 | 0.324 | 0.244 | 0.344 | 0.218 | 0.306 | 0.225 | 0.314 |
| NASDAQ 60 | **0.233** | **0.323** | 0.238 | 0.328 | 0.259 | 0.340 | 0.282 | 0.357 | 0.318 | 0.401 | 0.262 | 0.339 | 0.265 | 0.339 |
| NASDAQ Avg | **0.177** | **0.273** | 0.186 | 0.281 | 0.197 | 0.289 | 0.210 | 0.302 | 0.228 | 0.331 | 0.194 | 0.285 | 0.198 | 0.286 |
| Wiki 3 | 6.116 | 0.372 | 6.209 | 0.392 | 6.234 | 0.402 | 7.470 | 0.496 | 6.254 | 0.438 | 6.190 | 0.387 | **6.112** | **0.380** |
| Wiki 6 | **6.419** | **0.388** | 6.475 | 0.402 | 6.460 | 0.401 | 8.326 | 0.544 | 6.579 | 0.467 | 6.696 | 0.404 | 6.425 | 0.395 |
| Wiki 9 | **6.665** | **0.402** | 6.702 | 0.418 | 6.697 | 0.416 | 8.869 | 0.564 | 6.776 | 0.508 | 6.768 | 0.411 | 6.743 | 0.426 |
| Wiki 12 | 6.824 | 0.411 | 6.902 | 0.426 | 6.899 | 0.426 | 9.394 | 0.608 | 6.927 | 0.513 | 7.168 | 0.424 | **6.814** | 0.414 |
| Wiki Avg | **6.506** | **0.393** | 6.572 | 0.409 | 6.572 | 0.411 | 8.515 | 0.553 | 6.634 | 0.481 | 6.705 | 0.406 | 6.523 | 0.404 |
| SP500 24 | **0.153** | **0.271** | 0.159 | 0.288 | 0.181 | 0.317 | 0.193 | 0.334 | 0.189 | 0.330 | 0.181 | 0.315 | 0.164 | 0.298 |
| SP500 36 | **0.205** | **0.315** | 0.218 | 0.343 | 0.224 | 0.341 | 0.259 | 0.389 | 0.250 | 0.363 | 0.239 | 0.365 | 0.221 | 0.341 |
| SP500 48 | **0.250** | **0.348** | 0.264 | 0.367 | 0.280 | 0.384 | 0.324 | 0.439 | 0.291 | 0.398 | 0.283 | 0.394 | 0.278 | 0.397 |
| SP500 60 | **0.293** | **0.383** | 0.322 | 0.416 | 0.332 | 0.416 | 0.391 | 0.486 | 0.377 | 0.475 | 0.341 | 0.438 | 0.321 | 0.409 |
| SP500 Avg | **0.225** | **0.329** | 0.241 | 0.353 | 0.254 | 0.365 | 0.291 | 0.412 | 0.277 | 0.391 | 0.261 | 0.378 | 0.246 | 0.361 |
| DowJones 24 | **7.325** | **0.666** | 8.327 | 0.683 | 8.000 | 0.683 | 7.974 | 0.690 | 7.590 | 0.670 | 7.758 | 0.672 | 7.641 | 0.670 |
| DowJones 36 | **10.422** | **0.800** | 11.192 | 0.813 | 12.011 | 0.823 | 11.907 | 0.837 | 10.986 | 0.803 | 11.456 | 0.808 | 11.210 | 0.807 |
| DowJones 48 | **13.975** | **0.917** | 15.278 | 0.945 | 14.814 | 0.933 | 15.821 | 0.969 | 14.157 | 0.922 | 14.696 | 0.921 | 14.866 | 0.935 |
| DowJones 60 | **16.106** | **1.016** | 20.997 | 1.067 | 18.932 | 1.054 | 19.320 | 1.077 | 18.018 | 1.035 | 18.058 | 1.032 | 17.947 | 1.036 |
| DowJones Avg | **11.957** | **0.850** | 13.948 | 0.877 | 13.439 | 0.873 | 13.755 | 0.893 | 12.688 | 0.857 | 12.992 | 0.858 | 12.916 | 0.862 |
| CarSales 24 | **0.318** | **0.314** | 0.320 | 0.318 | **0.318** | 0.319 | 0.359 | 0.347 | 0.354 | 0.350 | 0.319 | 0.326 | 0.319 | 0.319 |
| CarSales 36 | 0.332 | **0.327** | 0.332 | 0.331 | **0.331** | 0.330 | 0.373 | 0.360 | 0.368 | 0.365 | 0.333 | 0.335 | 0.332 | 0.330 |
| CarSales 48 | 0.346 | 0.340 | 0.345 | 0.343 | **0.342** | **0.341** | 0.385 | 0.370 | 0.382 | 0.379 | 0.349 | 0.344 | 0.347 | 0.344 |
| CarSales 60 | 0.357 | 0.352 | **0.355** | **0.351** | 0.352 | 0.349 | 0.399 | 0.385 | 0.388 | 0.380 | 0.359 | 0.349 | **0.355** | 0.348 |
| CarSales Avg | 0.338 | **0.333** | 0.338 | 0.336 | **0.336** | 0.335 | 0.379 | 0.365 | 0.373 | 0.368 | 0.340 | 0.338 | 0.338 | 0.335 |
| Power 24 | **1.293** | **0.865** | 1.341 | 0.881 | 1.410 | 0.916 | 1.491 | 0.944 | 1.390 | 0.916 | 1.410 | 0.913 | 1.468 | 0.935 |
| Power 36 | **1.334** | **0.875** | 1.420 | 0.914 | 1.590 | 0.968 | 1.621 | 0.994 | 1.518 | 0.957 | 1.538 | 0.953 | 1.593 | 0.972 |
| Power 48 | **1.408** | **0.917** | 1.567 | 0.963 | 1.680 | 1.009 | 1.775 | 1.052 | 1.610 | 0.995 | 1.652 | 1.008 | 1.710 | 1.020 |
| Power 60 | **1.456** | **0.940** | 1.609 | 0.988 | 1.776 | 1.053 | 1.958 | 1.122 | 1.679 | 1.020 | 1.752 | 1.049 | 1.829 | 1.064 |
| Power Avg | **1.373** | **0.899** | 1.484 | 0.937 | 1.614 | 0.986 | 1.711 | 1.028 | 1.549 | 0.972 | 1.588 | 0.981 | 1.650 | 0.998 |
| Website 3 | 0.083 | 0.209 | 0.086 | 0.215 | 0.084 | 0.213 | 0.191 | 0.320 | 0.159 | 0.288 | 0.080 | **0.207** | 0.089 | 0.217 |
| Website 6 | **0.112** | **0.238** | 0.124 | 0.248 | 0.116 | 0.242 | 0.235 | 0.356 | 0.182 | 0.302 | 0.116 | 0.241 | 0.121 | 0.246 |
| Website 9 | 0.154 | **0.265** | 0.159 | 0.275 | 0.151 | 0.269 | 0.276 | 0.372 | 0.220 | 0.330 | 0.150 | 0.268 | 0.157 | 0.273 |
| Website 12 | 0.199 | **0.294** | 0.204 | 0.306 | 0.194 | 0.297 | 0.409 | 0.484 | 0.255 | 0.355 | 0.196 | 0.301 | 0.200 | 0.302 |
| Website Avg | 0.137 | **0.252** | 0.143 | 0.261 | 0.136 | 0.255 | 0.278 | 0.383 | 0.204 | 0.319 | **0.135** | 0.254 | 0.141 | 0.259 |
| Unemp 3 | **0.010** | **0.052** | 0.015 | 0.074 | 0.012 | 0.062 | 0.161 | 0.289 | 0.072 | 0.200 | 0.013 | 0.068 | 0.012 | 0.060 |
| Unemp 6 | **0.036** | **0.117** | 0.057 | 0.154 | 0.043 | 0.130 | 0.229 | 0.345 | 0.115 | 0.255 | 0.046 | 0.139 | 0.043 | 0.127 |
| Unemp 9 | **0.081** | **0.180** | 0.109 | 0.213 | 0.107 | 0.216 | 0.369 | 0.443 | 0.191 | 0.329 | 0.095 | 0.198 | 0.093 | 0.190 |
| Unemp 12 | **0.127** | **0.234** | 0.195 | 0.293 | 0.155 | 0.255 | 0.475 | 0.500 | 0.240 | 0.386 | 0.148 | 0.250 | 0.164 | 0.261 |
| Unemp Avg | **0.064** | **0.146** | 0.094 | 0.183 | 0.079 | 0.166 | 0.308 | 0.394 | 0.154 | 0.292 | 0.075 | 0.163 | 0.078 | 0.160 |

## D.5 RESULTS FOR HYPERPARAMETER ANALYSIS

In this section, we also explore the effect of the hyperparameters used in our experiments, including the depth-wise convolution kernal size $K$, the depth-wise convolution kernal stride size $s$, the channel change $c$ and $\lambda$ on the loss function.

The channel change $c$ signifies the alteration in the number of channels during the downsampling process, where values below 1 denote a reduction in channel quantity, whereas values exceeding 1 indicate channel expansion.

For $\lambda$, according to FreDF Wang et al. (2025), the loss function is a weighted sum of the time-domain MSE and the frequency-domain MAE. $\lambda$ represents the proportion of the frequency MAE in the loss function, and $(1 - \lambda)$ represents the proportion of the time-domain MSE.

We show the experimental results from Table.18 to Table.21.

Table 18: Impact of kernal size. A lower MSE or MAE indicates a better performance.

| Models | Metrics | Weather | | ETTh2 | | ETTm2 | |
|---|---|---|---|---|---|---|---|
| | | 96 | 336 | 96 | 336 | 96 | 336 |
| $K = 1$ | MSE | 0.151 | 0.274 | 0.289 | 0.393 | 0.169 | 0.289 |
| | MAE | 0.194 | 0.325 | 0.334 | 0.411 | 0.248 | 0.326 |
| $K = 3$ | MSE | 0.148 | 0.256 | 0.280 | 0.393 | 0.165 | 0.289 |
| | MAE | 0.191 | 0.282 | 0.329 | 0.410 | 0.244 | 0.325 |
| $K = 5$ | MSE | 0.149 | 0.259 | 0.286 | 0.393 | 0.168 | 0.296 |
| | MAE | 0.192 | 0.285 | 0.332 | 0.410 | 0.247 | 0.331 |
| $K = 7$ | MSE | 0.150 | 0.258 | 0.286 | 0.394 | 0.167 | 0.292 |
| | MAE | 0.193 | 0.283 | 0.331 | 0.411 | 0.245 | 0.327 |

Table 19: Impact of stride size. A lower MSE or MAE indicates a better performance.

| Models | Metrics | Weather | | ETTh2 | | ETTm2 | |
|---|---|---|---|---|---|---|---|
| | | 96 | 336 | 96 | 336 | 96 | 336 |
| $s = 1$ | MSE | 0.151 | 0.259 | 0.281 | 0.423 | 0.170 | 0.297 |
| | MAE | 0.194 | 0.284 | 0.331 | 0.421 | 0.247 | 0.331 |
| $s = 2$ | MSE | 0.148 | 0.256 | 0.280 | 0.393 | 0.165 | 0.289 |
| | MAE | 0.191 | 0.282 | 0.329 | 0.410 | 0.244 | 0.325 |
| $s = 3$ | MSE | 0.149 | 0.259 | 0.274 | 0.393 | 0.167 | 0.290 |
| | MAE | 0.192 | 0.284 | 0.325 | 0.410 | 0.246 | 0.326 |
| $s = 4$ | MSE | 0.148 | 0.257 | 0.279 | 0.395 | 0.167 | 0.291 |
| | MAE | 0.190 | 0.282 | 0.326 | 0.411 | 0.246 | 0.327 |

Table 20: Impact of channel change. A lower MSE or MAE indicates a better performance.

| Models | Metrics | Weather | | ETTh2 | | ETTm2 | |
|--------|---------|---------|-----|-------|-----|-------|-----|
| | | 96 | 336 | 96 | 336 | 96 | 336 |
| $c = 0.25$ | MSE | 0.149 | 0.259 | 0.278 | 0.396 | 0.167 | 0.291 |
| | MAE | 0.193 | 0.284 | 0.326 | 0.412 | 0.245 | 0.327 |
| $c = 0.5$ | MSE | 0.148 | 0.256 | 0.280 | 0.393 | 0.165 | 0.289 |
| | MAE | 0.191 | 0.282 | 0.329 | 0.410 | 0.244 | 0.325 |
| $c = 1$ | MSE | 0.149 | 0.261 | 0.284 | 0.408 | 0.171 | 0.293 |
| | MAE | 0.191 | 0.287 | 0.333 | 0.415 | 0.250 | 0.328 |
| $c = 2$ | MSE | 0.149 | 0.257 | 0.281 | 0.401 | 0.169 | 0.294 |
| | MAE | 0.192 | 0.283 | 0.332 | 0.415 | 0.247 | 0.327 |
| $c = 4$ | MSE | 0.152 | 0.261 | 0.292 | 0.398 | 0.173 | 0.293 |
| | MAE | 0.196 | 0.287 | 0.337 | 0.415 | 0.250 | 0.328 |

Table 21: Impact of $\lambda$ in loss. A lower MSE or MAE indicates a better performance.

| Models | Metrics | Weather | | ETTh2 | | ETTm2 | |
|--------|---------|---------|-----|-------|-----|-------|-----|
| | | 96 | 336 | 96 | 336 | 96 | 336 |
| $\lambda = 0.1$ | MSE | 0.149 | 0.257 | 0.289 | 0.392 | 0.168 | 0.297 |
| | MAE | 0.191 | 0.283 | 0.333 | 0.412 | 0.246 | 0.332 |
| $\lambda = 0.3$ | MSE | 0.149 | 0.259 | 0.289 | 0.394 | 0.167 | 0.297 |
| | MAE | 0.192 | 0.284 | 0.332 | 0.411 | 0.246 | 0.332 |
| $\lambda = 0.5$ | MSE | 0.149 | 0.259 | 0.286 | 0.394 | 0.169 | 0.298 |
| | MAE | 0.191 | 0.284 | 0.331 | 0.412 | 0.246 | 0.332 |
| $\lambda = 0.7$ | MSE | 0.149 | 0.259 | 0.290 | 0.396 | 0.169 | 0.297 |
| | MAE | 0.191 | 0.283 | 0.332 | 0.414 | 0.247 | 0.332 |
| $\lambda = 1$ | MSE | 0.148 | 0.256 | 0.280 | 0.393 | 0.165 | 0.289 |
| | MAE | 0.191 | 0.282 | 0.329 | 0.410 | 0.244 | 0.325 |

## E   MORE DETAILS OF COMPUTATIONAL COSTS

To comprehensively evaluate the efficiency and scalability of DMANet, we conducted controlled experiments on both synthetic and real-world datasets. Our analysis focuses on two key aspects: the computational overhead of our proposed components and the overall model's performance compared to state-of-the-art methods.

### E.1   EFFICIENCY AND SCALABILITY ANALYSIS ON SYNTHETIC DATA

We first use synthetic data to perform a fine-grained analysis under controlled conditions, isolating the impact of sequence length and channel dimensions. With fixed hyperparameters (look-back window=96, batch size=64, etc.), we measure inference speed (ms) and peak GPU memory (MB) under two scenarios: (1) fixing the number of channels $C$ while varying the sequence length $T$, and (2) fixing $T$ while varying $C$. Each experiment was repeated 500 times for stability. The results are presented in Table.22.

Table 22: Inference Speed (ms) and Memory Usage (MB) Comparison Across Different Models and Configurations. Values for speed are reported as mean $\pm$ std over 500 runs.

| Configuration | DMANet | | w/o-ESR | | Chebyshev | |
|---|---|---|---|---|---|---|
| | Speed | Memory | Speed | Memory | Speed | Memory |
| $T = 256$ | 1.376±0.122 | 15.71 | 1.309±0.359 | 15.71 | 1.916±0.150 | 15.71 |
| $T = 512$ | 1.372±0.107 | 31.29 | 1.325±0.115 | 31.29 | 1.942±0.173 | 31.29 |
| $T = 1024$ | 1.677±0.137 | 65.38 | 1.600±0.346 | 65.38 | 2.249±0.182 | 65.38 |
| $T = 2048$ | 2.915±0.151 | 146.49 | 2.846±0.256 | 146.49 | 3.576±0.019 | 146.49 |
| $C = 48$ | 1.518±0.103 | 61.36 | 1.444±0.222 | 61.36 | 1.523±0.206 | 61.36 |
| $C = 96$ | 1.982±0.141 | 122.40 | 1.882±0.154 | 122.40 | 2.640±0.187 | 122.40 |
| $C = 192$ | 3.731±0.047 | 251.77 | 3.580±0.072 | 251.77 | 4.250±0.182 | 251.77 |
| $C = 336$ | 7.029±0.025 | 471.27 | 6.760±0.043 | 471.27 | 7.399±0.167 | 471.27 |

| Configuration | Linear | | TransConv | | Attention | |
|---|---|---|---|---|---|---|
| | Speed | Memory | Speed | Memory | Speed | Memory |
| $T = 256$ | 1.228±0.352 | 15.89 | 1.405±0.128 | 15.92 | 3.065±0.234 | 75.04 |
| $T = 512$ | 1.201±0.158 | 32.02 | 1.605±0.124 | 31.68 | 3.412±0.239 | 280.09 |
| $T = 1024$ | 1.838±0.196 | 68.39 | 2.191±0.168 | 66.70 | 8.710±0.117 | 1084.94 |
| $T = 2048$ | 5.079±0.117 | 158.52 | 3.818±0.089 | 147.73 | 29.417±0.224 | 4270.15 |
| $C = 48$ | 1.414±0.225 | 61.43 | 1.915±0.161 | 61.33 | 3.892±0.186 | 298.42 |
| $C = 96$ | 1.794±0.078 | 122.31 | 2.432±0.303 | 119.46 | 5.278±1.450 | 332.12 |
| $C = 192$ | 3.374±0.054 | 252.63 | 4.418±0.281 | 236.45 | 8.956±0.035 | 404.04 |
| $C = 336$ | 5.642±0.150 | 471.27 | 8.116±0.019 | 414.28 | 15.971±0.102 | 518.38 |

From these results, we draw two key conclusions:

**1. The computational overhead of our dynamic anti-aliasing (ESR Filter) is negligible.**   A direct comparison between DMANet and its ablated version (w/o ESR) reveals that the peak memory usage is nearly identical across all configurations. The time overhead introduced by the ESR filter is minimal, with a worst-case relative increase of only 2.4% (at T=2048). Furthermore, DMANet is consistently faster than the variant using a classical Chebyshev filter. This empirically proves that our dynamic anti-aliasing mechanism is a computationally lightweight strategy that does not introduce a performance bottleneck.

**2. The efficiency and scalability of frequency-domain interpolation for upsampling.**   We further validate our choice of upsampling mechanism by comparing it with common alternatives. The Attention-based method is not viable for long sequences due to the explosive, quadratic growth in its memory and time costs. While a simple Linear layer is fast, it scales poorly when processing very long sequences (e.g., at T=2048, its speed degrades significantly). Although Transposed Con-

volution is lightweight, our method is faster in most scenarios. In conclusion, our chosen frequency-domain interpolation achieves an excellent balance of cost-effectiveness and scalability across different data shapes.

## E.2 EFFICIENCY COMPARISON WITH STATE-OF-THE-ART MODELS ON REAL-WORLD DATASETS

Follow the TimeKAN Huang et al. (2025), we benchmark the overall efficiency of DMANet against leading SOTA models on real-world datasets. We fix the input and prediction lengths ($T = 96, F = 96$) to ensure a fair comparison and report on model parameters (Params), multiply-accumulate operations (MACs) and predictive accuracy.

Table 23: A comparison of model parameters (Params) and multiply-accumulate operations (MACs) for DMANet and seven other models. To ensure a fair comparison, we fix the prediction length $F = 96$ and the input length $T = 96$.

| Model | ETTm2 | | | | Weather | | | | Electricity | | | |
|---|---|---|---|---|---|---|---|---|---|---|---|---|
| | Params | MACs | MSE | MAE | Params | MACs | MSE | MAE | Params | MACs | MSE | MAE |
| iTransformer | 224.22 K | 19.86 M | 0.184 | 0.267 | 4.83 M | 1.16 G | 0.175 | 0.216 | 4.83 M | 16.29 G | 0.148 | 0.240 |
| TimeMixer | 77.77 K | 24.18 M | 0.175 | 0.257 | 104.43 K | 82.62 M | 0.161 | 0.208 | 106.83 K | 1.26 G | 0.156 | 0.247 |
| TimesNet | 1.19 M | 36.28 G | 0.189 | 0.266 | 1.19 M | 36.28 G | 0.169 | 0.219 | 150.30 M | 4.61 T | 0.168 | 0.272 |
| PatchTST | 10.06 M | 17.66 G | 0.183 | 0.268 | 6.90 M | 35.30 G | 0.176 | 0.217 | 6.90 M | 539.68 G | 0.180 | 0.273 |
| DLinear | 18.62 K | 0.60 M | 0.193 | 0.293 | 18.62 K | 0.60 M | 0.196 | 0.256 | 18.62 K | 0.60 M | 0.210 | 0.302 |
| TimeKAN | 38.12 K | 16.66 M | 0.174 | 0.257 | 20.94 K | 29.86 M | 0.163 | 0.208 | 23.34 K | 456.50 M | 0.175 | 0.268 |
| FilterNet | 49.61 K | 1.67 M | 0.175 | 0.257 | 49.64 K | 1.03 M | 0.166 | 0.210 | 50.24 K | 15.78 M | 0.167 | 0.256 |
| **DMANet** | 19.08 K | 92.29 K | 0.173 | 0.253 | 77.02 K | 0.33 M | 0.155 | 0.201 | 8.49 M | 73.95 M | 0.146 | 0.243 |

As shown in Table.23, DMANet demonstrates a state-of-the-art balance between efficiency and performance. Compared to Transformer-based models (e.g., iTransformer, PatchTST) and recent computationally intensive architectures like TimeKAN, DMANet requires less memory and fewer MACs while maintaining superior forecasting accuracy. While simple baselines like DLinear and FilterNet are exceptionally fast, DMANet provides a substantial accuracy improvement with only a marginal increase in computational cost. These results confirm that the lightweight and scalable design choices validated in our synthetic experiments translate directly to a highly competitive efficiency in real-world applications. Notably, on the high-dimensional Electricity dataset, DMANet achieves the best MSE (0.146) with a computational cost of only 73.95M MACs. This is significantly lower than that of PatchTST (539.68G MACs), iTransformer (16.29G MACs), and even TimeKAN (456.5M MACs). These results confirm that DMANet is not only a lightweight solution for simple tasks but also a highly scalable and efficient architecture for complex real-world applications.

## F MORE DETAILS OF PRE-SAMPLING FILTERING

To comprehensively evaluate the robustness of our model and its generalization ability to different types of signal disturbance, we synthesized noise and superimposed it onto the original clean signals $x_{\text{clean}}$ to generate noisy signals $x_{\text{noisy}}$ for model testing. The synthetic noise was generated using a unified framework that supports multiple noise types, with precise control over the intensity of the noise through parameters. Specifically, we implemented the following noise types:

- **Frequency-Domain Noise:** It includes High-frequency noise, Low-frequency noise, and Broadband noise. This type is generated by taking the Fast Fourier Transform (FFT) of the original signal, generating a band-limited or broadband random Gaussian noise spectrum in the frequency domain, and then converting it back to the time domain via Inverse Fast Fourier Transform (IFFT). The frequency band division for high- and low-frequency noise is controlled by the $r_{\text{cut}}$ parameter, defined as the cutoff proportion in the frequency space.
- **Trend Noise:** Simulates slow-varying, non-periodic disturbances. This noise is generated by creating a low-order (e.g., quadratic) polynomial with random coefficients to simulate the trend component in the time series and adding it to the original signal.
- **Seasonal Noise:** Simulates periodic disturbances. This noise is generated by superimposing one or more sine waves with predefined base frequencies specified by the parameter $f_{\text{seasonal}}$, each having a random initial phase.

The noise intensity is precisely controlled by $\epsilon$, which defines the desired ratio of noise energy $E_{\text{noise}}$ to clean signal energy $E_{\text{clean}}$, i.e., $E_{\text{noise}}/E_{\text{clean}}$. After generating the noise, which can be denoted as **noise**, the noise energy is calculated and scaled accordingly to ensure that the noise added to the clean signal has a relative energy level consistent with $\epsilon$. The final noisy signal $x_{\text{noisy}}$ is obtained by adding the scaled noise $\textbf{noise}_{\text{scaled}}$ to the original clean signal: $x_{\text{noisy}} = x_{\text{clean}} + \textbf{noise}_{\text{scaled}}$.

Then, we systematically analyze the performance of the model when faced with various signal distortions. In our experiments, concretely, we fixed the $r_{\text{cut}}$ at 0.3, set $f_{\text{seasonal}}$ to $\{1/24, 1/12\}$, and used $\epsilon$ values of $\{0.1, 0.2, 0.5\}$ in different experimental groups. The results are shown in Table.24.

Comparative Study of Anti-Aliasing Strategies. To further investigate our proposed Equivalent Sampling Rate mechanism and explore efficient anti-aliasing strategies, we conducted a comparative study on the Weather dataset using a 96-step lookback to predict a 720-step horizon. We benchmarked three distinct anti-aliasing configurations:

- **DMANet (ESR-based):** Our proposed model, which uses the architecture-aware ESR to dynamically determine the cutoff frequency for a sharp filter.
- **DMANet_but (Butterworth):** A variant where the ESR-based filter is replaced by a traditional 4th-order Butterworth low-pass filter, a well-established mathematical filter known for its maximally flat passband.
- **DMANet_mix (Fusion-based):** A hybrid model that first uses ESR to partition the spectrum and then processes the high- and low-frequency bands through separate convolutional layers before fusing them, designed to explore the utility of preserved high-frequency information.
- **DMANet_wo (No Filter):** A baseline variant that removes the anti-aliasing filter entirely, processing the raw input directly through the network to assess the necessity and impact of frequency-domain filtering.

The results under various noise conditions are summarized in Table.24. Overall, most of configurations demonstrate notable robustness, with only graceful performance degradation as noise intensity increases. This highlights the general effectiveness of incorporating a pre-sampling filtering stage to enhance noise resistance.

Our ablation study reveals a insight into the effectiveness of different anti-aliasing strategies. Theoretically, one might expect the Butterworth filter (DMANet_but), with its maximally flat passband, to excel at handling low-frequency and trend noise by preserving the signal fidelity in that band Yin et al. (2024). Conversely, our ESR-based hard-cutoff filter (DMANet) should be superior against high-frequency and seasonal noise due to its removal of aliasing-prone components.

Table 24: Robustness analysis of DMANet variants under different types and intensities of synthetic noise on the Weather dataset. All experiments use a 96-step lookback to predict a 720-step horizon.

| Model Variant | Noise Type | $\epsilon = 1\%$ | | $\epsilon = 5\%$ | | $\epsilon = 10\%$ | |
|---|---|---|---|---|---|---|---|
| | | MSE | MAE | MSE | MAE | MSE | MAE |
| DMANet | Seasonal | 0.343 | 0.339 | 0.342 | 0.341 | 0.341 | 0.343 |
| | Trend | 0.344 | 0.340 | 0.345 | 0.345 | 0.351 | 0.358 |
| | All (Broadband) | 0.345 | 0.341 | 0.345 | 0.342 | 0.346 | 0.344 |
| | Low-Frequency | 0.345 | 0.340 | 0.347 | 0.342 | 0.349 | 0.347 |
| | High-Frequency | 0.344 | 0.340 | 0.343 | 0.340 | 0.343 | 0.341 |
| DMANet_but | Seasonal | 0.346 | 0.340 | 0.342 | 0.340 | 0.340 | 0.343 |
| | Trend | 0.347 | 0.342 | 0.348 | 0.347 | 0.354 | 0.359 |
| | All (Broadband) | 0.349 | 0.342 | 0.347 | 0.343 | 0.346 | 0.344 |
| | Low-Frequency | 0.352 | 0.344 | 0.349 | 0.345 | 0.350 | 0.347 |
| | High-Frequency | 0.345 | 0.341 | 0.343 | 0.340 | 0.343 | 0.343 |
| DMANet_mix | Seasonal | 0.353 | 0.343 | 0.350 | 0.345 | 0.350 | 0.348 |
| | Trend | 0.348 | 0.342 | 0.353 | 0.350 | 0.356 | 0.359 |
| | All (Broadband) | 0.352 | 0.344 | 0.353 | 0.344 | 0.357 | 0.348 |
| | Low-Frequency | 0.354 | 0.345 | 0.351 | 0.345 | 0.353 | 0.348 |
| | High-Frequency | 0.358 | 0.346 | 0.353 | 0.345 | 0.352 | 0.347 |
| DMANet_wo | Seasonal | 0.348 | 0.343 | 0.350 | 0.346 | 0.346 | 0.349 |
| | Trend | 0.348 | 0.343 | 0.351 | 0.351 | 0.357 | 0.361 |
| | All (Broadband) | 0.350 | 0.344 | 0.352 | 0.346 | 0.347 | 0.345 |
| | Low-Frequency | 0.355 | 0.346 | 0.355 | 0.349 | 0.353 | 0.349 |
| | High-Frequency | 0.350 | 0.343 | 0.350 | 0.345 | 0.351 | 0.344 |

Interestingly, our empirical results in Table.24 show that while performance is competitive on seasonal and high-frequency noise, DMANet consistently and significantly outperforms DMANet_but on trend and low-frequency noise. This seemingly counter-intuitive result highlights a critical limitation of applying classical filters naively within a deep learning pipeline. While the Butterworth filter is static and optimally preserves its predefined passband, it is architecture-agnostic. It may still pass frequencies that, while low, are too high for the subsequent strided convolution to process without aliasing. In contrast, our ESR-based approach is architecture-aware. It does not aim to be a perfect mathematical filter in isolation; its sole purpose is to perfectly prepare the signal for the next layer. By dynamically calculating a precise cutoff based on the network's own parameters, it ensures that no aliasing occurs at any stage, even if this means a slightly more aggressive filtering. This architectural synergy proves to be more practically effective.

Furthermore, the fusion-based DMANet_mix consistently underperforms the other two variants. This result empirically supports our design rationale for employing a strict cutoff strategy: for a lightweight model, it is more effective to concentrate its limited capacity on core, learnable patterns rather than attempting to fit the complex and often noisy dynamics of high-frequency information. As observed in prior work like FITS Xu et al. (2024), removing a significant portion of high-frequency components largely preserves a time series' dominant trends. The poor performance of DMANet_mix indicates that simply preserving and processing this high-frequency content is less effective than principled filtering, likely because this band is dominated by noise that the model cannot distinguish from a true signal.

Collectively, these results validate that our ESR-based approach provides the most robust and adaptive solution. By dynamically and precisely removing only the frequencies that would cause aliasing, it not only focuses the model on the most decisive, learnable patterns but also achieves this with superior adaptability compared to static classical filters, all without the need for manual filter design.

## G MORE DETAILS OF OUR METHOD

### G.1 THE RATIONALE FOR THE EMBEDDING FIRST ARCHITECTURE

A critical challenge in multi-scale time series analysis is the fusion of features from different scales without introducing signal distortion. A common approach, which we term multi-scale first, involves downsampling the raw signal and then embedding each scale. However, this seemingly intuitive process hides a significant pitfall: the upsampling step required for feature fusion inevitably causes spectral distortion due to its reliance on a limited reconstruction basis. To circumvent this fundamental issue, our DMANet adopts a principled **embedding first** architecture, ensuring all operations are conducted with high fidelity within a unified feature space.

Given this architectural choice, we must clarify the nature of operations performed in this latent space. Although our model operates on embedded latent features, we distinguish our approach from general feature dimension reduction by preserving the sequential topology. While the initial projection transforms the raw time series into a latent dimension, the subsequent incorporation of learnable positional encodings and convolutional inductive biases compels the model to organize these features into a strict sequence with local dependencies, establishing what we term a latent time axis. Unlike standard dimension reduction which treats features as unordered vectors, reducing the resolution of this organized axis via strided operations constitutes mathematical downsampling strictly governed by the Nyquist-Shannon theorem. Disregarding the sampling rate in this context leads to aliasing, where high-frequency latent patterns generate spurious correlations. So we employ the term downsampling to explicitly highlight this critical risk often overlooked in conventional dimension reduction perspectives.

#### G.1.1 THE PITFALL OF PREMATURE MULTI-SCALE DECOMPOSITION

The multi-scale first approach, seen in models like TimeMixer Wang et al. (2024a), begins by decomposing the raw signal $X$ into a set of time series $\{X_m \in \mathbb{R}^{C \times s_m}, s_m < L\}$. While feasible, the core flaw lies in the subsequent step of unifying these scales for feature fusion. To restore the original length $L$, each short sequence $s_m$ must be upsampled using a linear layer, $g_m : \mathbb{R}^{s_m} \to \mathbb{R}^L$. This process is inherently problematic due to its limited representational capacity:

- **Limited Basis Vectors:** The weight matrix $W \in \mathbb{R}^{L \times s_m}$ of the upsampling layer provides only $s_m$ **column vectors**. These vectors form the *entire basis* available to reconstruct the output signal. Consequently, all reconstructed signals are confined to a very small, $s_m$-**dimensional subspace** of the target space $\mathbb{R}^L$.

- **Deformed Basis Vectors:** To approximate the diverse signals in the training data from this constrained basis, the model is forced to learn complex, **non-smooth, and oscillatory** basis vectors as a poor compromise.

- **Inevitable Spectral Distortion:** When a signal is reconstructed as a linear combination of these deformed basis vectors, it unavoidably inherits their unnatural properties. This leads to severe **spectral distortion**, corrupting the signal's fidelity and polluting the final prediction.

#### G.1.2 EMBED FIRST: A PRINCIPLED APPROACH IN A UNIFIED FEATURE SPACE

Our DMANet architecture is designed to completely avoid the aforementioned reconstruction problem by first establishing a unified workspace for all operations.

1. **Defining a Unified Workspace:** We begin by projecting the information-complete raw signal $X \in \mathbb{R}^{C \times L}$ into a new feature basis space using a linear layer. This generates a feature sequence $X' \in \mathbb{R}^{C \times T}$, creating a unified and consistent workspace for all subsequent synergistic operations.

2. **High-Fidelity Operations:** Within this consistent feature space, all core operations are performed in a principled manner:
   - **Downsampling:** Our anti-aliasing downsampling module pre-filters features in the frequency domain before reducing resolution, preventing information aliasing and ensuring reliable feature transfer across scales.

- **Upsampling:** To restore resolution for feature fusion, we employ zero-padding in the frequency domain. This method is equivalent to ideal interpolation and relies on the **Fourier basis (sines and cosines)**—a **fixed, universal, and complete orthogonal basis**. Adhering to the Nyquist-Shannon sampling theorem, this ensures the **smoothest possible reconstruction**, free from the uncontrolled high-frequency artifacts generated by the alternative approach.

By ensuring all features are derived and processed with high-fidelity operations within the same basis space, we maintain inherent consistency and make feature fusion fundamentally more reliable.

### G.1.3 EMPIRICAL VALIDATION

To validate our theoretical analysis, we conducted a comprehensive comparison between our DMANet (Embedding First) and the alternative architecture (Multi-Scale First). We also performed an ablation study by removing the initial embedding module (w/o embed) to verify the effectiveness of operating within a latent space.

The results in Table.25 provide strong empirical support for our design.

Table 25: Comparative analysis and ablation study for the Embedding First architecture. Our full DMANet model is compared against the Multi-Scale First approach and a variant without the initial embedding module.

| Model | Metric | ETTh1 | ETTm1 | Weather | Elect | Wiki | ILI | Unemp | Dowjone |
|---|---|---|---|---|---|---|---|---|---|
| **DMANet (Embedding First)** | MSE | **0.428** | **0.373** | **0.236** | **0.172** | **6.506** | **1.763** | **0.064** | **11.957** |
| | MAE | **0.429** | **0.385** | **0.262** | **0.265** | **0.393** | **0.824** | **0.146** | **0.850** |
| **Multi-Scale First** | MSE | 0.441 | 0.385 | 0.242 | 0.181 | 6.555 | 2.097 | 0.074 | 12.420 |
| | MAE | 0.435 | 0.391 | 0.268 | 0.273 | 0.407 | 0.858 | 0.167 | 0.860 |
| **w/o embed** | MSE | 0.436 | 0.389 | 0.249 | 0.188 | 6.551 | 2.084 | 0.073 | 12.330 |
| | MAE | 0.427 | 0.392 | 0.274 | 0.277 | 0.406 | 0.879 | 0.163 | 0.857 |

The Multi-Scale First approach consistently underperforms our model. This performance gap is a direct, practical consequence of the spectral distortion introduced by its unprincipled, basis-limited reconstruction step.

Furthermore, we acknowledge that the initial linear mapping carries a potential risk of losing some temporal dependencies. This is a deliberate design choice, and its justification is twofold. First, we incorporate a learnable positional encoding to preserve crucial temporal context. Second, as our ablation study will demonstrate, the benefits of analyzing the series in a latent space—where patterns are more suitable for anti-aliasing and feature extraction—outweigh the alternative of operating directly on the raw signal. The placeholder for the w/o embed results in Table.25 will provide strong evidence for this superiority.

## G.2 PRINCIPLED ANTI-ALIASING VIA DYNAMIC FREQUENCY CUTOFF

For any given downsampling layer $l$ in DMANet, its anti-aliasing operation is the application of a low-pass filter with a mathematically-derived, strict cutoff frequency. Given the layer's convolutional parameters—kernel size $k$, stride $s$, and channel ratio $c$—we first calculate its Effective Sampling Ratio ($\mathrm{ESR}^l$) using Equation.3 to determine its true signal processing capability. This allows us to establish a new Nyquist frequency, $f_{\mathrm{Nyquist}}^l$. As shown in Equation.4, all frequency components above this threshold are strictly zeroed out via a frequency-domain mask. Our method does not partially retain or vaguely attenuate high-frequency components; it employs a principled cutoff scheme where the threshold is dynamically determined for each layer.

### G.2.1 THE RATIONALE: FOCUSING ON LEARNABLE CORE PATTERNS

The core motivation for this strict cutoff strategy is to concentrate the model's capacity on learnable, core patterns. High-frequency information in time series often contains significant noise or stochastic fluctuations that are difficult to model, and typically exceed the learning capacity of a lightweight model. Attempting to fit these complex dynamics can hinder the model from capturing the more decisive, underlying trends.

As observed in prior work like FITS Xu et al. (2024), removing a significant portion of high-frequency components largely preserves the overall shape and dominant trends of a time series. Our strategy builds on this insight: by proactively simplifying the learning task, we focus the model's limited capacity on the low-frequency periodic and trend patterns that are most critical for the forecasting task, thereby achieving both efficient and accurate predictions.

### G.2.2 DYNAMIC ADAPTABILITY AND PARAMETER-FREE DESIGN

A key advantage of our method is its dynamic nature. In a complex multi-scale architecture, different layers may employ varying downsampling parameters $(k, s, c)$. Our framework automatically derives a matching, optimal cutoff frequency for each specific layer. This ensures that the anti-aliasing protection remains effective and theoretically grounded across any architectural variation, eliminating the need for tedious, manual parameter tuning required by classical filters or the randomness of heuristic approaches.

Furthermore, this framework possesses theoretical flexibility. By adjusting the convolution parameters, the ESR can be controlled to retain more, or even all, frequency components. For instance, if parameters are set such that ESR $= 1$ (e.g., $s = \min(K, C_{\mathrm{out}}/C_{\mathrm{in}})$), the cutoff frequency matches the original signal's Nyquist frequency, meaning no valid frequency components are attenuated.

### G.2.3 ADDRESSING THE HIGH-FREQUENCY INFORMATION TRADE-OFF

We acknowledge that this design is built upon a core trade-off: we filter high-frequency components to prevent aliasing at the cost of potentially discarding useful information. This is a deliberate choice motivated by the efficiency and robustness goals for a lightweight model.

We also recognize that high-frequency information can be critical in certain scenarios, such as forecasting sharp spikes or in contexts where high-frequency harmonics are themselves key features. It is precisely for this reason that we deliberately conducted extensive supplementary experiments across a diverse range of domains (including Electricity, Weather, Transportation, Health, Web, Market, Energy, Society, Finance, etc.). The goal was to proactively probe the application boundaries of our method and provide a clear reference for its practical use.

To further investigate this trade-off, we will conduct a controlled experiment comparing our strict cutoff method with an alternative that handles frequencies differently. As shown in Table.26, we will compare our standard DMANet against a variant where, after identifying the cutoff frequency, both the low-frequency and the zeroed-out high-frequency components are independently passed through linear layers and then fused. This will help quantify the practical impact of the information contained in the high-frequency bands.

Table 26: Ablation study on the handling of high-frequency components. We compare our strict cutoff method with a variant that uses linear fusion for high and low frequencies.

| Model | Metric | ETTh1 | ETTm1 | Weather | Elect | Wiki | ILI | Unemp | Dowjone |
|---|---|---|---|---|---|---|---|---|---|
| **DMANet (Strict Cutoff)** | MSE | **0.428** | **0.373** | **0.236** | **0.172** | **6.506** | **1.763** | **0.064** | **11.957** |
| | MAE | **0.429** | **0.385** | **0.262** | **0.265** | **0.393** | **0.824** | **0.146** | **0.850** |
| **DMANet_mix (Fusion-based)** | MSE | 0.434 | 0.374 | 0.237 | 0.171 | 6.528 | 1.986 | 0.076 | 12.288 |
| | MAE | 0.433 | 0.385 | 0.263 | 0.265 | 0.398 | 0.867 | 0.161 | 0.858 |

In summary, DMANet's anti-aliasing employs a precise, dynamic cutoff strategy tailored to each layer's actual sampling capability. This design combines theoretical robustness with practical, parameter-free convenience, and its effectiveness is validated through extensive empirical analysis.

## H   MORE DETAILS OF DEPENDENCY MODELING

We visualize the temporal dependencies and channel-wise relationships within a batch of the Electricity in Figure.8 and Figure.9 and for Weather in Figure.10 and Figure.11, comparing their states before and after processing by DMANet's components. To further illustrate the differences between scenarios with and without the anti-aliasing filter, we selected the Electricity dataset to visualize the temporal dependency differences of upsampling before and after applying the anti-aliasing filter in Figure.12, as well as the channel dependency correlation differences of downsampling with and without the anti-aliasing filter in Figure.13.

DMANet tends to leverage more effective dependencies to capture future trends. Comparing the cases with and without the anti-aliasing filter, the figures reveal that the pre-processing anti-aliasing operation, acting as a low-pass filter, smooths or attenuates fine-grained dependencies that are susceptible to aliasing during sampling. This process helps to highlight the main temporal dependency patterns and channel relationships. Furthermore, convolution, leveraging its local receptive field, focuses on local patterns at neighboring time points. Thus, the combination of filtering and convolutional downsampling effectively extracts stable temporal features.

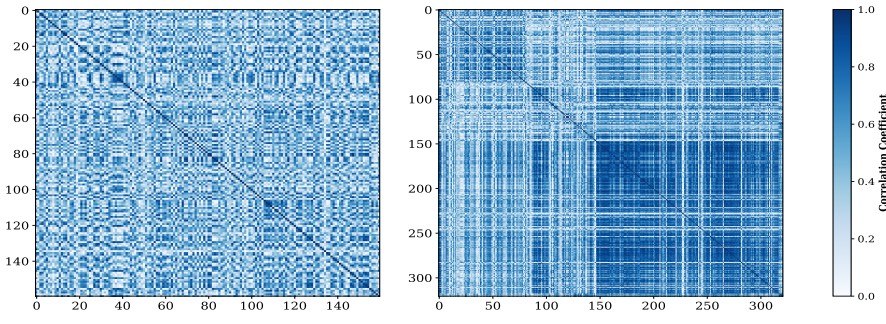

(a) w / Pre-Sampling filtering. **Left:** Downsample corr: 0.432, **Right:** Upsample corr: 0.534.

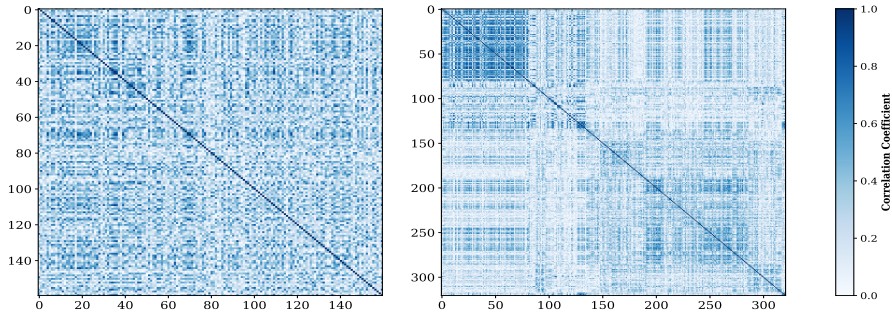

(b) w/o Pre-Sampling filtering. **Left:** Downsample corr: 0.311, **Right:** Upsample corr: 0.254.

Figure 8: Visualization for channel dependency modeling on Electricity in the first layer of the second multiscale encoder block.

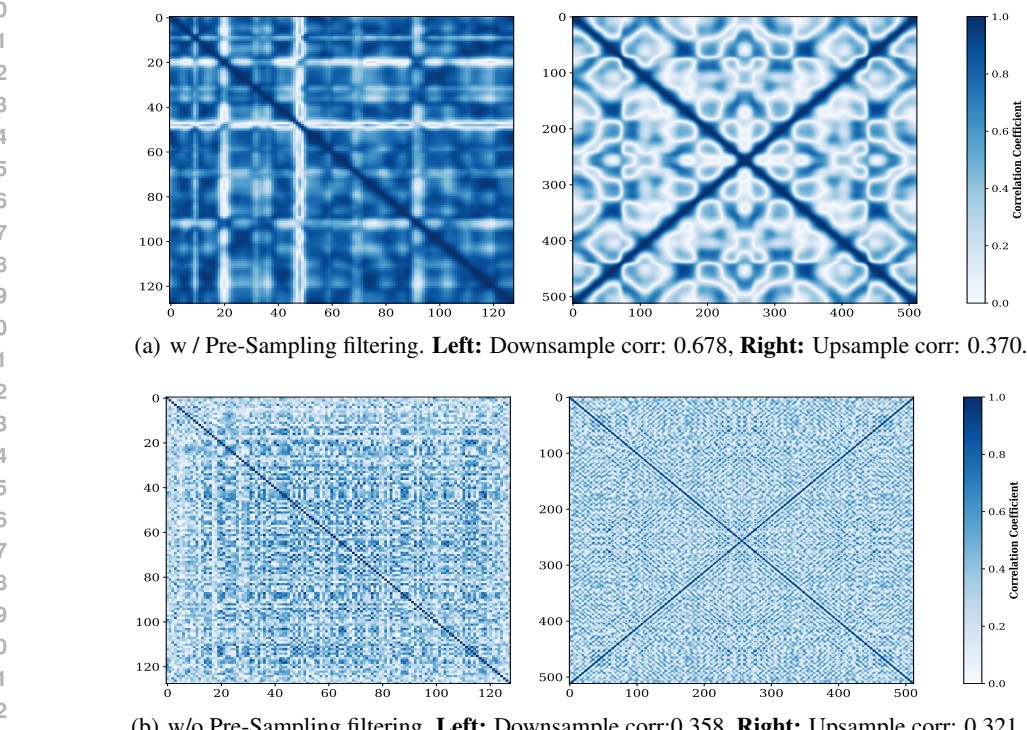

(a) w / Pre-Sampling filtering. **Left:** Downsample corr: 0.678, **Right:** Upsample corr: 0.370.

(b) w/o Pre-Sampling filtering. **Left:** Downsample corr:0.358, **Right:** Upsample corr: 0.321.

Figure 9: Visualization for temporal dependency modeling on Electricity in the first layer of the second multiscale encoder block.

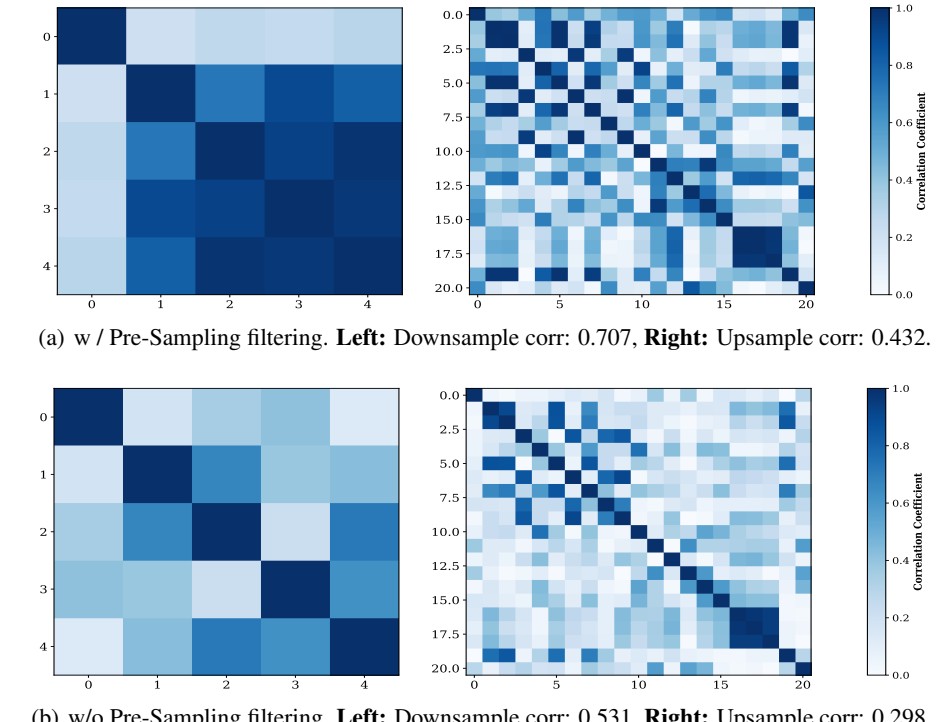

(a) w / Pre-Sampling filtering. **Left:** Downsample corr: 0.707, **Right:** Upsample corr: 0.432.

(b) w/o Pre-Sampling filtering. **Left:** Downsample corr: 0.531, **Right:** Upsample corr: 0.298.

Figure 10: Visualization for channel dependency modeling on Weather in the first layer of the first multiscale encoder block.

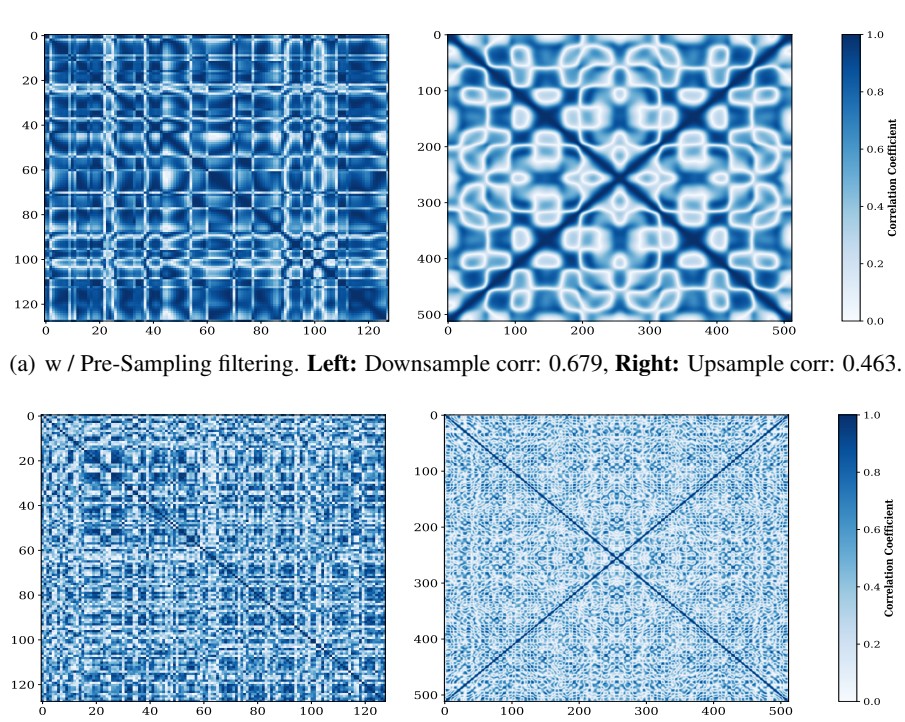

(a) w / Pre-Sampling filtering. **Left:** Downsample corr: 0.679, **Right:** Upsample corr: 0.463.

(b) w/o Pre-Sampling filtering. **Left:** Downsample corr: 0.520, **Right:** Upsample corr: 0.332.

Figure 11: Visualization for temporal dependency modeling on Weather in the first layer of the first multiscale encoder block.

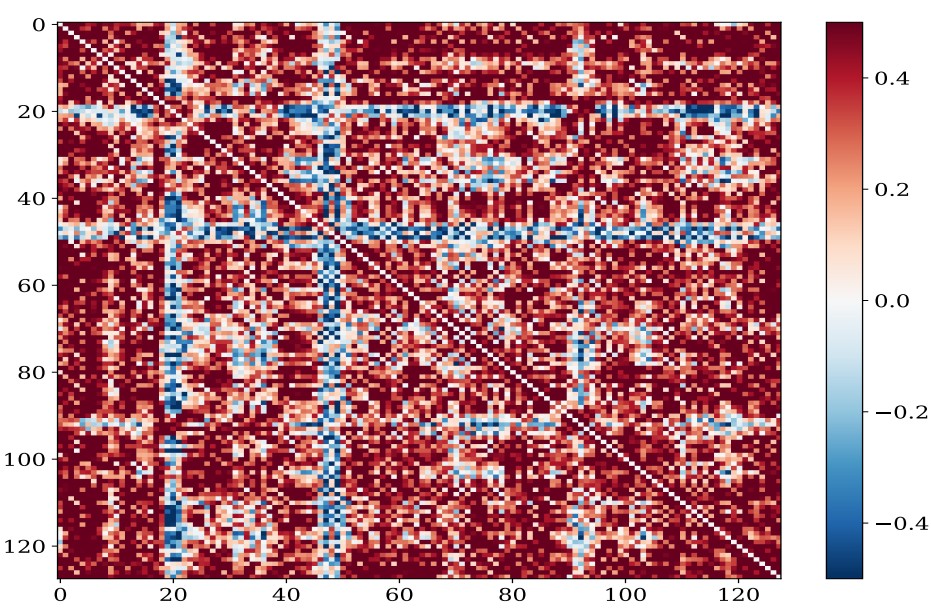

Figure 12: Temporal dependency differences in up-sampling with or without the application of an anti-alias filter on Electricity. Red indicates increased dependency after use anti-alias filter.

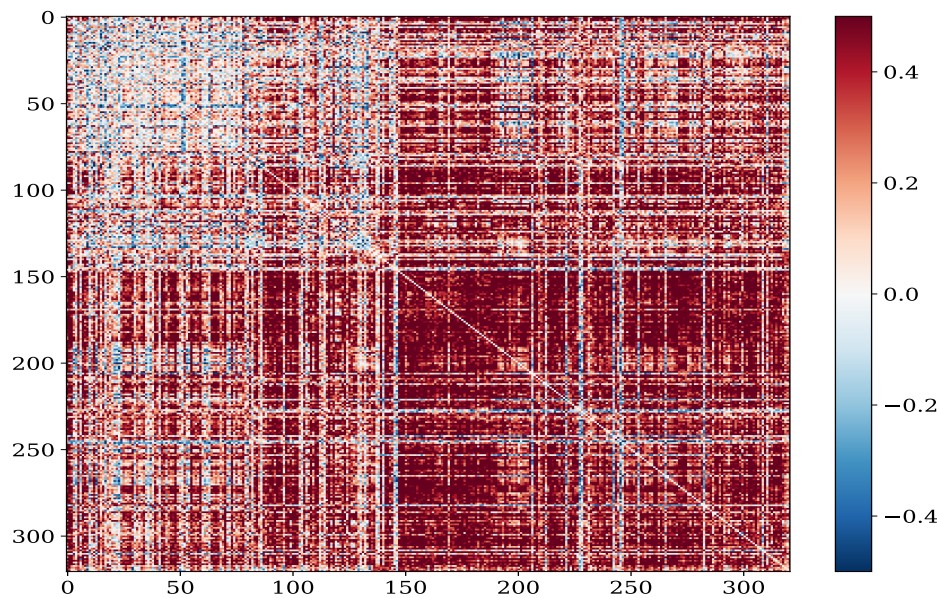

Figure 13: Channel dependency differences in down-sampling with or without the application of an anti-alias filter on Electricity. Red indicates increased dependency after use anti-alias filter.

## I  THE USE OF LARGE LANGUAGE MODELS

Large Language Models were employed as general-purpose assistive tools throughout the research process. Specifically, LLMs were used to polish the language and improve the readability of this manuscript, including refining grammar, improving clarity, and restructuring sentences for better readability. The authors take full responsibility for the content of this paper.

## J  EMPIRICAL VERIFICATION OF ALIASING RISKS VIA SPECTRAL INJECTION ATTACK

### J.1  MOTIVATION AND DATASET CONSTRUCTION

To valid the aliasing risks discussed in the main text and quantitatively evaluate the spectral fidelity of different architectures, we designed a controlled experiment named Spectral Injection Attack. Specifically, we constructed a synthetic dataset based on real-world patterns to simulate scenarios where high-frequency information is crucial for forecasting but highly susceptible to aliasing during downsampling. The dataset is generated by injecting a specific high-frequency component into a base time-series signal:

$$\mathbf{x}_{\text{attack}}(t) = \mathbf{x}_{\text{base}}(t) + A \cdot \cos(2\pi t f_{\text{attack}}), \tag{33}$$

where $\mathbf{x}_{\text{base}}(t)$ represents the background signal, and $\mathbf{x}_{\text{attack}}(t)$ represents the signal after injection attack. We utilized the normalized ETTh1 dataset to retain the chaotic and non-stationary characteristics of real-world time series. $f_{\text{attack}}$ is the injected attack frequency, set to $0.38$ Hz (assuming a unit sampling rate $f_s = 1.0$ Hz). $A$ is the amplitude of the injected signal, set to $2.5$, making the high-frequency pattern distinct in both time and frequency domains. According to the Nyquist-Shannon Sampling Theorem, the new sampling rate becomes $f'_s = f_s/2 = 0.5$ Hz, and the corresponding Nyquist frequency becomes $f'_{\text{Nyq}} = 0.25$ Hz. Since $f_{\text{attack}}(0.38 \text{ Hz}) > f'_{\text{Nyq}}(0.25 \text{ Hz})$, naive downsampling methods, e.g., average pooling or strided convolution without filtering, theoretically guarantee **Spectral Aliasing.** The high-frequency component will be indistinguishably folded into a spurious low frequency $f_{\text{alias}} = |0.5 - 0.38| = 0.12$ Hz. This experiment aims to test whether models can disentangle and preserve this high-frequency information or succumb to aliasing and smoothing effects.

### J.2  EXPERIMENTAL SETTINGS

To benchmark spectral fidelity, we conducted a comprehensive comparison between DMANet and seven representative baselines spanning diverse architectures, including MLP-based (TimeMixer Wang et al. (2024a), TimeXer Wang et al. (2024b)), Linear Decomposition (DLinear Zeng et al. (2023)), Transformer (iTransformer Liu et al. (2024)), Frequency-domain (FreTS Yi et al. (2024b), FilterNet Yi et al. (2024a)), and KAN-based (TimeKAN Huang et al. (2025)) models. All experiments were standardized with a fixed look-back window and prediction horizon of $L = T = 96$, employing a downsampling factor of $s = 2$ for multi-scale architectures and utilizing **MSE Loss** for optimization. To rigorously quantify the fidelity of signal reconstruction in the frequency domain, we denote the amplitude spectra of the ground truth and predicted signals as $S_{\text{true}}(f) = |\mathcal{F}(\mathbf{Y})|$ and $S_{\text{pred}}(f) = |\mathcal{F}(\hat{\mathbf{Y}})|$, respectively, which serve as the basis for our spectral evaluation metrics. We introduced two specific metrics:

- **Spectral Distortion (SD)**: This metric measures the overall structural divergence between the predicted and ground truth spectra. To focus on shape rather than absolute scale, we calculate the Euclidean distance between the normalized amplitude spectra:

$$\text{SD} = \sqrt{\sum_k \left( \frac{S_{\text{pred}}(f_k)}{\sum_j S_{\text{pred}}(f_j)} - \frac{S_{\text{true}}(f_k)}{\sum_j S_{\text{true}}(f_j)} \right)^2} \tag{34}$$

  A lower SD indicates that the model has successfully reconstructed the frequency patterns without introducing significant noise or aliasing artifacts.

- **High-Frequency Capture (HFC)**: This metric specifically quantifies the preservation of the injected high-frequency component. We define a local frequency window $\Omega = [f_{\text{attack}} - \delta, f_{\text{attack}} + \delta]$ centered at the attack frequency (with $\delta = 0.02$ Hz) and calculate the energy ratio:

$$\text{HFC} = \frac{\sum_{f \in \Omega} S_{\text{pred}}(f)^2}{\sum_{f \in \Omega} S_{\text{true}}(f)^2} \times 100\% \tag{35}$$

An HFC value close to $100\%$ indicates perfect disentanglement and reconstruction of the high-frequency signal. Values significantly lower than $100\%$ imply signal loss due to smoothing or aliasing, while values exceeding $100\%$ indicate spectral overshoot or instability.

### J.3    RESULTS AND OBSERVATIONS

As illustrated in Figure.5, DMANet demonstrates superior spectral fidelity, achieving a near-perfect HFC of 99.8% and the lowest SD of 0.0062, which validates the effectiveness of our ESR-based anti-aliasing filter in cleanly disentangling high-frequency signals. In stark contrast, FilterNet and DLinear suffered from severe signal attenuation with HFCs of only 76.9% and 82.8% respectively, confirming that their inherent pooling or moving average mechanisms function as aggressive low-pass filters that irreversibly erode critical high-frequency features. Meanwhile, TimeKAN exhibited spectral instability; despite a high energy capture, its excessive HFC (114.6%) and elevated SD (0.0206) indicate significant spectral overshoot and spurious oscillations arising from unconstrained non-linear fitting. Furthermore, while FreTS and TimeXer managed to capture the target frequency relatively well (103.6% and 95.8% HFC), their SD values remained over three times higher than that of DMANet, revealing the introduction of substantial background noise and aliasing artifacts during reconstruction.

