# OpenReview forum: "A Dynamic Multiscale Anti-Aliasing Network for Time Series Forecasting"
_ICLR.cc/2026/Conference — Submitted to ICLR 2026_

### Official Review · Reviewer_b74H · 2025-10-28

**Soundness:** 2
**Presentation:** 2
**Contribution:** 2
**Rating:** 2
**Confidence:** 5

**Summary:**

This paper addresses the challenge of modeling complex, multiscale temporal patterns in time series, which are often entangled and can lead to inaccurate predictions if not properly handled. The authors argue that conventional downsampling operations in multiscale convolutional architectures can introduce aliasing, folding high-frequency components into low frequencies and degrading feature quality. To tackle this, they propose a Decomposition-Prevention-Fusion (DMANet) framework, which aims to mitigate aliasing during downsampling and better disentangle time-series features across scales. The method introduces mechanisms for both pre-emptive prevention and post-hoc suppression of aliasing. Experiments on multiple benchmarks show modest performance improvements while maintaining a parameter-efficient design.

**Strengths:**

1. The paper highlights the often-overlooked issue of spectral aliasing in multiscale downsampling for deep learning–based time series analysis and proposes DMANet to address it, with experiments showing modest performance gains.
2. The architecture follows a clear Decomposition–Prevention–Fusion design, which is easy to understand and implements hierarchical multiscale feature extraction in a systematic way.
3. The experiments cover multiple benchmarks, demonstrating the method’s applicability across different datasets and providing a reasonable parameter-efficient design.

**Weaknesses:**

1. Unclear problem motivation. The paper mixes up the concept of temporal patterns, which are normal structural features in time series (like trends, cycles, or sudden changes), with aliasing in chaotic systems. It treats the interaction among these patterns as some kind of “noise-like interference,” which is conceptually confusing. In chaotic systems, the complexity mainly comes from deterministic nonlinearity, not random aliasing.
2. Limited novelty. The proposed DMANet architecture mainly relies on standard downsampling and upsampling operations. Using anti-aliasing (low-pass) filters before downsampling is a well-established practice in signal processing, not a new technical contribution. Overall, the design feels more like a reorganization of existing components than a fundamentally new method.
3. Weak performance gains. The improvements over baselines are generally within 1% (e.g. ), which is quite marginal. Moreover, the results don’t include any statistical validation (mean ± variance or multiple runs), making it hard to judge robustness.
4. Uninformative ablation study. The contributions of individual modules are not clearly distinguishable. The ablations do not convincingly show that the proposed design specifically addresses the claimed aliasing issue, so the experiments don’t strongly support the main claim.
5 Inconsistent baselines. It is unclear why the baselines differ across Table 1 and Table 2. Using inconsistent baselines makes it hard to fairly compare the results and assess the method’s effectiveness.

**Questions:**

1. Could you provide stronger experimental evidence showing that the proposed method actually mitigates prediction errors caused by spectral aliasing? Right now, it is not clear whether the model effectively addresses this issue.
2. Include statistical results (e.g., mean ± variance over multiple runs) to demonstrate the robustness of your reported performance improvement.
3. Provide more convincing evidence that each component of the model contributes meaningfully. The current ablation study does not clearly show how the proposed modules address the claimed aliasing problem.

---

> ### Author Response · Authors · 2025-11-24
> **Author Feedback for Reviewer b74H - Part 1**
>
> > ### **W1: Unclear problem motivation.**
>
> We greatly appreciate you for your insights regarding chaotic systems. We fully agree with your definition: the complexity in real-world time series stems primarily from deterministic non-linearity, and temporal patterns (such as trends, cycles, and abrupt changes) are meaningful structural features, not random noise.
>
> To resolve the confusion, we wish to clarify the specific motivation of this paper: **We are not addressing the inherent frequency coupling within the chaotic system itself, but rather the aliasing artificially introduced by the model's downsampling operations (Model-Induced Aliasing).**
>
> 1. **Aliasing Stems from the Operation, Not the System.**
>
>      When multi-scale models (e.g., CNNs or TimeMixer) perform downsampling via strided convolution or pooling to expand the receptive field, they must adhere to the Nyquist-Shannon sampling theorem. If they do not, high-frequency patterns in the original signal, regardless of whether they arise from chaotic dynamics, are erroneously folded into lower frequency bands.
> 2. **Clarifying Noise-like Interference.**
>
>       The terms "noise-like interference" or "spurious patterns" used in our paper do **not** refer to the high-frequency dynamics of the original signal. Instead, they strictly refer to the **mathematical artifacts** generated by this improper downsampling process. The original high-frequency abrupt changes are meaningful. After downsampling, these high-frequency changes appear as inexplicable fluctuations in the low-frequency band. It is these **algorithmically induced low-frequency fluctuations which do not exist in the physical system that we classify as interference.**
>
> Therefore, the goal of DMANet is **not** to filter out the system's inherent complex patterns. On the contrary, by strictly following signal processing principles, our objective is to ensure that the model preserves these normal structural features faithfully, preventing them from being distorted into misleading artifacts during the feature extraction process.
>
> > ### **W2: Limited novelty.**
>
> We thank the reviewer for the comment. We fully acknowledge that anti-aliasing filtering is a cornerstone of classical signal processing. However, the core contribution of DMANet is not reinventing the mathematics of filtering, but **systematically addressing a fundamental yet long-overlooked flaw in the specific domain of Deep Learning for Time Series Forecasting (TSF).**
>
> Regarding the concern about "reorganization of existing components," we wish to highlight three substantive technical innovations:
>
> 1. **Bridging a Critical Blind Spot in TSF Research**
>
>       The Nyquist-Shannon theorem is systematically ignored in current SOTA TSF models (e.g., TimeMixer, SCINet, MICN). Existing downsampling methods (e.g., strided convolutions) rely solely on data-driven learning, operating under the false assumption that the model can automatically manage aliasing. Our **Spectral Injection Attack (Figure 5)** proves this assumption wrong: SOTA models suffer from severe aliasing and distortion. By rigorously re-integrating these classical principles into a deep learning framework, DMANet corrects this design bias.
> 2. **The ESR Mechanism: Architecture-Aware & Dynamic Filtering**
>
>       DMANet does not simply apply a fixed low-pass filter before the network. We propose the **Equivalent Sampling Rate (ESR)** mechanism, which is an **architecture-aware** design. It dynamically calculates the information bottleneck based on the specific configuration of each layer (Kernel size $k$, Stride $s$, Channel ratio $C_{\text{out}}/C_{\text{in}}$). It derives a layer-specific, precise cutoff frequency without manual hyperparameter tuning. This distinguishes DMANet from traditional static preprocessing, embedding signal constraints into the neural network's structural design. Meanwhile, our ablation study (**Table 7** in the paper) provides more evidence. We benchmarked DMANet against both heuristic filters (Max, Random) and classical filters (Ideal, Chebyshev, Gaussian, Butterworth). The results demonstrate that **DMANet consistently outperforms these established methods** (e.g., on the ILI dataset, DMANet achieves an MSE of 1.763, surpassing the best classical alternative, Butterworth, at 1.940).
> 3. **Spectral Fidelity**
>
>       While the downsampling is noted, our contribution equally relies on the **Frequency-Domain Upsampling**. Unlike standard linear interpolation, we employ frequency-domain zero-padding. This is mathematically equivalent to Sinc interpolation using the full Fourier basis, offering the optimal solution for lossless reconstruction of band-limited signals within a deep learning pipeline.
>
> In summary, DMANet is a theoretically grounded framework that effectively bridges the gap between Signal Processing principles and Modern Deep Learning architectures, solving a specific, demonstrable problem (aliasing) that holds back current SOTA models.

---

> ### Author Response · Authors · 2025-11-24
> **Author Feedback for Reviewer b74H - Part 2**
>
> > ### **W3: Weak performance gains.**
>
> We respectfully disagree with the assessment that performance gains are "generally within 1%" and that there is a "lack of statistical validation." We believe this impression stems from a focus on specific saturated benchmarks, overlooking the model's substantial breakthroughs in challenging tasks and the comprehensive statistical evidence provided in the Appendix.
>
> 1. **Performance Improvements Across Extensive Domains and Tasks**
>
> Our method demonstrates state-of-the-art performance across diverse scenarios ( Long-term, Short-term, Noise-injected) and domains including Electricity, Weather, Transportation, Health, Web, Market, Energy, Society, and Finance.
>
> - **Complex Short-term Forecasting:** Contrary to the "within 1%" observation, DMANet achieves huge leaps in short-term forecasting, particularly on non-stationary data. For instance, it improves by 9.4% on ILI ( MSE 1.763 vs. 1.947 ), 12.4% on COVID-19 ( MAE 0.670 vs. 0.765 ), and 14.7% on Unemp ( MSE 0.064 vs. 0.075 ). These margins are practically significant.
> - **Long-Sequence Extrapolation Capability:** DMANet consistently outperforms SOTA baselines in both standard (L=96) and extended lookback (L=720) settings. In the challenging long-term modeling, DMANet maintains superior stability over models like ModernTCN, demonstrating exceptional generalization and extrapolation capabilities in capturing long-range dependencies.
>
> **2. Comprehensive Statistical Validation & Robustness Proof (Appendix D.1 & F)**
>
> - **Statistical Significance (5 Runs):** We respectfully point out that statistical validation is already explicitly provided in **Appendix D.1 (Table 10)**. We conducted experiments using 5 different random seeds and reported the Mean ± Standard Deviation. Significance tests confirm that DMANet outperforms the runner-up (TimeMixer) with a confidence level of >99%, proving the gains are not random fluctuations.
> - **Structural Robustness:** Beyond random seeds, we evaluated the model's resilience via noise injection experiments **(Table 24)**. DMANet maintains stable performance even in high-noise environments where baselines degrade significantly, further proving its robustness.
>
> > ### **W4: Uninformative ablation study.**
>
> We sincerely appreciate the reviewer’s constructive criticism. We agree that standard performance metrics alone are insufficient to prove that the improvements stem specifically from mitigating aliasing. To address this, we have strengthened our ablation study in three dimensions: (1) Disentangling Module Contributions; (2) Spectral Verification; and (3) Comparative Superiority.
>
> 1. **Disentangling Module Contributions (Expanded Table 5)**
>
> We extended the ablation study to isolate the contributions of Anti-Aliasing Downsampling and Frequency-Domain Upsampling.
>    - Base vs. w/o-Post: Even with traditional linear upsampling, adding our ESR Filter consistently improves performance. This proves the downsampling module provides a distinct gain by preventing high-frequency corruption, independent of the upsampling strategy.
>    - Base vs. w/o-Pre: Replacing linear interpolation with our frequency upsampling yields significant gains, confirming the value of high-fidelity reconstruction.
>    - The full DMANet combines both to achieve the best performance, validating the "Prevention-Fusion" framework.
>
> 2. **Direct Evidence of Addressing Aliasing (New Figure 5 & Appendix J)**
>
>   - To convincingly demonstrate that these gains result from specifically addressing aliasing, we conducted a Spectral Injection Attack. We injected a high-frequency signal (0.38Hz) that theoretically guarantees aliasing under downsampling and proposed two metrics: Spectral Distortion (SD), which measures the Euclidean distance between predicted and actual spectral distributions, and High-Frequency Capture (HFC), which quantifies the model's ability to preserve the injected signal energy.
>
>   - While baselines like TimeMixer suffered from high Spectral Distortion (SD > 0.024) and TimeKAN showed spectral overshoot (HFC > 100%), DMANet achieved near-perfect signal preservation (HFC 99.8%) with the lowest distortion (SD 0.0062). This provides direct physical evidence that our design effectively prevents the folding of artifacts.
>
> 3. **Superiority over Existing Filters & Robustness (Table 7 & Table 24)**
>
> Finally, we verified that our design is superior to simply applying standard filters:
>    - Architecture-Awareness (Table 7): DMANet outperforms heuristic (Max/Random) and classical filters (Ideal/Chebyshev/Gaussian/Butterworth). This confirms that our ESR mechanism, which dynamically calculates cutoffs based on the network's specific kernel size $k$, stride $s$, channel ratio $C_{\text{out}}/C_{\text{in}}$, is more effective for deep learning architectures than static classical filters.
>    - Robustness Under Noise (Table 24): In scenarios with noise, DMANet maintains performance stability while baselines degrade significantly.

---

> ### Author Response · Authors · 2025-11-24
> **Author Feedback for Reviewer b74H - Part 3**
>
> > ### **W5: Inconsistent baselines.**
>
> We thank the reviewer for raising this point regarding baseline consistency. We wish to clarify that the selection of baselines for Table 1 and Table 2 was driven by adherence to established experimental protocols and the principle of fair comparison, rather than arbitrary selection.
>
> 1. **Adhering to Different Evaluation Protocols for Fairness**
>
> Different SOTA models are optimized for different input contexts. To ensure we compared DMANet against the strongest version of each competitor, we followed the original settings reported in their respective papers:
>
> - Table 1 (Standard Setting, $L=96$): This follows the classic benchmark protocol established by TimesNet/iTransformer. Most models are optimized and reported under this setting. We compared DMANet against these standard baselines to verify general effectiveness.
> - Table 2 (Long-Context Scaling, $L=720$): This specific setting addresses the **Scaling Law** in forecasting. Newer models like TVNet, TSLANet, and ModernTCN are specifically designed to leverage long lookback windows, whereas some older architectures degrade or were not originally evaluated in this regime. To ensure a rigorous evaluation of DMANet's long-range modeling capability, we specifically selected these strong long-context baselines for Table 2.
>
> 2. **Ensuring Comprehensive Comparison (Appendix D)**
>
> We agree that a consistent comparison is valuable. Therefore, to address concerns about selectivity, we have provided the more detailed comparison results in **Appendix D (Table 11, Table 12, Table 13, Table 14)**. These tables report the performance of many available baselines.  Crucially, DMANet consistently achieves state-of-the-art performance in these comprehensive tables, regardless of which subset of baselines is chosen.
>
> In summary, the split between Table 1 and Table 2 aims to benchmark DMANet against the most relevant and competitive SOTA models for each specific task (standard vs. long-context), ensuring the comparison is both fair and challenging.
>
> > ### **Q1: Provide stronger experimental evidence.**
>
> We thank the reviewer for this crucial question. To provide strong experimental evidence that DMANet specifically mitigates prediction errors **caused by spectral aliasing**, we present a multi-level verification approach that establishes a direct causal link:
>
> 1. **Direct Physical Verification via Spectral Injection (Figure 5 & Appendix J)**
>
> We designed the Spectral Injection Attack to simulate a scenario where errors are solely driven by aliasing. We injected a high-frequency signal (0.38 Hz) that theoretically guarantees aliasing under downsampling (s=2). In this controlled setting, any deviation from the ground truth spectrum constitutes a prediction error caused by aliasing artifacts. As shown in **Figure 5**, baselines like TimeMixer exhibited high **Spectral Distortion (SD = 0.0248)**, directly quantifying the error introduced by aliasing. In contrast, DMANet achieved the **lowest SD (0.0062)** and near-perfect **High-Frequency Capture (99.8%)**, which proof that DMANet successfully prevents the generation of aliasing artifacts.
>
> 2. **Isolating the Source of Error Reduction (Table 5)**
> - **Stage 1: Prevention (Anti-Aliasing Downsampling)**
>
>    Comparing Base vs. w/o-Post shows that adding the ESR filter reduces error (e.g., Unemp MAE 0.173 -> 0.155). This proves that physically removing aliasing-prone high frequencies reduces error. We provide a detailed proof for ESR in Appendix.B
> - **Stage 2: Reconstruction (Frequency-Domain Upsampling)**
>
>    Only avoiding aliasing is not enough, we must also prevent distortion during reconstruction. Comparing Base vs. w/o-Pre shows significant gains (e.g., Unemp MAE  0.173 -> 0.161). Standard linear interpolation acts as a crude low-pass filter that degrades spectral structure. Our frequency-domain method ensures lossless reconstruction of the preserved features, further reducing error.
>
> 3. **Qualitative Evidence: Removing Spurious Correlations (Figure 4)**
>
> To explain why removing aliasing reduces error, we visualize the downstream impact on feature learning. Comparing feature maps with and without the filter (Figure 4 and Appendix H), we observe that aliasing artifacts manifest as spurious, dense correlations across irrelevant time steps (appearing as messy textures). DMANet’s anti-aliasing operation filters out these unreliable high-frequency components, resulting in a significantly sparser and cleaner dependency structure. This qualitative evidence confirms that the error reduction stems from the model shifting its focus from aliasing noise to robust, structural temporal patterns.
>
> By combining **physical verification** (elimination of artifacts), **ablation isolation** (direct error reduction), and **feature visualization** (cleaner patterns), we confirm that the performance gains are not accidental, but the direct result of effectively addressing the spectral aliasing problem.

---

> ### Author Response · Authors · 2025-11-24
> **Author Feedback for Reviewer b74H - Part 4**
>
> > ### **Q2: Demonstrate the robustness.**
>
> We thank the reviewer for emphasizing the importance of statistical robustness. We respectfully point out that **statistical results are already provided in Appendix D.1 (Table 10)** of the submission. We conducted independent experiments using 5 different random seeds.The results are reported in the format of Mean ± Standard Deviation for both MSE and MAE. We performed statistical significance tests comparing DMANet against the second-best model (TimeMixer). The results confirm that DMANet’s performance improvement is statistically significant with a confidence level of >99% across the evaluated datasets.
>
> > ### **Q3: Provide more evidence for each component.**
>
> Thank you for this critical suggestion. We agree that demonstrating how each module specifically addresses aliasing requires a deep dive into the underlying mechanisms. To this end, we elucidate the mechanism following the model's actual data processing pipeline: **RevIN (Stabilization) 、 Anti-Aliasing Downsampling (Prevention) 、 Frequency-Domain Upsampling (Reconstruction).**
>
> **1. RevIN**
>
> Consistent with other frequency-domain works like FilterNet and FITS, spectral stability is a prerequisite for effective filtering. As detailed in our response to Reviewer **3p8m**, the time-domain mean is strictly equivalent to the frequency-domain **DC component $\mathcal{X}[0]$**. Non-stationarity causes the DC component to fluctuate violently, leading to **Spectral Leakage** that contaminates low frequencies. RevIN stabilizes the DC component, ensuring the subsequent anti-aliasing filter operates on a clean spectrum rather than one polluted by leakage artifacts. This creates the necessary low-leakage environment for subsequent modules.
>
> **2. Anti-Aliasing Downsampling**
>
> Built upon a clean spectrum, this module prevents high-frequency corruption during resolution reduction.
>
> **(1) Theoretical Mechanism**
>
> Standard strided convolution performs "blind" decimation, mathematically guaranteeing aliasing if high frequencies exist. Our module enforces **strict adherence to the Nyquist-Shannon Theorem**. By dynamically calculating the **Equivalent Sampling Rate (ESR)**, it strictly band-limits the input to match the actual sampling capability defined by the network architecture. This fundamentally ensures that **no frequency component exists that could fold into a lower band** during the subsequent downsampling. Specifically, we posit that true information fidelity is determined by the **synergistic interaction** of three key parameters:
>
> - **Stride ($s$):** This is the direct cause of downsampling and the primary source of aliasing risk, effectively lowering the Nyquist limit.
> - **Kernel ($k$):** It defines the temporal receptive field . As shown in **Figure 3**, a kernel larger than the stride (k≥s) has the ability to see and integrate information from time steps that would otherwise be discarded between strides, thus compensating for temporal information loss.
> - **Channel expansion ($c >1$):** This creates feature space to preserve information that can no longer be stored in the compressed time dimension, transferring it to the channel dimension (Dimension-domain Compensation, see **Figure 3**).
> - The ESR formula (Appendix B) rigorously captures this interaction. It is not an empirical heuristic but a mathematical quantification of the Information Bottleneck, defining the maximum bandwidth preservable without structural loss.
>
> **(2) Experimental Evidence**
>
> - **Isolation & Architecture-Awareness:** Comparing Base vs. w/o-Post (Table 5) proves that adding the ESR filter reduces error even with standard upsampling. Furthermore, the ESR mechanism operates in an architecture-aware manner. Experiments show it outperforms static classical filters (Table 7) and remains effective under high-noise conditions (Table 24).
> - **Physical Verification:** In the Spectral Injection Attack (Figure 5), baselines without this design (e.g., TimeMixer) suffer from high Spectral Distortion (SD > 0.024). DMANet achieves 99.8% High-Frequency Capture and 0.0062 SD, physically proving it effectively intercepts aliasing artifacts.
>
> **3. Frequency-Domain Upsampling**
>
> - Finally, to fuse multi-scale features, resolution must be restored without introducing new artifacts. Linear interpolation acts as a rough low-pass filter that distorts the spectrum. Our method uses frequency-domain zero-padding. According to Fourier Transform properties, this operation is mathematically equivalent to ideal Sinc interpolation in the time domain. It reconstructs the signal under the explicit assumption of zero energy in high-frequency regions. This guarantees that spectral structures from different scales remain intact during resolution restoration, ensuring no new, spurious artifacts are created during fusion (see Appendix G).
> - Comparing Base vs. w/o-Pre (Table 5) shows that replacing linear interpolation with our method yields significant gains.

---

### Official Review · Reviewer_3p8m · 2025-10-29

**Soundness:** 3
**Presentation:** 3
**Contribution:** 2
**Rating:** 6
**Confidence:** 3

**Summary:**

This paper addresses the aliasing problem in multi-scale time series modeling by proposing a dynamic anti-aliasing scheme. The core idea is to apply adaptive low-pass filtering in the frequency domain, while also employing frequency-domain zero-padding during the upsampling stage to ensure signal reconstruction fidelity. Experiments on multiple datasets validate the method's effectiveness.

**Strengths:**

1. The motivation is clear, and the paper is well-structured, logically progressing from the challenge of aliasing to the proposed method and its experimental validation.
2. The design is simple and effective while also demonstrating good efficiency.
3. The experimental analysis is thorough, with detailed ablations on specific design choices like ESR and the "embedding first" variant.

**Weaknesses:**

1. Lack of Direct Evidence for Anti-Aliasing: The paper's central claim is "reducing aliasing." However, it lacks a quantitative analysis of how much aliasing is caused by existing methods and how much the proposed method reduces it.
2. Incomplete Ablation: The current ablation study fails to disentangle the individual contributions of the anti-aliasing downsampling and the band-limited upsampling modules. To properly attribute the performance gains, the study must be extended to isolate each component's effect. Specifically, it should include comparisons such as: (A). Proposed Downsampling + Traditional Upsampling vs. (B). Traditional Downsampling + Traditional Upsampling. This comparison would quantify the net benefit of the proposed downsampling method alone.

**Questions:**

1. The ESR derivation seems tightly coupled to your specific DWConv+PWConv architecture. Have you considered whether this anti-aliasing approach can be generalized as a "plug-in" to enhance other downsampling-based models?
2. The model's performance drops sharply without RevIN (w/o RevIN). Does this indicate a strong coupling between RevIN and the proposed anti-aliasing module? If so, what could be the underlying reason?

---

> ### Author Response · Authors · 2025-11-24
> **Author Feedback for Reviewer 3p8m - Part 1**
>
> > ### **W1: Lack of direct evidence**
>
> We agree that the quantitative evidence for our anti-aliasing claims is important. While quantifying aliasing on standard real-world datasets is challenging due to the lack of a ground truth for downsampled signals, we have designed a new Spectral Injection Attack experiment to provide precisely the evidence requested. The detailed methodology is now in **Appendix J**, with key results visualized in the new **Figure 5**.
>
> **1. Experimental Design: Creating a Quantifiable Aliasing Scenario**
>
> We injected a signal with a frequency of 0.38Hz into the ETTh1 dataset. Under a downsampling factor of 2 (Nyquist frequency =0.25Hz), this signal is theoretically guaranteed to alias. This setup provides a clear ground truth to measure deviations. We proposed two specific metrics to quantify the extent of aliasing:
>
> - **Spectral Distortion (SD):** Measures the Euclidean distance between the predicted and ground-truth spectra. A higher SD directly indicates severe spectral deformation caused by aliasing or noise.
> - **High-Frequency Capture (HFC):** Quantifies the percentage of the injected high-frequency energy preserved by the model.
>
> **2. Quantitative Results on Aliasing Reduction**
>
> The results quantitatively demonstrate the extent of aliasing reduction. While TimeMixer captures high-frequency energy (HFC = 94.6%), it suffers from a high SD of 0.0248, indicating that the signal is corrupted by aliasing artifacts. **In contrast, DMANet achieves the lowest SD (0.0062) with near-perfect energy preservation (HFC of 99.8%).** This corresponds to an approximate 75% reduction in spectral distortion compared to TimeMixer, validating that DMANet achieves precise signal disentanglement rather than mixing aliased noise. Additionally, we observed that TimeKAN exhibits severe spectral overshoot (abnormal HFC of 114.6%) driven by aliasing, whereas DMANet effectively eliminates these spurious oscillations. Crucially, DMANet does not merely act as a low-pass filter, but preserves critical information while minimizing aliasing-induced distortion.
>
> > ### **W2: Incomplete ablation.**
>
> We sincerely thank you for this insightful comment. We agree that properly attributing performance gains requires isolating the effects of the anti-aliasing downsampling and the frequency upsampling. To address this, we extended our ablation study (now presented as **Table 5** in the revised manuscript) based on the framework you suggested:
>
> - Base: A generic baseline, using the sub-optimal combination from our original Table 4 (standard convolutional downsampling + linear interpolation upsampling + ReVIN).
> - w/o-Post: Proposed Downsampling + Traditional Upsampling (linear interpolation).
> - w/o-Pre: Traditional Downsampling(standard convolution) + Proposed Upsampling.
> - DMANet: Proposed Downsampling + Proposed Upsampling.
>
> **(1) Isolating the Benefit of Proposed Downsampling (Base vs. w/o-Post)**
>
> As specifically requested, comparing Base and w/o-Post reveals the net benefit of our downsampling method alone. Even with traditional linear interpolation, adding the Pre-Sampling ESR filter consistently improves performance (e.g., Unemp MAE 0.173 -> 0.155). By filtering out frequencies that would otherwise cause corruption, we ensure the model learns from a cleaner representation of macroscopic trends at coarser scales, rather than from aliasing artifacts distorted by high-frequency noise.
>
> **(2) Isolating the Benefit of Proposed Upsampling (Base vs. w/o-Pre)**
>
> Similarly, comparing Base and w/o-Pre isolates the contribution of our frequency-domain upsampling. We observed that replacing linear interpolation with our method yields significant gains (e.g., Unemp MAE  0.173 -> 0.161). This highlights the critical role of high-quality signal reconstruction. Our frequency-domain interpolation, based on the Fourier basis, theoretically enables a lossless restoration of temporal patterns.
>
> **(3) Deep Insight on Synergy (Why DMANet is Best)**
>
> Finally, comparisons against the full DMANet reveal the deeper rationale:
>
> - Pre-Sampling Filtering (w/o-Pre vs. DMANet): Adding the filter further improves performance (Unemp MAE 0.161 -> 0.146). This quantifies the value of proactively preventing aliasing: we prevent high-frequency information from corrupting low-frequency features from the outset, allowing the model to learn untainted macroscopic trends rather than artifacts.
>
> - Post-Sampling Interpolation (w/o-Post vs. DMANet): The improvement here is substantial (e.g.,  Unemp MAE 0.155 ->0.146). This highlights the critical role of high-quality reconstruction. When fusing features from different scales (e.g., hourly fluctuations, daily cycles, weekly trends), our method restores them to the same resolution without loss, enabling precise fusion of long- and short-term dependencies.

---

> ### Author Response · Authors · 2025-11-24
> **Author Feedback for Reviewer 3p8m - Part 2**
>
> > ### **Q1: The Plug-in Potential of the Anti-Aliasing Module.**
>
> We sincerely thank the reviewer for this constructive suggestion. We agree that demonstrating the generalization capability of our anti-aliasing approach is crucial. While the ESR derivation in the paper illustrates the specific case of DW+PW convolution, the core principle, dynamically calculating the cutoff frequency based on the Effective Sampling Rate, is theoretically applicable to any downsampling operation.
>
> **Action:**
> To empirically validate this, we conducted a new plug-in experiment selecting three representative downsampling-based models with distinct architectures: MLP-based (TimeMixer) and CNN-based (MICN, SCINet). We integrated our ESR-based Pre-Sampling Filter directly before their respective downsampling layers without modifying their original backbones.
>
> As detailed in **Table 6**, the results demonstrate that this plug-in design yields consistent reductions in both MSE and MAE. For instance, SCINet on the Power dataset achieves an MSE reduction of 0.023 (from 0.205 to 0.182). The improvements are observed across different model types, confirming that our anti-aliasing approach is highly generalizable. By virtue of being architecture-aware, the ESR mechanism analyzes the specific downsampling bottleneck (e.g., stride) of the host model to calculate a theoretically optimal cutoff frequency. This proactively eliminates high-frequency components that would otherwise cause spectral folding, ensuring that these baselines extract features from clean, alias-free representations, thereby enhancing their predictive capability.
>
> > ### **Q2: The Role and Necessity of RevIN.**
>
> As the reviewer observed, removing RevIN leads to a performance drop. We confirm that a strong coupling exists between the two. The fundamental reason is that **RevIN stabilizes the Zero-Frequency Component, creating the necessary and clean spectral environment for the anti-aliasing filter.** We explain this using the frequency-domain derivation:
>
> From a frequency perspective, the mean in the time domain is strictly equivalent to the zero-frequency component in the frequency domain. Given an input signal $x[n]$ of length $N$, its Discrete Fourier Transform (DFT) $\mathcal{X}[k]$ is obtained as: $ \mathcal{X}[k] = \frac{1}{N} \sum\_{n=0}^{N-1} x[n]e^{-2\pi jnk/N}$. Setting $k=0$ to examine the zero-frequency component, the equation simplifies to: $ \mathcal{X}[0] = \frac{1}{N} \sum\_{n=0}^{N-1} x[n]e^{0} = \frac{1}{N} \sum\_{n=0}^{N-1} x[n] $. According to the equations above, the mean in the time domain $\frac{1}{N} \sum x[n]$ is equal to the zero-frequency component $\mathcal{X}[0]$ , which is referred to as the DC (Direct Current) component in the frequency domain.
>
> Based on this mathematical fact, the coupling arises from:
>
>   **1. Frequency Manifestation of Non-stationarity:** Non-stationarity (distribution shift) implies that the mean of the data fluctuates drastically. Without RevIN, this directly results in an extremely unstable and large $\mathcal{X}[0]$.
>
>   **2. Interference with Anti-Aliasing (Spectral Leakage):** In FFT over a finite window, a dominant and unstable $\mathcal{X}[0]$ causes energy to leak into adjacent low-frequency bands (Spectral Leakage).
>
>   **3. The Coupling Mechanism:** Our anti-aliasing filter is designed to precisely preserve valid low-frequency information. However, without RevIN, the low-frequency band becomes contaminated by leakage from the unnormalized DC component. The filter cannot distinguish between the true signal and leakage artifacts, rendering it ineffective.
>
> Therefore, RevIN acts as a crucial **spectral preprocessing** step. By removing the interference from the zero-frequency component, it ensures that the anti-aliasing filter processes faithful frequency pattern. In addition, this practice of RevIN is a common prerequisite in frequency-domain models like FITS and FilterNet to ensure effective and reliable filtering.

---

> > ### Comment · Reviewer_3p8m · 2025-11-27
> >
> > Thank you for the detailed response. Your replies have completely resolved my concerns, so I have raised the scores for soundness, presentation, and confidence. However, I am keeping the overall rating unchanged, mainly because I believe the contribution of this work in addressing aliasing caused by downsampling remains somewhat limited.

---

### Official Review · Reviewer_X4XE · 2025-10-31

**Soundness:** 2
**Presentation:** 4
**Contribution:** 2
**Rating:** 4
**Confidence:** 4

**Summary:**

This paper proposes DMANet—a "Decomposition-Prevention-Fusion" architecture—to solve the aliasing problem in multiscale time series forecasting, where conventional downsampling distorts high-frequency components into spurious low-frequency noise. It equips DMANet with Multiscale Convolutional Downsampling (for capturing temporal and inter-channel dependencies) and an Anti-Aliasing Operation (pre-sampling ESR-based filtering and post-sampling frequency-domain interpolation) to preserve feature fidelity. Through experiments on benchmarks like ETT, PEMS, and COVID-19 datasets, the paper shows DMANet achieves state-of-the-art performance in long/short-term forecasting, matching large models while being more parameter-efficient.

**Strengths:**

1. This study proposes DMANet, a "Decomposition-Prevention-Fusion" architecture, to address aliasing in multiscale time series forecasting—where traditional downsampling distorts high frequencies into spurious low-frequency noise.
2. It designs ESR-based pre-sampling filtering (dynamic Nyquist frequency calculation) and post-sampling frequency-domain interpolation to preserve feature fidelity, with convolutional downsampling for efficient dependency modeling.
3. This study validates DMANet on datasets like ETT, PEMS, and COVID-19, showing it achieves SOTA in long/short-term forecasting (matching large models) while being parameter-efficient, with ablations confirming core components’ necessity.

**Weaknesses:**

1. This study’s labeling of the operation post-embedding as "downsampling" is debatable. Though the operation reduces temporal resolution (via stride-based convolution), it differs from traditional signal processing downsampling— which typically directly reduces sample points of raw signals. Here, the operation acts on embedded latent features (after Linear projection and normalization), blurring the line with "feature dimension reduction" rather than strict signal downsampling, lacking explicit clarification on this conceptual distinction .
2. This study’s heavy reliance on frequency-domain MAE loss raises doubts about its method’s intrinsic effectiveness. Ablation shows removing this loss (using MSE instead) degrades performance significantly (e.g., Unemp dataset MAE rises from 0.146 to 0.166) . This over-dependence suggests the model’s "anti-aliasing advantage" may be overly tied to the loss function, rather than the proposed architecture (like ESR filtering) alone, weakening confidence in the method’s core design merit.

**Questions:**

See weaknesses.

---

> ### Author Response · Authors · 2025-11-24
> **Author Feedback for Reviewer X4XE - Part 1**
>
> > ### **W1: Clarification on the terminology "downsampling".**
>
> Thank you for your insightful feedback. We agree that our operation differs from traditional signal downsampling on raw inputs, as it acts on latent features. However, we justify our terminology and architectural choices based on the following three key points:
>
> **1. Alignment with the Paradigm:** Following recent advancements in computer vision [1-3], many methods apply classical sampling theory to internal network representations. In this emerging paradigm, feature maps are treated as high-dimensional signals. Reducing their resolution is mathematically equivalent to downsampling and is strictly governed by the Shannon-Nyquist theorem. We use the term "downsampling" to maintain consistency with these works and to explicitly highlight the risk of aliasing within the feature space.
>
> **2. Distinction from General "Feature Dimension Reduction":** Although we project the raw sequence into a latent space, our architecture enforces a strict sequential topology on this dimension via Positional Encodings and Convolutional Layers. This effectively constructs an ordered "Latent Time Axis." Unlike standard dimension reduction (e.g., PCA or MLP) which treats features as unordered vectors, our operation reduces the resolution of this organized sequence. Therefore, it is not merely compression but signal downsampling, where preserving the frequency content of latent patterns is critical.
>
> **3. Rationale for Embedding-First:** As detailed in the newly added **Appendix G.1**, operating in this latent space is necessary. Directly downsampling raw signals restricts the reconstruction basis, leading to severe spectral distortion during upsampling due to limited and deformed basis vectors. By projecting into a high-fidelity latent space first, we can perform anti-aliasing downsampling and ideal Fourier-based interpolation, thereby fundamentally avoiding spectral distortion.
>
> **Action:** We have added **Appendix G.1** to explicitly define "Latent Signal Downsampling" and illustrate the spectral distortion issue inherent in raw-signal approaches.
>
> [1] Alias-free latent diffusion models: Improving fractional shift equivariance of diffusion latent space. CVPR 2025.
>
> [2] FrequencyLowCut Pooling - Plug & Play against Catastrophic Overfitting. ECCV 2022.
>
> [3] When Semantic Segmentation Meets Frequency Aliasing. ICLR 2024.

---

> ### Author Response · Authors · 2025-11-24
> **Author Feedback for Reviewer X4XE - Part 2**
>
> > ### **W2: Concerns regarding the  frequency-domain loss.**
>
> Thank you for this critical analysis. While we acknowledge that the frequency-domain loss enhances performance, we respectfully disagree that the model's effectiveness relies *solely* on it. We clarify the intrinsic merit of our architecture from two perspectives: **Mechanism Orthogonality** and **Empirical Evidence under MSE Loss**.
>
> **1. Theoretical Orthogonality: Feature Fidelity and Unbiased Optimization**
>
> As rigorously derived in the newly added **Appendix B.6**, the total forecasting error can be decomposed into two independent terms: $\mathcal{E}\_{\text{total}} \approx \mathcal{E}\_{\text{feat}} + \mathcal{E}\_{\text{loss}}$.
>
> - **DMANet (Internal Architecture)** minimizes the **Feature Representation Error ($\mathcal{E}\_{feat}$)**. By applying the ESR-based filter before downsampling, DMANet physically removes the spectral aliasing term $\sum\_{k \neq 0} F(X)(\omega - k\omega\_s')$, ensuring that the latent features $h$ mathematically approximate the ideal, uncorrupted signal $h\_{\text{ideal}}$.
> - **Frequency Loss (External Objective)** minimizes the **Optimization Bias ($\mathcal{E}\_{\text{loss}}$)**. It leverages the spectral decorrelation property of the Fourier transform to mitigate the gradient bias caused by temporal autocorrelation.
>
> Removing the frequency loss increases $\mathcal{E}\_{\text{loss}}$, leading to the observed performance drop in the ablation study. However, this does not negate DMANet's unique contribution to minimizing $\mathcal{E}\_{\text{feat}}$ (eliminating aliasing): without this architectural design, no loss function can recover information destroyed by spectral folding. We employ FreDF as the default setting because it intuitively aligns with our design philosophy: DMANet **preserves** spectral fidelity physically, while FreDF **supervises** it explicitly. They form a synergistic effect, but this does not imply that the architecture lacks independent value.
>
> **2. Direct Verification via MSE Loss**
>
> First, to rigorously quantify the fidelity of signal reconstruction in the frequency domain, we propose two novel metrics, named  **Spectral Distortion (SD)** and **High-Frequency Capture (HFC).** Given these, to isolate the architectural contribution and prove DMANet's intrinsic anti-aliasing capability, we highlight our **Spectral Injection Attack experiment (Appendix J)**, which was trained **using only the standard MSE Loss** (as stated in Line 2523).
>
> - **Results:** Even without the aid of frequency loss, DMANet achieved a near-perfect **High-Frequency Capture (99.8%)** and the lowest **Spectral Distortion (0.0062).**
> - **Comparison:** In contrast, baselines like FilterNet and DLinear suffered severe signal attenuation (~76-82% HFC), while TimeKAN exhibited spectral instability. This demonstrates that the ability to disentangle and reconstruct high-frequency signals without aliasing is an **intrinsic property of the DMANet architecture**, completely independent of the loss function.
>
> This architectural robustness is further proven in our ablation study. For instance, on the Unemp dataset, DMANet with a standard MSE loss (0.076, Table 4) still outperforms TimeMixer (0.094) and rivals FilterNet (0.079), whose results are in Table 5.
>
> **Action:** We have added Appendix B.6 and Appendix J to provide the detailed information of the orthogonality between the anti-aliasing mechanism and the loss function.

---

### Official Review · Reviewer_pqCd · 2025-11-01

**Soundness:** 3
**Presentation:** 3
**Contribution:** 2
**Rating:** 6
**Confidence:** 4

**Summary:**

This paper identifies that existing downsampling procedures are prone to severe aliasing, and proposes a novel multiscale convolutional downsampling framework DMANet built around a Decomposition–Prevention–Fusion architecture to perform principled downsampling and effectively disentangle time-series features.

**Strengths:**

1. This paper introduces novel mechanisms for preemptive prevention and post-hoc suppression of aliasing, implemented explicitly within the multiscale decomposition process.
2. DMANet features a parameter-efficient design.
3. Extensive experiments on both long- and short-term forecasting tasks demonstrate that DMANet achieves competitive performance against strong baselines.

**Weaknesses:**

1. The motivation needs more rigorous substantiation. It would be helpful to include experiments on real or synthetic datasets that (i) empirically verify the aliasing risks claimed in the Introduction and (ii) demonstrate that existing decomposition methods, such as TimeMixer and TimeMixer++, are insufficient to resolve these issues.
2. As presented through dependency modeling, Figure 4 seems insufficient to demonstrate that the anti-aliasing filter effectively mitigates aliasing.
3. The efficiency experiments in Table 21 should include additional lightweight baselines (e.g., FilterNet, TimeKAN) and larger-scale datasets (e.g., Traffic, Electricity).
4. The baselines described in Section 4.1 should have their results presented and analyzed in the main text rather than relegating part of them to the appendix, as this split is confusing.

**Questions:**

1. Figure 2 requires a more comprehensive and detailed caption to facilitate understanding.

---

> ### Author Response · Authors · 2025-11-24
> **Author Feedback for Reviewer pqCd**
>
> > ### **W1: The Validation of Aliasing Risks.**
>
> Thank you for your constructive suggestions. To rigorously substantiate the aliasing risks discussed in the paper and demonstrate the insufficiency of existing methods, we designed a controlled experiment termed **Spectral Injection Attack** (detailed in **Appendix J** and the newly added **Figure 5**).
>
> Specifically, we injected a high-frequency signal ($0.38 \text{Hz}$) into the real-world ETTh1 dataset. Under a downsampling factor of 2, the new Nyquist frequency becomes $0.25 \text{Hz}$. The injected signal exceeds the Nyquist limit, theoretically inducing aliasing. We proposed two quantitative metrics: **Spectral Distortion (SD)**, which measures the Euclidean distance between the predicted and actual spectral distributions, and **High-Frequency Capture (HFC)**, which quantifies the model's ability to preserve the injected signal energy.
>
> The experimental results reveal two primary failure modes in existing SOTA methods:
>
> **1. Aliasing and Spurious Oscillations:** Both TimeMixer and TimeKAN employ downsampling operations. Although TimeMixer attempts to preserve information via multi-scale mixing (achieving an HFC of 94.6%), its high Spectral Distortion (SD = 0.0248, which is **4x higher** than DMANet) indicates a failure to distinguish between aliasing artifacts and the true signal, leading to significant spectral deformation. TimeKAN exhibits severe **Spectral Overshoot** (HFC reaching 114.6%), demonstrating that without anti-aliasing constraints, the KAN network erroneously amplifies aliased signals into spurious oscillations.
>
> **2. Over-smoothing and Information Loss:** Although DLinear does not involve explicit downsampling, its core moving average mechanism acts as an uncontrollable low-pass filter, suggesting that it misidentifies the injected high-frequency signal as noise and filters it out. FilterNet similarly suffers from signal attenuation (HFC of only 76.9%), confirming that it lacks specific anti-aliasing designs fail to preserve high-frequency features.
>
> DMANet is the only model that strikes a balance between these extremes. **By leveraging the ESR-guided filter to precisely disentangle high-frequency signals prior to downsampling, DMANet achieves a near-perfect HFC of 99.8% and the lowest SD of 0.0062.** This empirically verifies that DMANet  addresses the aliasing issues inherent in downsampling models while overcoming the over-smoothing limitations of traditional decomposition methods.
>
> > ### **W2: The Efficacy of the Anti-aliasing Filter.**
>
> Thank you for your valuable point that quantitatively demonstrating the mitigation of aliasing is necessary.
>
> We wish to clarify that the primary objective of Figure 4 is not to directly verify spectral fidelity, a task now addressed by the Spectral Injection Attack in the newly added Figure 5 and Appendix J. **Instead, Figure 4 serves to visualize the downstream impact of the anti-aliasing filter on the model's feature learning process.**
>
> By comparing the scenarios with and without the filter (showed in Figure 4 and Appendix H), we observe that aliasing artifacts often manifest as spurious, dense correlations across irrelevant time steps or channels. DMANet’s anti-aliasing operation filters out these unreliable high-frequency components, resulting in a sparser and cleaner dependency structure. This visualization complements our quantitative results, explaining why removing high-frequency noise helps the model focus on robust, structured temporal patterns, thereby enhancing forecasting performance.
>
> > ### **W3: The Supplemented Efficiency Experiments.**
>
> We sincerely appreciate this valuable suggestion.
>
> We have significantly expanded the efficiency analysis in Appendix E.2 (Table 23) of the revised manuscript. Specifically, for new baselines, we added FilterNet and TimeKAN to benchmark against distinct lightweight design paradigms. For large-scale dataset, we included the Electricity dataset (321 variates) to evaluate model performance under high-dimensional settings.
>
> We have expanded Appendix E.2 (Table 23) to include FilterNet, TimeKAN, and the Electricity dataset. Results on Electricity show DMANet achieves the best MSE (0.146) with only 73.95M MACs. This is orders of magnitude more efficient than PatchTST (539.68G) and 6× lighter than TimeKAN (456.50M), while significantly outperforming lightweight baselines like FilterNet in accuracy. This confirms DMANet’s superior scalability on large-scale data. These results confirm that DMANet is not only a lightweight solution but also a highly scalable architecture for complex, high-dimensional tasks.
>
> > ### **W4 & Q1: On Improving Presentation Clarity.**
>
> Thank you for pointing these out. We have updated our manuscript to clarify the baselines selected in the main body and appendix. In addition, we have provided more detailed information for Figure 2 to help readers get better understandings.

---

### Author Response · Authors · 2025-12-03
**The Summary of Author Feedback**

Dear Area Chair,

We sincerely thank the reviewers for their recognition and constructive suggestions. We appreciate that they acknowledged our strengths in terms of a novel research perspective filling a critical gap (Reviewer pqCd, X4XE, 3p8m, b74H), an innovative and easy-to-understand architecture (Reviewer pqCd, b74H), parameter efficiency (Reviewer pqCd, X4XE, 3p8m, b74H), extensive experimentation (Reviewer pqCd, 3p8m), and broad applicability (Reviewer pqCd, X4XE, b74H). During the rebuttal phase, we have further strengthened the manuscript by adding rigorous empirical verification, deepening the theoretical analysis, and expanding comparative experiments. The major revisions (highlighted in **red**) are summarized below:

**1. Rigorous Verification of Aliasing Risks (Reviewer pqCd, 3p8m, b74H)**

Addressing concerns regarding the direct evidence of aliasing, we **added a Spectral Injection Attack experiment (Fig. 5 and Appendix J)** and proposed two quantitative metrics: **Spectral Distortion (SD)** and **High-Frequency Capture (HFC)**. The results demonstrate that SOTA models like TimeMixer fail to distinguish aliasing noise, and TimeKAN suffers from spectral overshoot. In contrast, DMANet achieves near-perfect signal disentanglement.

**2. Theoretical Strengthening & Concept Clarification (Reviewer X4XE,  b74H)**

*   **Orthogonality Proof (Appendix B.6):** We provided a mathematical derivation proving that our architectural gain is orthogonal to the frequency-domain loss. Furthermore, the model maintains excellent performance when trained using only MSE Loss.
*   **Clarification on Core Motivation:** We clarified that DMANet addresses **"aliasing artifacts" artificially introduced by downsampling operations, a problem overlooked by existing methods, rather than the system's inherent noise.** We also clarified the definition of "Downsampling" in **(Appendix A.3, G.1)**.

**3. Experimental Expansion (Reviewer pqCd, 3p8m)**

*   **Complete Ablation (Table 5):** Expanded ablation studies to strictly isolate and verify the independent contributions of Pre-Sampling Filtering and Post-Sampling Interpolation.
*   **Generalizability (Table 6):** Demonstrated the **Plug-in** capability of our ESR filter.
*   **Efficiency Analysis (Table 23):** Added tests on the high-dimensional **Electricity** dataset (321 variates).

**4. Clarification on Experimental Rigor (Reviewer pqCd, b74H)**

*   **Performance Gains:** Our original draft covered Long-term( $L=96$ and $L=720$ ), Short-term, and Univariate scenarios. We clarified the misconception of marginal gains: in challenging tasks like Short-term forecasting on non-stationary data, DMANet achieves substantial improvements.
*   **Fairness of Baselines:** Due to the Scaling Law, we explained that the different baselines in Table 1 and Table 2 were selected to adhere to the **original experimental protocols** of those models, ensuring fair comparison against their strongest reported settings.
*   **Statistical Significance:** We reiterated that statistical validation over **5 random seeds** ($>99\%$ confidence) was **already present in the original submission (Appendix D.1)**.

**Conclusion**

Finally, we emphasize that spectral aliasing is a universal yet long-overlooked problem in deep learning-based time series analysis. It affects any architecture involving downsampling. By explicitly addressing this fundamental issue, DMANet offers a theoretically grounded and widely applicable solution to enhance feature fidelity in forecasting models.



Best regards,

The Authors

---

### Meta-Review · Area_Chair_NX43 · 2026-01-16

**Summary:**

W1. Several of the reviewers pointed out gaps in the paper's motivation due to an unconvincing illustration of the aliasing problem and how other models fail in this case.

W2. It is unclear how much of the performance gain is due to the MAE loss rather than the MSE loss.

W3. Limited novelty, due to reliance of upsampling/downsampling operations.

W4. Inconsistent baselines (different baselines used for different experiments).

W5. The performance gains are marginal.

Other questions were raised concerning the technical details of the method, specifically how the downsampling is achieved (and why that is not simply an embedding, but actual downsampling), disentangling "the individual contributions of the anti-aliasing downsampling and the band-limited upsampling modules" and the importance of RevIN. These questions were appropriately resolved by the author response.

**Reviewer Concerns:**

W1. The authors added a spectral injection attack experiment in their response. This goes some of the way to illustrate the problem, however, the setting appears to be rather limited. Also, the fact that this was added only after almost all the reviewers pointed it out does not bode well for how the problem was presented in the first place. With no reviewer responses allowed, there is really no way of ascertaining whether the added experiment could be considered sufficient motivation.

W2. The authors provided an additional experiment which, at a first read, seems to address this concern.

W3. The authors explained that the method goes beyond simply applying a low-pass filter as a pre-processing step or as a fixed component that is tracked on at the start of the architecture. While they do a good job explaining how their mechanism works and how it is different than the aforementioned options, it is still a relatively simple way of solving the problem.

W4. The authors argued that the different s.o.t.a. models were selected as appropriate for eacb setting depending on the sequence length. I consider this to be an acceptable response.

W5. The authors attempted to address this in the response, but it reads as a reiteration of existing numbers, and, yes, the deviations are low indicating statistical significance, but the difference in average performance is still small. Overall, the response to this issue is not very convincing.




All in all, while the paper solves a potentially challenging problem, the motivation in the submitted version was lacking a demonstration of the methodological gap (added in the response), it employs a relatively simple solution with relatively unimpressive numerical results. For these 3 reasons (W1, W3, W5 only partially resolved), the paper is not ready for publication in its current form.

**Reviewer Scores:**

I have no way of knowing how the reviewers would have changed their scores.

---

### Decision · Program_Chairs · 2026-01-26

Reject